# I/O-efficient iterative matrix inversion with photonic integrated circuits

Minjia Chen [1], Yizhi Wang[1], Chunhui Yao[1], Adrian Wonfor [1], Shuai Yang[1], Richard Penty[1] & Qixiang Cheng [1,2] ✉

Photonic integrated circuits have been extensively explored for optical processing with the aim of breaking the speed and energy efficiency bottlenecks of digital electronics. However, the input/output (IO) bottleneck remains one of the key barriers. Here we report a photonic iterative processor (PIP) for matrix-inversion-intensive applications. The direct reuse of inputted data in the optical domain unlocks the potential to break the IO bottleneck. We demonstrate notable IO advantages with a lossless PIP for real-valued matrix inversion and integral-differential equation solving, as well as a coherent PIP with optical loops integrated on-chip, enabling complex-valued computation and a net inversion time of 1.2 ns. Furthermore, we estimate at least an order of magnitude enhancement in IO efficiency of a PIP over photonic single-pass processors and the state-of-the-art electronic processors for reservoir training tasks and multiple-input and multiple-output (MIMO) precoding tasks, indicating the huge potential of PIP technology in practical applications.

The input/output (IO) data movement in a processor typically has limited bandwidth and consumes considerable energy, leading to performance bottlenecks in processing systems that receive input data and store output data through an attached host with memory[1–4]. The IO issue becomes particularly severe when the problem size exceeds the capacity of the processor, a situation that is often encountered in practice. Thus, enhancing the computation-to-IO (C-to-IO) ratio becomes a key focus, i.e. increasing the number of implemented operations for a given input/output data size. In computation intensive tasks such as matrix multiplications and inversions, where multiple operations are performed on each piece of input data, the C-to-IO ratio can be improved by reusing the input data. The tensor processing unit (TPU) exemplifies this approach in accelerating neural network inference tasks. The TPU utilizes a systolic architecture to compute matrix multiplications in parallel by reusing the input data, demonstrating significant speed and energy efficiency improvements over central processing units (CPUs) and graphics processing units (GPUs)[2]. Nevertheless, the challenge of keeping pace with the ever-growing demands of compute-intensive applications has urged people to keep seeking faster and more energy-efficient solutions.

Photonic processors have long been considered as a promising alternative in algebra computations, owing to its inherent high parallelism and low energy consumption[4–7]. Photonic processors have shone through in the field of matrix multiplications, notably been demonstrated in voice recognition[8], image classification[9,10], and optical communication[11]. However, few research has been done on using photonic processors to accelerate matrix inversion, a more computationally expensive operation that is fundamental to many scientific and engineering problems such as numerical computations[12–16], statistics[12,17], wireless communication systems[18–22] and neural network training[23,24]. In most practical applications, the matrices to be inverted are diagonally dominant or sparse, and approximate inversion results are often sufficient for subsequent processing stages. Iterative inversion methods generally outperform direct methods in these scenarios since one can easily balance speed and accuracy by terminating the iterative process after several iterations, while interrupting the inversion process of direct methods midway yields incorrect results[12]. Classical iterative matrix inversion algorithms essentially perform matrix multiplications and additions iteratively. Implementing such algorithms on a widely reported photonic single-pass processor

[1]Centre for Photonic Systems, Electrical Engineering Division, Department of Engineering, University of Cambridge, Cambridge CB3 0FA, UK. [2]GlitterinTech Limited, Xuzhou 221000, China. ✉e-mail: qc223@cam.ac.uk

(PSP)[8,9,25,26] necessitates repetitive inputting and outputting of computation results in each iteration, thus generating heavy IO traffic.

In this paper, we report a photonic iterative processor (PIP) based on reconfigurable photonic integrated circuits for speeding up matrix inversion tasks. The inclusion of optical loopback enables iterative computations for direct matrix inverting with an enhanced I/O efficiency. The processor core is a matrix-vector multiplier comprising Mach-Zehnder interferometer (MZI) units, which reduces the computation overhead by directly encoding the matrix elements on MZI arrays. Table 1 compares the C-to-IO ratios for inverting an $N \times N$ matrix using the iterative Richardson method[12] on four different platforms including the CPU, TPU, photonic single-pass processor (PSP), and our proposed PIP. The Richardson method is assumed to converge after $P + 1$ iterations. According to the table, the PIP exhibits the highest C-to-IO ratio of $N^3 P/(2N^2 + N)$ among the four computing platforms, followed by the PIP, the TPU, and the CPU. In most application scenarios, $P$ is much larger than $N$, indicating an improvement of at least $N$ times in the C-to-IO ratio for PIPs compared to other platforms (See Supplementary 1 for more details).

We demonstrate a lossless reconfigurable PIP system that is capable of directly inverting matrices and solving integral and differential equations. Such a PIP system computes $4 \times 4$ real-valued matrix inversions with an accuracy of >97%, a C-to-IO ratio improvement of up to >7 times, a core energy efficiency of 4.6 MOPS/W (mega operations per second per watt), and a net inversion time of 2.6 μs, which is solely bounded by the length of fibre-based optical loops. The lossless PIP is then reconfigured to numerically solve real-valued integral and differential equations, reaching a mean absolute error of <0.02, and up to >1.8 times C-to-IO ratio improvement. The first coherent PIP with on-chip optical loops is also demonstrated to break the loop-length limitation on the net inversion time. The coherent PIP is demonstrated to operate $2 \times 2$ complex-valued matrix inversions with an accuracy of >98%, a core energy efficiency of 3.6 GOPS/W (giga operations per second per watt), and a net inversion time of 1.2 ns. Its enhancement in C-to-IO ratio reaches up to 2.8 times. Benefiting from the much-reduced IO demand, the proposed PIP is capable of reaching at least an order of magnitude IO efficiency enhancement compared with a single-pass optical processor and the state-of-the-art electronic processors, by emulating MIMO precoding tasks and reservoir training tasks. Our results indicate a promising way towards ever-powerful optical processors that could surpass IO limits.

**Table 1 | A comparison of C-to-IO ratios for inverting an $N \times N$ matrix using the Richardson method[a]**

| | CPU | TPU | PSP | PIP |
|---|---|---|---|---|
| Input data size | $2N^2 \cdot P$ | $N^2 \cdot P$ | $N^2 \cdot P$ | $N$ |
| Output data size | $N^2 \cdot P$ | $N^2 \cdot P$ | $N^2 \cdot P$ | $N^2$ |
| Memory access counts[b] | $3N^2 \cdot P$ | $N^2 \cdot (2P+1)$ | $N^2 \cdot (2P+1)$ | $2N^2 + N$ |
| Number of operations[c] | $N^3 \cdot P$ | $N^3 \cdot P$ | $N^3 \cdot P$ | $N^3 \cdot P$ |
| **C-to-IO ratio** | $N/3$ | $NP/(2P+1)$ | $NP/(2P+1)$ | $N^3 P/(2N^2 + N)$ |

[a]$N$ is the matrix size. $P + 1$ is the number of iterations for the Richardson method to converge.
[b]This includes the number of times a processor needs to fetch input data, store output data, and load weight matrix (the matrix to be multiplied in each iteration). For CPU, the weight matrix is also fetched in a similar way as the input data and is therefore included in the input data size.
[c]Note the computations in the first iteration are neglected in the analyses for two reasons: (1) if the process converges for $P = 1$, then the matrix to be inverted is essentially an identity matrix whose inverse is itself and does not need to be computed. (2) In the first iteration, the algorithm essentially implements $N$ multiply-and-accumulate (MAC) operations, which can be neglected compared to the $N^2$ MAC operations in the following iterations.

## Results

### Photonic iterative processor architecture

The proposed PIP is tailored for matrix inversion problems that cannot be easily solved by traditional single-pass photonic processors. Solving integral and differential equations can be reduced to basic matrix computations including addition, subtraction, multiplication and inversion, which can all be solved optically by iterative algorithms using the PIP. Matrix inversion is computed through the Richardson method:

$$\mathbf{X}^{(k+1)} = (\mathbf{I}_N - \omega \mathbf{A})\mathbf{X}^{(k)} + \omega \mathbf{I}_N \ (k = 0,1,2,\ldots) \quad (1)$$

where $\mathbf{A}$ is an $N \times N$ matrix operand to be inverted, $\mathbf{I}_N$ is the $N \times N$ identity matrix, $\omega$ is a parameter used to adjust the convergence of the inversion algorithm and the matrix operand is encoded in the weight bank via $\mathbf{I}_N - \omega \mathbf{A}$. $\mathbf{X}^{(k+1)}$ and $\mathbf{X}^{(k)}$ are output matrices after $k + 1$ and $k$ iterations, and $\mathbf{X}^{(0)} = \omega \mathbf{I}_N$ is the initial input matrix that initiates the computation. The PIP generates matrix computation results one column at a time. Full-matrix inversion can be realised by using $N$ PIPs or by a multiplexing technique based on a different architecture we proposed[27].

Figure 1 depicts the architecture of the proposed PIP based on Richardson method, with grey arrows indicating signal flows. As highlighted by red arrows, the input pulse (representing the $j^{th}$ column of the initial input matrix, $\omega \mathbf{e}_j$) is generated by modulating the continuous wave (CW) laser in modulators. Upon entering the loop, part of the pulse is split and sent to detectors for outputs readout (one column of $\mathbf{X}^{(k+1)}$). The remaining part first passes an $N \times N$ weight bank with each MZI encoding one element of the $N \times N$ matrix, $\mathbf{I}_N - \omega \mathbf{A}$. Subsequently, the weighted pulses are summed by waveguide couplers to implement a matrix-vector multiplication (MVM). The MVM results (one column of $(\mathbf{I}_N - \omega \mathbf{A})\mathbf{X}^{(k)}$) are then amplified to compensate for any loop loss. Optical filters are attached to remove excess amplified spontaneous emission (ASE) noise. With the switches configured to the "On" state, the "clean" pulses are sent for summation with the input pulse to implement the addition of one column of $(\mathbf{I}_N - \omega \mathbf{A})\mathbf{X}^{(k)}$ and one column of $\omega \mathbf{I}_N$. Then part of the summed signals is split out and detected for outputs. The remaining parts enter the next circulation for iterative computation. Such recursive operation is terminated either when the optical switches are set to "Off" or when the input pulse ends. The PIP is also capable of computing matrix addition and multiplication in a single iteration by proper configuration. Detailed descriptions can be found in the supplementary 5.2.

### Real-valued matrix inversions

A $4 \times 4$ chip is taped out on the Silicon Nitride (SiN) platform, which is used as a processor core to form a lossless PIP system with off-the-shelf components as shown in Fig. 2a. The SiN chip integrates 4 adders, 4 splitters with a $4 \times 4$ sized MZI weighting bank, and their enlarged views are shown in Fig. 2b–d, respectively. Light is coupled in and out of the chip via edge couplers. Complete optical loopback paths are formulated by fibre components. The continuous-wave (CW) laser and optical modulator (Mod) correspond to the laser and modulator blocks in Fig. 1, generating an input vector $\omega \mathbf{e}_j$ that is coupled into the $4 \times 4$ chip. One column of the inverse results is computed each time by launching an optical input pulse to one input port and full matrix inversion is realized by sweeping different input ports (I1–I4). The input pulse is first split into four copies on chip and then gets imprinted by the set of weights. The weighted signals are subsequently summed by 4 adders to perform an MVM, which are then coupled out of the chip and amplified by Erbium-doped fibre amplifiers (EDFAs), followed by bandpass filters (BPFs) to supress ASE noises. The $1 \times 2$ splitters allow part of the optical signals to be collected by the oscilloscope (OSC) after outputs readout in photodetectors (PDs) and transimpedance amplifiers (TIAs), while the remaining part are sent

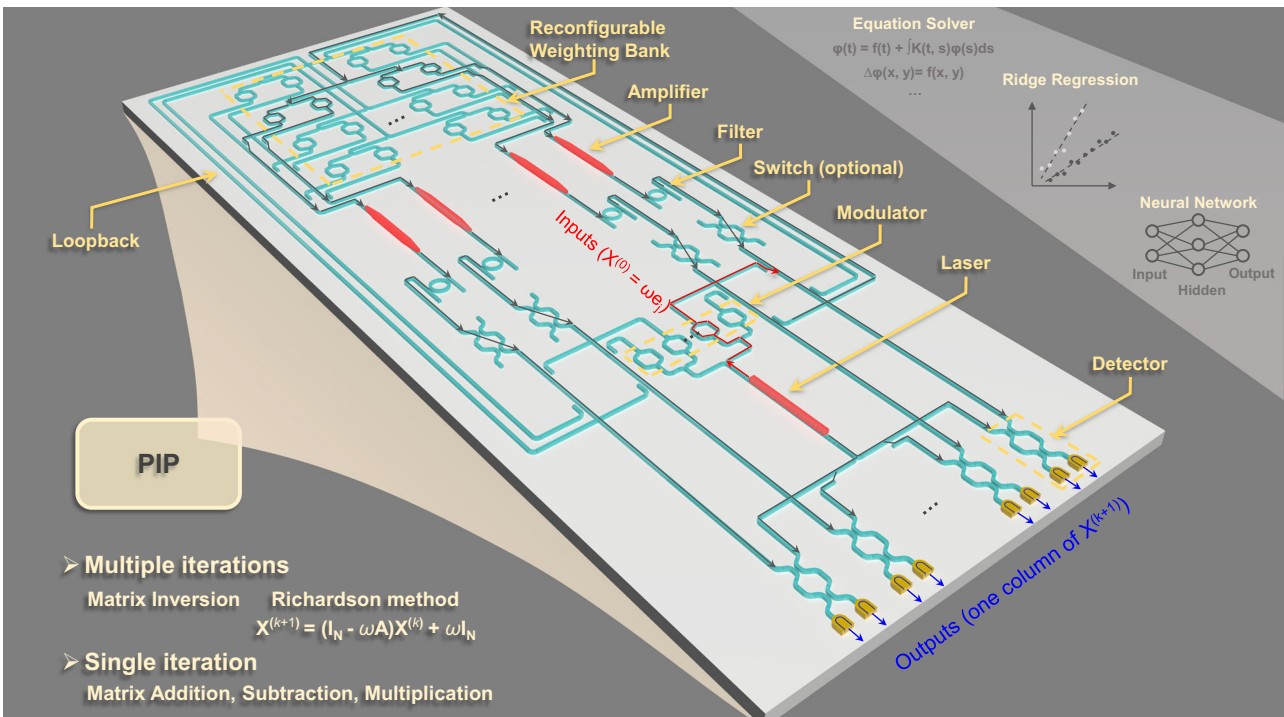

**Fig. 1 | Conceptual figure of the photonic iterative processor (PIP).** Architecture of the proposed iterative photonic processor. The PIP serves as a photonic accelerator for inverting matrices which is widely used in equation solving, statistics, communication systems, artificial intelligence etc. Matrix inversion is solved by the Richardson method, whose results can be obtained by multiple iterations of the light signal in the PIP. Matrix addition, subtraction, and multiplication results can also be computed by a single iteration of the light signal in the PIP. For the lossless PIP system, the 4 × 4 reconfigurable weighting bank, adders and splitters are integrated on-chip. For the coherent PIP system, the 2 × 2 reconfigurable weighting bank, adders, splitters, optical switches, 90° hybrid, and optical loopback are integrated on-chip. Monolithic integrations are possible and separately discussed in the section titled "Photonic integration techniques for a fully integrated PIP". The arrows in the main figure represent the signal flows.

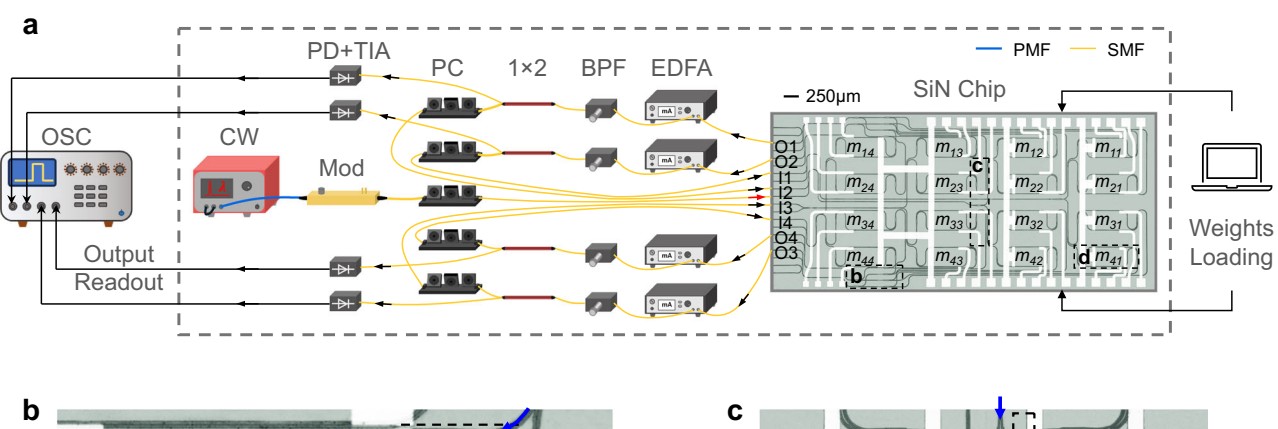

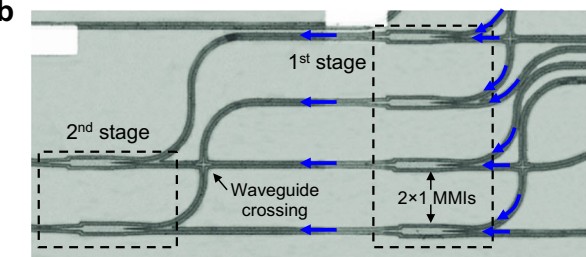

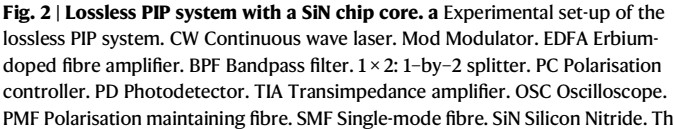

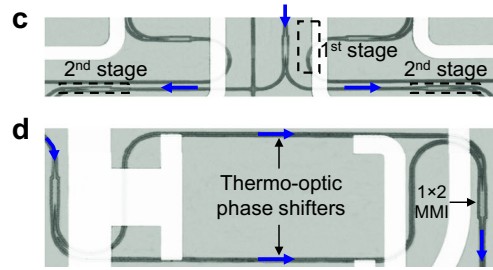

**Fig. 2 | Lossless PIP system with a SiN chip core. a** Experimental set-up of the lossless PIP system. CW Continuous wave laser. Mod Modulator. EDFA Erbium-doped fibre amplifier. BPF Bandpass filter. 1 × 2: 1–by–2 splitter. PC Polarisation controller. PD Photodetector. TIA Transimpedance amplifier. OSC Oscilloscope. PMF Polarisation maintaining fibre. SMF Single-mode fibre. SiN Silicon Nitride. The red and black arrows indicate the signal flows. **b** Enlarged view of two adders comprising two stages of cascaded 2 × 1 multimode interferometers (MMIs). **c** Enlarged view of a splitter consisting of two stages of cascaded 1 × 2 MMIs. **d** Enlarged view of a weight unit comprising of a 1 × 1 thermo-optic (TO) MZI. The blue arrows in (**b**–**d**) indicate the signal flows.

back to the chip via the optical loops. Polarisation controllers (PCs) are used to align the polarisation state of optical inputs to the chip. Electrical control is used for loading matrix weights, setting the modulator and operating outputs readout.

We use the set-up shown in Fig. 2a to demonstrate two real-valued matrix inversion examples (detailed matrix values, $A_1$ and $A_2$, can be found in Supplementary 4.1 and 4.2). As shown in Fig. 3a and d, $A_1$ and $A_2$ are loaded into the weighting bank as $M_1 = I_4 - A_1$ and $M_2 = I_4 - A_2$ respectively (See Methods and Supplementary for matrix weights characterisation). By injecting different unit vectors $X_j^{(0)} = e_j$ ($j = 1,2,3,4$), different columns of the inversed matrices are obtained. Figure 3b and e indicate a very good agreement between the ideal inverse results and the measured inverse results for the two inversion examples, respectively. The evolutions of inversion accuracy of $A_1$ and $A_2$ during convergence are traced and exhibited in Fig. 3c and f, reaching an inversion accuracy of 97.5% and 97.2% respectively.

### Solving real-valued integral and differential equation with matrix inversions

Integral and differential equations offer a powerful tool for quantifying the dynamics of systems that change over time or space, making them widely used in scientific research and engineering. Numerical solutions instead of analytical solutions are usually taken for real-world problems, which is essentially the problem of matrix inversion (See Method and Supplementary 3 and 5). Using the lossless PIP system that operates iteratively to solve integral and differential equations provides an alternative computing paradigm that significantly reduces the demand of data movement. We adopt the system to solve an integral equation (IE, Fredholm integral equation of the second kind, Eq. (2)), a second order ordinary differential equation (2nd order ODE, Eq. (3)) with both using an 8-point discretization, and a partial differential equation (PDE, Poisson equation, Eq. (4)) using a 4-point discretization.

$$f(t) = 1 + \int_{-1}^{1} 0.2\sqrt{t^2 + s^2} f(s) ds, t \in [-1,1] \quad (2)$$

$$d^2/dx^2 [f(x)] - 2x \cdot d/dx [f(x)] - 50f(x) = -1, x \in [-1,1], f(-1) = f(1) = 1 \quad (3)$$

$$\partial^2 u(x,y)/\partial x^2 + \partial^2 u(x,y)/\partial y^2 = -2\pi^2 \sin(\pi x)\sin(\pi y), x,y \in [0,1], \partial_u = 0 \quad (4)$$

An $N$-point discretization corresponds to $N \times N$ matrix inversions for IE and ODE, while it corresponds to $N^2 \times N^2$ matrix inversions for PDE. Block matrix computation techniques are employed to bridge the size of the problem and our chip, which could be readily mitigated by integrating a larger-scale processor on chip. Though using block matrix inversion techniques on a PIP increases the memory access counts, the total memory access counts and hence processing time is still much lower than that of traditional electronic processors and photonic single-pass processors (See Supplementary 6.4 for a detailed analysis). Figure 4a-c showcase the measured solutions based on the chosen discretization resolution, ideal solutions (from a conventional 64-bit digital computer) based on the chosen discretization resolution, and ideal solutions based on a finer discretization resolution, respectively. Mean absolute errors (MAEs = $\frac{1}{n}\sum_{j=1}^{n}|x_j^M - x_j^I|$, $x^M$ is the measured solution, $x^I$ is the ideal solution using the same discretization resolution) are shown at the top of each sub figure, indicating a very good agreement between the measured solutions and ideal solutions.

### Complex-valued matrix inversions

One of the major applications of complex-valued matrix inversion is in MIMO decoding and precoding[19–21], which are fundamental to 5 G/6 G wireless communication systems. These algorithms require the inversion of complex-valued channel matrices, whose dimensions significantly increase owing to growing data demands from an increased number of end-users. Large scale complex-valued matrix inversion is computationally expensive and faces speed and power efficiency limitations in digital electronic computers. Unlike traditional methods that store a complex number as two separate real parts and process each part individually, photonic processors manipulate the amplitude and phase of optical signals simultaneously, enabling truly complex-valued computations with enhanced efficiency. Additionally, the PIP system can further improve the C-to-IO ratio and increase the inversion speed through its iterative optical loopback, significantly reducing IO access.

Here we show a coherent PIP system as shown in Fig. 5a for complex-valued matrix inversions. With optical loopback paths integrated on-chip, stable phase control can be achieved together with ultrafast processing time. Optical switches are integrated on-chip to facilitate device characterisation. Coherent outputs are read out by on-chip 90° hybrids together with off-chip balanced photodetectors (BPDs) and captured in the OSC. Enlarged views of the above integrated components are shown in Fig. 5b–e. EDFAs are used to compensate for coupling loss only and BPFs are used to remove excess ASE noise. We use the coherent PIP system to demonstrate three complex-valued matrix inversion examples (detailed matrix values, $A_3$, $A_4$ and $A_5$, can be found in Supplementary 4.3–4.5). The measured and ideal inversion results for the three matrices including both the amplitude and the phase changes of each element of the inverse matrices are shown in Fig. 6a, c, and e respectively, indicating a good agreement with the ideal process. Figure 6b, d, and f show the evolutions of inversion accuracy of three matrices during multiple iterations. The inversion accuracy reaches 98.8%, 98.3% and 98.9% for $A_3$, $A_4$ and $A_5$, respectively.

### IO advantages of the PIPs

We use saved processing time, $T_{save}$, saved energy consumption, $E_{save}$, improvement in total processing time, $\frac{t_{total\_PSP}}{t_{total\_PIP}}$, and improvement in C-to-IO ratio, $\frac{C\text{-}to\text{-}IO_{PIP}}{C\text{-}to\text{-}IO_{PSP}}$ to quantify the demonstrated IO advantages of the lossless PIP and the coherent PIP. The results are shown in Table 2. $q$ is the number of times a matrix is decomposed before computing on a PIP, $N$ is the PIP size, $t_{loop}$ is the single-iteration's processing time, and $P$ is the number of iterations for convergence. $T_0 = \frac{B}{DR}$ and $E_0 = B \cdot 1$ $pJ/bit$[28] are the processing time and energy consumption of a single IO access, where $B$ is the bit resolution of the data, and DR is the data transfer rate. Details about calculating each specs can be found in Supplementary 6.

According to the table, >7 times and >2 times improvement in the C-to-IO ratio are achieved for $4 \times 4$ real-valued matrix inversions and $2 \times 2$ complex-valued matrix inversions respectively. For solving the integral equation and the ordinary differential equation, $8 \times 8$ matrix inversions are performed on the $4 \times 4$ lossless PIP through one decomposition, still resulting in ~1.8 times improvement in IO efficiency. For solving the partial differential equation, $16 \times 16$ matrix inversions are performed on the $4 \times 4$ lossless PIP through two decompositions yet resulting in ~1.2 times improvement in IO efficiency. These results well verify the IO advantages of the PIP for matrix inversions.

## Discussion

### Processing time (latency) of an $N \times N$ PIP

The processing time of a PIP includes both the core processing time and the IO access time. The core processing time refers to the duration starting from when a signal is launched into the processor until the computation results are ready for acquisition, which is determined by the single-iteration's processing time, $t_{loop}$, and the number of iterations to reach convergence, $P$. The PIP system shown in Fig. 1 generates the matrix inversion results one column a time, corresponding to a core processing time of $t_{core} = t_{loop} \cdot P \cdot N$. Using wavelength multiplexing techniques can reduce the core

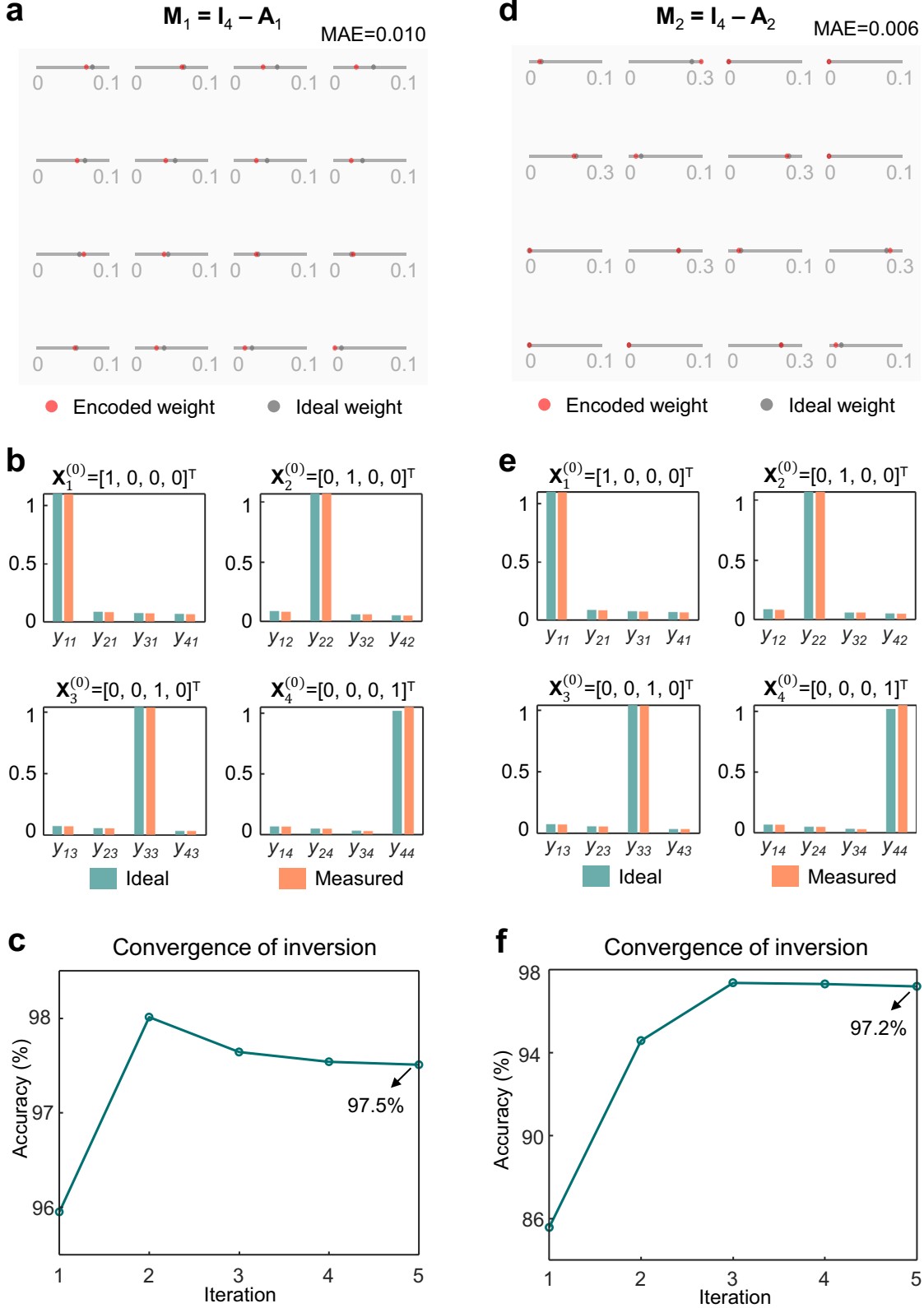

**Fig. 3 | Real-valued matrix inversion examples. a** Ideal and encoded weight matrix $\mathbf{M}_1$. MAE Mean absolute error. MAE $= \frac{1}{16}\sum_{j=1}^{4}\sum_{i=1}^{4}\left|\mathbf{M}^{\text{encode}}(i,j) - \mathbf{M}^{\text{ideal}}(i,j)\right|$. **b** Ideal and measured inverse matrix $\mathbf{A}_1^{-1}$. $\mathbf{X}_j^{(0)}$ is the $j^{th}$ column of the initial input matrix $\mathbf{X}^{(0)}$. **c** Evolution of inversion accuracy of $\mathbf{A}_1$ during convergence. Accuracy = $(1 - ||\mathbf{A}_{\text{meas}}^{-1} - \mathbf{A}_{\text{ideal}}^{-1}||/||\mathbf{A}_{\text{ideal}}^{-1}||) \times 100\%$. **d** Ideal and encoded weight matrix $\mathbf{M}_2$. **e** Ideal and measured inverse matrix $\mathbf{A}_2^{-1}$. **f** Evolution of inversion accuracy of $\mathbf{A}_2$ during convergence.

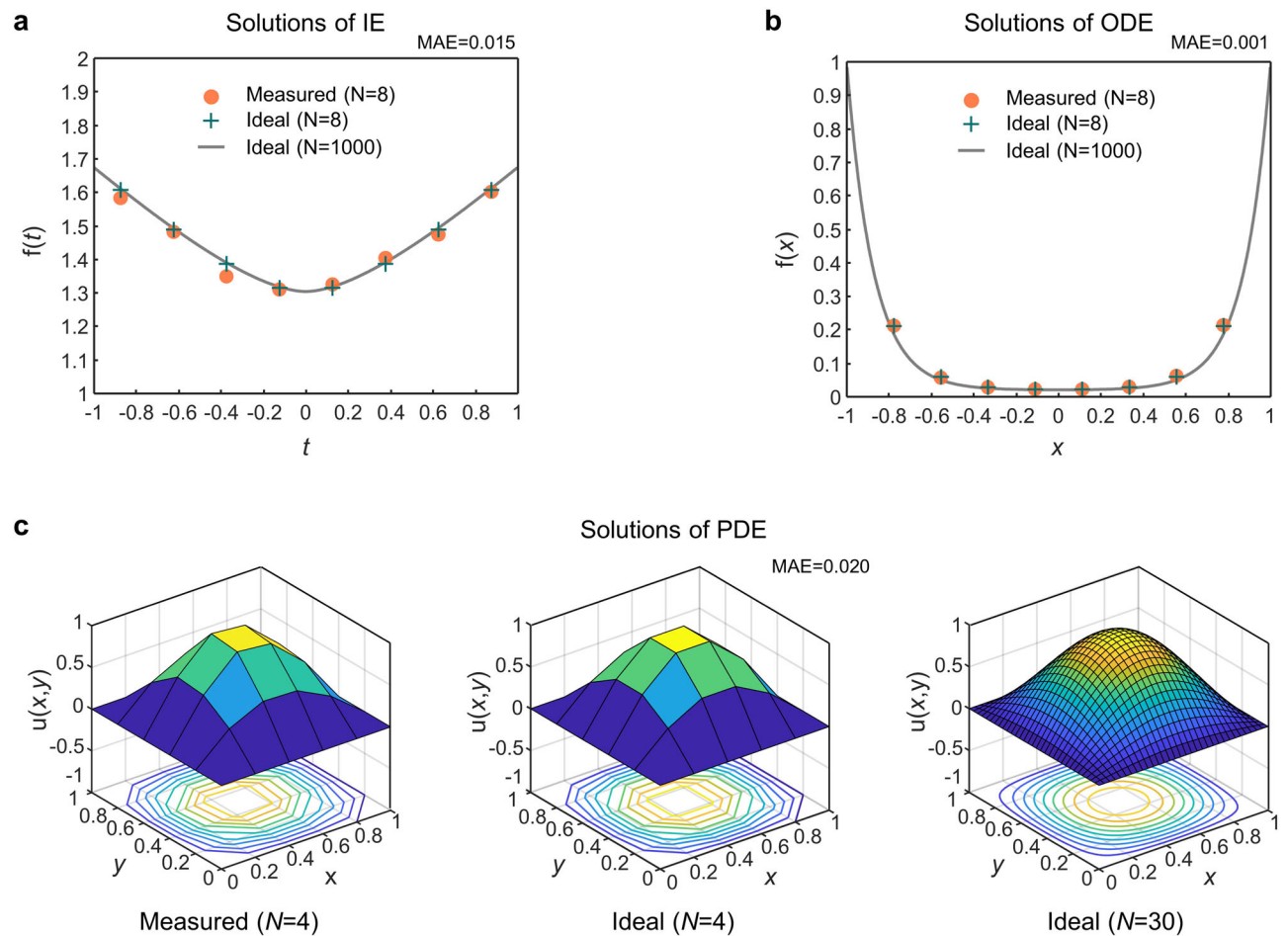

**Fig. 4 | Solving real-valued integral and differential equations. a** Solutions to a Fredholm integral equation of the second kind. **b** Solutions to the 2nd order ordinary differential equation. **c** Solutions to the partial differential equation (Poisson equation). Mean absolute errors (MAEs) are indicated at the top of each sub figure.

processing time to $t_{core\_wdm} = t_{loop} \cdot P$. According to Table 2, the best demonstrated net inversion time, $t_{core}$, are 2.6 μs and 1.2 ns for the 4 × 4 and 2 × 2 cases, respectively. The loop length of the lossless PIP is mainly limited by the long fibre length (~35 m) inside the EDFA, which can be removed by using semiconductor optical amplifiers (SOA) or on-chip gain components. It is worth noting that for the coherent PIP device, additional delay lines (~3.9 cm) are integrated as part of the loopback paths to ease the electrical readout, which can be shortened for a faster computation speed. To explore the future scaling of the PIP, we estimate the loop length and the single iteration's processing time of a fully integrated PIP with sizes ranging from 2 × 2–256 × 256 as shown in Fig. 7a, b. Both the loop length and the single iteration's processing time scale approximately linearly with PIP size.

The IO access time is defined as $t_{IO} = \frac{M \cdot B}{DR}$, where $M$ is memory access counts shown in Table 1, $B$ is the bit resolution of the data, and DR is the data transfer rate. $t_{IO}$ includes the weight matrix loading time, input data fetching time, and output data storing time, and contributes to a heavy burden for digital electronic processors and single-pass photonic processors, while it contributes much less to the total inversion time for the PIP. The total processing (inversion) time of a PIP and a PSP can be estimated according to $t_{total\_PIP} = t_{core\_PIP} + t_{IO\_PIP}$, and $t_{total\_PSP} = t_{core\_PSP} + t_{IO\_PSP}$, respectively. More details can be found in Supplementary 7.2–7.3.

### Inversion accuracy of an $N \times N$ PIP
Inversion accuracy is defined in terms of matrix norm as: $(1 - \varepsilon) \times 100\% = (1 - \|\mathbf{A}_{meas}^{-1} - \mathbf{A}_{ideal}^{-1}\| / \|\mathbf{A}_{ideal}^{-1}\|) \times 100\%$, where $\varepsilon$ is the

inversion error, $\mathbf{A}_{meas}^{-1}$ is the measured or simulated inversion results, and $\mathbf{A}_{ideal}^{-1}$ is the ideal or theoretical inversion results calculated on a 64-bit traditional digital electronic computer. Three main error sources when performing matrix inversions on a fully integrated PIP include: (1) quantization error, (2) ASE noise introduced during amplification, and (3) thermal and shot noise introduced during detection. The estimated inversion accuracy of a PIP with sizes ranging from 2 × 2 to 256 × 256 is shown in Fig. 7c, indicating a relatively good inversion accuracy for a processor size of up to 256 × 256. More details can be found in Supplementary 7.4.

### Energy efficiency of an $N \times N$ PIP
Energy efficiency of the PIP core is defined as: Energy efficiency $= \frac{\text{Number of operations/Second}}{\text{Energy consumption}}$, which is the number of operations per second per energy consumption of the PIP core. The components consuming energy during the matrix inversion process include lasers, modulators, semiconductor optical amplifiers (SOAs), photodetectors (PDs), the MZI weight bank, and optical switches. The energy efficiency of the lossless PIP and the coherent PIP, together with the estimated energy efficiency of the PIP (with and without wavelength multiplexing techniques) and that of the state-of-the-art electronic and photonic processors are shown in Fig. 7d, indicating that the PIP structure is an energy-efficient architecture for matrix inversion tasks. More details can be found in Supplementary 7.5.

### Scalability of the PIP
In the above analyse, the PIP size is limited to 256 × 256. While this is sufficient for most real applications, with no more than 5

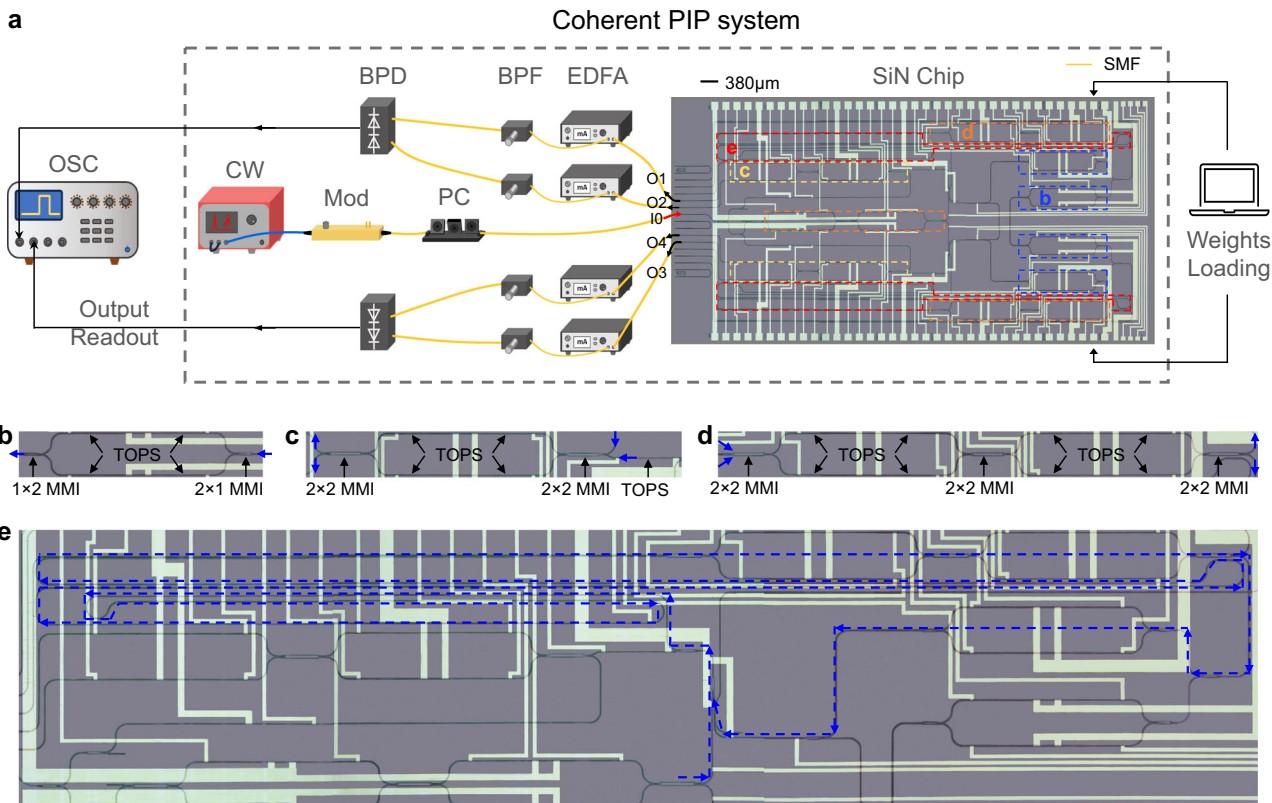

**Fig. 5 | Coherent PIP system with a SiN chip core. a** Experimental set-up of the coherent PIP system. CW Continuous wave laser. Mod: Modulator. EDFA Erbium-doped fibre amplifier. BPF Bandpass filter. 1 × 2: 1-by-2 splitter. PC Polarisation controller. BPD Balanced photodetector. OSC Oscilloscope. SMF Single-mode fibre. SiN Silicon Nitride. The red and black arrows indicate the signal flows. **b** Enlarged view of a complex-valued Mach-Zehnder Interferometer (MZI) weight unit consisting of two 1 × 2 multimode interferometers (MMIs) and two thermo-optic phase shifters (TOPSs) on each arm. **c** Enlarged view of a 90° hybrid which is a 2-in-2-out MZI unit with an extra TOPS at one of the input ports to introduce the 90° phase shift in a traditional 90° hybrid. **d** Enlarged view of an optical switch which is a 2-in-2-out MZI unit with an MZI-type variable splitter replacing one of the two 2 × 2 MMI couplers to ensure a high extinction ratio. The blue arrows (both solid and dashed) in (**b**–**e**) indicate the signal flows.

decompositions (see Supplementary 6 for more details) ensuring the IO advantage for problem sizes up to 8192 × 8192, larger-scale PIPs are still preferable for tackling even more complex problems with more significant IO advantages.

The main constraints for further scaling up the PIP are similar to those for any large-scale photonic integrated circuit (PIC), i.e., the chip insertion loss and the available reticle size of the lithography. As shown in Fig. 7c, for PIPs larger than 128 × 128, good accuracy can still be achieved but requires high input signal powers, indicating that the round-loop loss is not well sustainable for higher-radix crossbar array-based MVM cores. A singular value decomposition (SVD)-based MVM core can be used to reduce signal splitting and addition losses, however, the accumulated insertion loss of the increased building blocks remains as a challenge. In addition, the computation overhead for implementing SVD is non-negligible. Several approaches have been reported to effectively reduce the PIC insertion loss, including the use of a microelectromechanical-system (MEMS) MZI structure[29] and high-index doped silica glass waveguide[30]. Wafer-scale photonic integrated circuits are also demonstrated via inter-reticle waveguides[29], paving the way for very large-scale PIP systems.

### Envisioned IO advantages of the PIP for matrix-inversion-intensive applications

We further evaluate the IO advantages of our PIP in matrix-inversion-intensive problems, i.e., the MIMO precoding task and the reservoir training task which is essentially a ridge regression task. The MIMO precoding task solves $\hat{\mathbf{x}} = (\mathbf{H}^H\mathbf{H} + \lambda\mathbf{I})^{-1}\mathbf{H}^H\mathbf{y}$, while the reservoir training

task solves $\mathbf{W}_{out} = \mathbf{Y}_d\mathbb{O}_{total}^T(\mathbb{O}_{total}\mathbb{O}_{total}^T + \lambda\mathbf{I})^{-1}$. The matrices to be inverted in these applications are typically diagonally dominant, and their feasibility of being inverted on a PIP is verified in the experiments. To predict the advantages of using a PIP for matrix inversions in these two applications, we offload the matrix inversion task to a hypothetical PIP based on the previous analyses of the PIP's processing time, accuracy, and energy consumption, and implement the main model on electronic processors. The estimated IO advantages of these two tasks are summarised in Table 3. More details about these two tasks can be found in Supplementary 8.

For a normal MIMO system or the current massive MIMO system, the size of the channel matrix is typically 4 × 4 and up to 8 × 8. For the future massive MIMO system, the channel matrix will be scaled to 128 × 128[31,32] or even larger. At least an order of magnitude IO improvement of the PIP is predicted for PIP sizes larger than 8 × 8. In practice, the MIMO channel matrices are time-varying and continuous matrix inversions are required. The updating period of a MIMO channel ranges from several minutes in a quasi-static environment to several microseconds in high-speed scenarios. This means ~$10^3$–$10^{10}$ matrices need to be inverted per day, resulting in substantial amount of saved processing time and power consumption by using the PIP. For the reservoir training of the deep learning datasets, 8.5 times improvement in IO efficiency is predicted for a 10 × 10 PIP. Similarly, the total saved processing time and energy consumption becomes significant since a total of 60000 matrices needs to be inverted for the training task. The IO advantages of the PIP are more significant for datasets with more features. For both examples, wavelength

**Table 2 | Achieved IO advantages in the experiments***

| | Parameters | $T_{save}$ (µs) | $E_{save}$ (nJ) | $\frac{t_{total\_PSP}}{t_{total\_PIP}}$ | $\frac{C\_to\_IO_{PIP}}{C\_to\_IO_{PSP}}$ |
|---|---|---|---|---|---|
| $A_1^{-1}$ | $q = 0$, $N = 4$, $t_{loop} = 130$ ns, $P = 5$ | 58.3 | 1.4 | 4.3 | 4.9 |
| $A_2^{-1}$ | $q = 0$, $N = 4$, $t_{loop} = 130$ ns, $P = 8$ | 98.3 | 2.4 | 6.1 | 7.6 |
| IE | $q = 1$, $N = 4$, $t_{loop} = 130$ ns, $P_1 = 6$, $P_2 = 6$ | 156.7 | 3.8 | 1.7 | 1.75 |
| ODE | $q = 1$, $N = 4$, $t_{loop} = 130$ ns, $P_1 = 5$, $P_2 = 8$ | 170.0 | 4.1 | 1.8 | 1.82 |
| PDE | $q = 2$, $N = 4$, $t_{loop} = 130$ ns, $P_1 = P_3 = 9$, $P_2 = P_4 = 8$ | 460.0 | 11.0 | 1.2 | 1.2 |
| $A_3^{-1}$ | $q = 0$, $N = 2$, $t_{loop} = 300$ ps, $P = 3$ | 9.0 | 0.22 | 2.8 | 2.8 |
| $A_4^{-1}$ | $q = 0$, $N = 2$, $t_{loop} = 300$ ps, $P = 2$ | 5.0 | 0.12 | 2.0 | 2.0 |
| $A_5^{-1}$ | $q = 0$, $N = 2$, $t_{loop} = 300$ ps, $P = 2$ | 5.0 | 0.12 | 2.0 | 2.0 |

*A resolution of 10 bit is used for experiments involving real-valued matrix inversions and a resolution of 12 bit is used for experiments involving complex-valued matrix inversions, which is mainly limited by the resolution of the oscilloscope used. A data transmission rate of 24 Mb/s is employed for the Serial Peripheral Interface (SPI communication protocol. $q$ is the number of times a matrix is decomposed before being inverted on an $N \times N$ PIP. $N$ is the matrix size. $P$ is the number of iterations for the Richardson method to converge.

multiplexing techniques can be used to enhance the IO advantages, as indicated by the numbers in the brackets in Table 3, leading to up to two orders of magnitude improvement in IO efficiency for MIMO precoding tasks and at least an order of magnitude improvement for the MNIST training task.

**Photonic integration techniques for a fully integrated PIP**
According to Table 2, even if a PIP is not fully integrated on a single chip and with a limited scale, we can still gain notable benefits in speed and power consumption thanks to the reduced IO access counts. Moreover, as shown in Table 3, a fully integrated PIP as shown in Fig. 1 or a large-scale PIP makes the IO advantage more significant. Having optical gain in the loop supports a significantly increased number of iterations, making the PIP a more appealing technology in breaking the IO bottlenecks in practical matrix-inversion-intensive tasks. However, the integration of a light source or an amplifier on silicon chip still remains an active research area today, as silicon is an indirect bandgap material that precludes the possibility of a silicon laser. A number of hybrid and heterogeneous integration methods have been investigated to combine the power of III-V with the full capability of silicon-on-insulator (SOI). These include flip-chip bonding[33], die/wafer bonding[34,35], micro-transfer printing[36], direct epitaxial growth[37,38], optical wire bonding[39,40], and membrane technology[41,42]. There are an increasing number of in-house demonstrations in the full integration of active and passive components, and when they are released to the public in the future, a fully integrated PIP can be realised immediately.

In summary, we propose a PIP with reconfigurable photonic circuits that is capable of handling signals in the optical domain recursively, achieving much higher computation-to-input/output ratio than traditional single-pass optical processors and digital electronic processors. This heralds a different photonic computing paradigm that significantly reduces the data shuttling cost. We showcase a lossless PIP and a coherent PIP with on-chip optical loops to demonstrate its power. High-fidelity optical computations including real-valued matrix inversions, real-valued integral and differential equation solving, and complex-valued matrix inversions are performed. By emulating the MIMO precoding tasks and the reservoir training tasks, the proposed PIP is shown to be capable of reaching at least an order of magnitude IO efficiency enhancement compared with a single-pass optical processor, benefiting from the much-reduced IO demand. Our work paves the way towards the next generation photonic processor that significantly enhances IO efficiency and processing speed.

## Methods
### Chip design and fabrication of the lossless PIP
The photonic chip with a footprint of 2.8 × 6.6 mm² is fabricated on a SiN platform provided by CORNERSTONE multi-project wafer run using the standard deep ultraviolet lithography with a feature size of 250 nm. The platform comprises a 3 µm buried oxide layer, and a 2 µm silicon dioxide top cladding, and a 300 nm thick low pressure chemical vapour deposition (LPCVD) SiN layer, which provides propagation loss of <1 dB/cm. Basic building blocks including the strip waveguide, the 1 × 2 MMI coupler, the 90° bend are standard components provided by CORNERSTONE. The waveguide crossing and the edge coupler are customised using the ANSYS Lumerical FDTD simulator. The waveguide crossing has an extinction ratio of >30 dB (simulation). The edge coupler is based on a reverse taper structure, with a mode diameter of around 3.5 µm and a coupling loss of ~2.5 dB per facet (simulation).

### Chip design and fabrication of the coherent PIP
The photonic chip with a footprint of 4.4 × 9.8 mm² is fabricated on a SiN platform provided by CORNERSTONE multi-project wafer run using the same technology as that of the lossless PIP. Most component designs are same as those in the lossless PIP, including the strip waveguide, the 1 × 2 MMI coupler, the 90° bend, the waveguide crossing, and the edge coupler. In addition, a 90:10 directional coupler is integrated right before the input adder to serve as an additional monitor port for the computing results, contributing to a 10% extra loop loss. The directional coupler is customised using the ANSYS Lumerical FDTD simulator.

### Electrical control system
A Matlab programme is used to control a microcontroller unit (MCU, Arduino Mega 2560), which configures the output voltage of a digital-to-analogue conterter (DAC, AD5370) using the serial peripheral interface (SPI) protocol according to the pre-characterised lookup tables. The output voltages from the DAC are then amplified by a customised 40-channel driver circuit and then applied to the on-chip TOPSs to encode the matrix weight and configure other components. The real-time output signals recorded in the oscilloscope are sent back to the computer using the virtual instrument software architecture (VISA) standard.

### Chip characterisation of the lossless PIP
16 MZI weights are characterised independently by launching a laser into the chip and measuring the output light intensity while sweeping the applied voltage to the heaters on the MZI arms. 16 transmission-voltage (T-V) or transmission-power (T-P) curves are recorded and fitted to form lookup tables for loading matrix weights. The effective weight of each matrix element is a combination of the attenuation in the weighting bank, the loss in the loop including the MZI unit, and the gain in that loop. This is achieved by creating an optical loop between input and output waveguides for a single MZI. The effective weight is then determined by launching an optical pulse into the loop

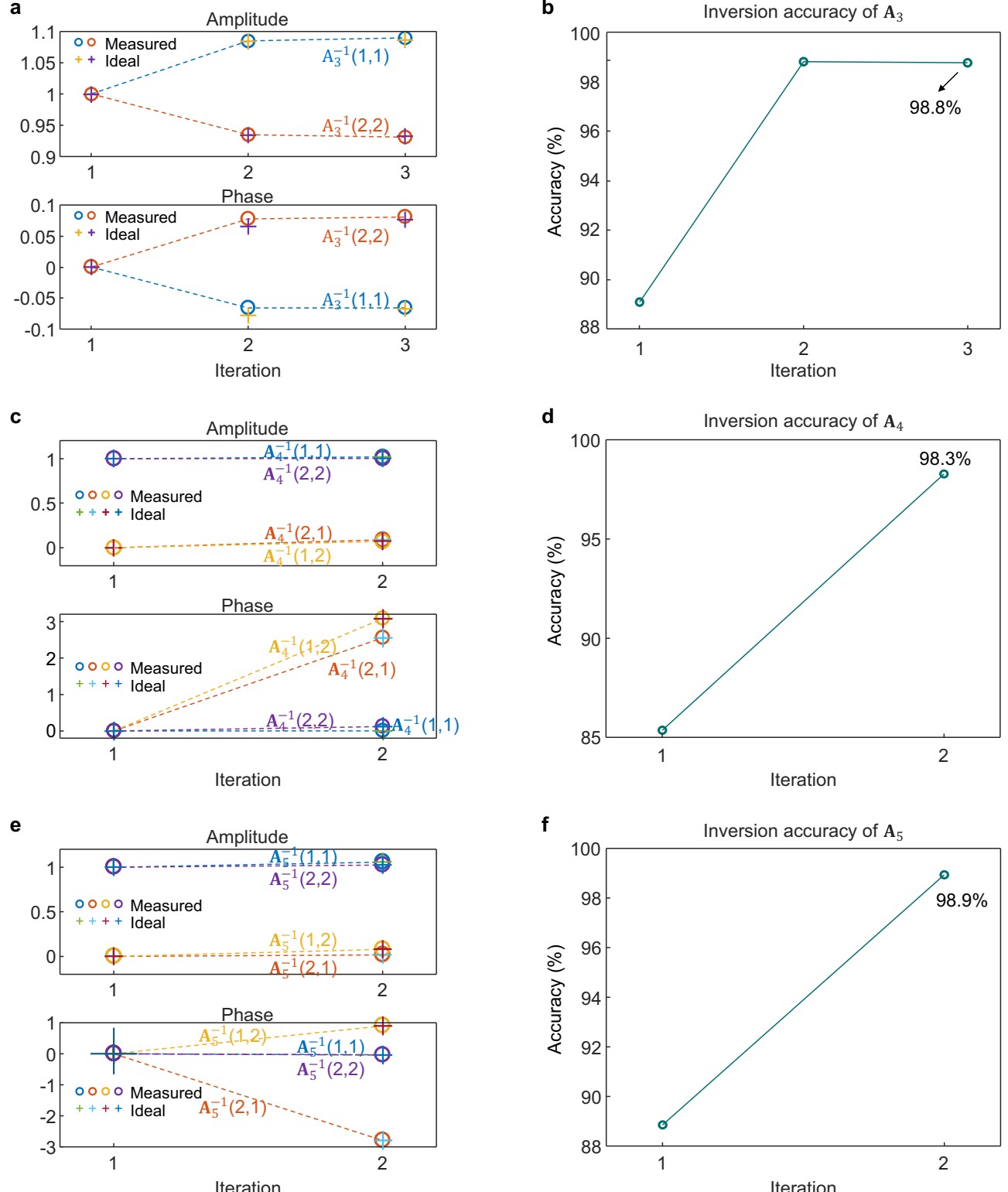

**Fig. 6 | Complex-valued matrix inversion examples. a** Ideal and measured inversion process of two diagonal elements of $A_3$. **b** Evolution of inversion accuracy of $A_3$ during convergence. **c** Ideal and measured inversion process of $A_4$.

**d** Evolution of inversion accuracy of $A_4$ during convergence. **e** Ideal and measured inversion process of $A_5$. **f** Evolution of inversion accuracy of $A_5$ during convergence.

and measuring the ratio of subsequent output pulses. Note that there is an EDFA within the loop whose purpose is to compensate for optical losses in the loop. Finally, the attenuation in the MZI unit is adjusted to provide the required attenuation for each matrix element.

## Chip characterisation of the coherent PIP

In order to characterise the 4 MZI weights, two loop switches, one input switch, and two coherent detectors need to be characterised first. The "On/Off (Bar/Cross)" states of two loop switches are characterised by measuring the transmission of a light signal through two edge-coupled

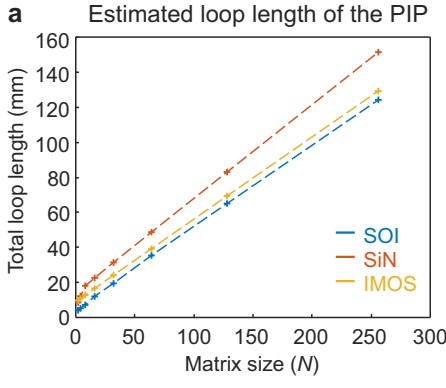

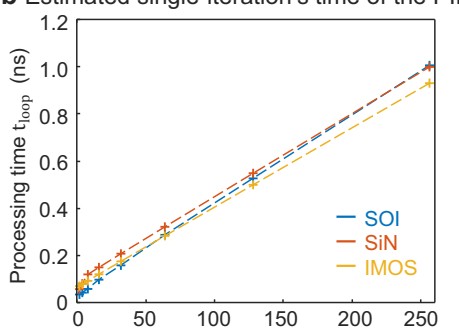

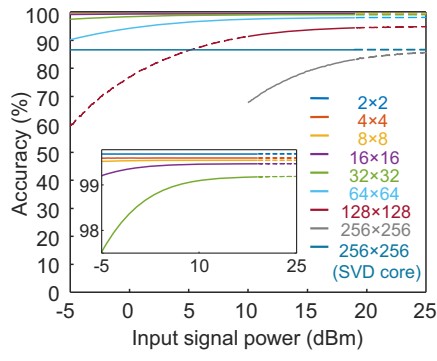

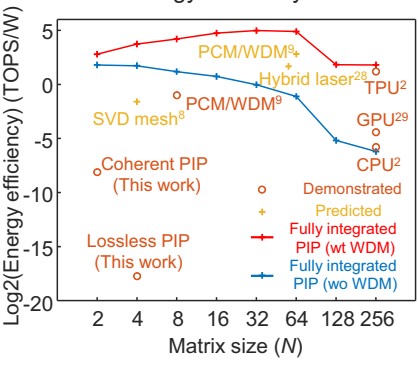

**Fig. 7 | Performance analyses of the PIP. a** Estimated loop length of the PIP with sizes ranging from $2 \times 2$–$256 \times 256$. Length of each component is based on the reported best values. **b** Estimated single-iteration's processing time with sizes ranging from $2 \times 2$–$256 \times 256$. **c** Estimated inversion accuracies of the PIP with sizes ranging from $2 \times 2$–$256 \times 256$. Note the $256 \times 256$ PIP with an SVD core only consider the 6 dB splitting and adding loss. **d** Estimated energy efficiency of the PIP core with sizes ranging from $2 \times 2$ to $256 \times 256$. References for state-of-the-art electronic and photonic processors: CPU and TPU[2], SVD mesh[8], PCM/WDM[9], Hybrid laser[28], GPU[43].

### Table 3 | Predicted IO advantages of two practical tasks[a]

| | Parameters | $T_{save}$ | $E_{save}$ | $\frac{t_{total\_PSP}}{t_{total\_PIP}}$ |
|---|---|---|---|---|
| MIMO | $q = 0$, $N = 4$, $t_{loop} = 83.8$ ps, $P = 54$, $N_{samp} = 10^3 - 10^6$ per day | 0.15 ms–25 min per day | 27.3 μJ–273 J per day | 8.1 (43.9)[b] |
| MIMO | $q = 0$, $N = 8$, $t_{loop} = 124.7$ ps, $P = 104$, $N_{samp} = 10^3 - 10^6$ per day | 1.18 ms–3.3 h per day | 211.8 μJ–2.1 kJ per day | 11.2 (90.9) |
| massive MIMO | $q = 0$, $N = 128$, $t_{loop} = 616.8$ ps, $P = 183$, $N_{samp} = 10^3 - 10^6$ per day | 0.53 s–1472 h per day | 95.7 mJ–957 kJ per day | 31.6 (178) |
| massive MIMO | $q = 0$, $N = 256$, $t_{loop} = 1.1$ ns, $P = 186$, $N_{samp} = 10^3 - 10^6$ per day | 2.16 s–6000 h per day | 0.39 J–3.9 MJ per day | 33.9 (181.6) |
| MNIST training | $q = 0$, $N = 10$, $t_{loop} = 94.3$ ps, $P = 14$, $N_{samp} = 6000$, $N_{epoch} = 10$ | 14.4 ms | 2.6 mJ | 8.5 (13.2) |

[a]$q$ is the number of times a matrix is decomposed before being inverted on a $N \times N$ PIP. $N$ is the matrix size. $P$ is the number of iterations for the Richardson method to converge.
[b]Numbers in the brackets represent the cases when wavelength multiplexing techniques are used.

ports of the switch while sweeping the applied voltage to the heaters on the MZI arms. Configuring the loop switches to "On/Off (Bar/Cross)" state corresponds to maintaining/terminating the iterative process. The "Bar/Cross" state of the input switch, which corresponds to injecting unit vectors to two input ports, is characterised with both loop switches configured to "Off" states. Then the two coherent detectors, each comprising an MZI, are characterized respectively by setting the input switch to either "Bar" or "Cross" state and measuring the photo-detected electrical signals with both loop switches in "Off" states. The effective attenuation and phase shift of each MZI weight unit is the total attenuation and phase shift within the loop respectively, which are determined by launching optical pulses into two input ports respectively and measuring the ratio of the first two pulses.

### Experimental setup of the lossless PIP
The light source, Thorlabs TLX1 tunable laser, is set to a 1550 nm wavelength and 5 dBm output power. Manual polarisation controllers,

Thorlabs FPC030, are used to align light polarisations to the chip. The input modulator (also serves the function of an optical switch) is Thorlabs LNA6213 intensity modulator, which is modulated by a Tektronix AFG3102C arbitrary function generator. A 50 ns pulse is used in the computations. Amplifiers are EDFAs from Connect Laser which can provide >35 dB gain. BPFs are filters from WL Photonics with 0.1 nm 3 dB bandwidth. Outputs are recorded in a 4-channel Keysight DSO-S 404 A oscilloscope.

### Experimental setup of the coherent PIP
The light source, polarisation controllers, EDFAs and BPFs are same as the lossless PIP. The input modulator is Thorlabs LNA6213 intensity modulator, which is modulated by a Tektronix AWG5204 arbitrary waveform generator after amplified by an SHF S807C amplifier. Pre-emphasis of the input pulse is required to ensure a flat pulse shape. A 5 ns pulse is used in the computations. The BPD from Finisar has a bandwidth of 0–43 GHz. Optical delay lines from Newport with a delay

range of 0–330 ps are inserted before two input ports of the BPD to match the arrival time of two signals. Outputs are recorded in a 4-channel Teledyne Lecroy WavePro 804HD oscilloscope.

## Numerical methods

Discretization is needed for numerically solving equations (see Supplementary 2). The rectangular integration technique is used for solving integral equations by approximating the integral by summing a series of rectangular partitions under the curve. Fredholm integral equations of the second kind can be written as

$$\mathrm{f}(t_i) = \mathrm{c}(t_i) + \frac{b-a}{N} \cdot \sum_{j=1}^{N} \mathrm{K}(t_i, s_j)\mathrm{f}(s_j) \tag{5}$$

$[a,b] = [-1,1]$ is the integral interval. $N = 8$ is the number of equally divided subintervals of $[a,b]$. Assume the divisions are the same along both $t$ and $s$ axes. $t_i$ and $s_j$ $(i,j = 1, 2, \ldots, N)$ are midpoints of the subintervals along both axes (called the discretized points). $\mathrm{K}(t_i, s_j)$ is the value of kernel function at the discretized point $(t_i, s_j)$. $\mathrm{f}(t_i)$ is the value of the function to be solved at discretized points $t_i$. $\mathrm{c}(t_i)$ is the value of the input function at discretized points $t_i$. Equation (5) can be expressed in a matrix form as $\mathbf{f} = \mathbf{c} + \frac{b-a}{N}\mathbf{K}\mathbf{f}$. The solution of the linear equation is $\mathbf{f} = (\mathbf{I} - \frac{b-a}{N}\mathbf{K})^{-1}\mathbf{c}$.

The finite difference method is used to solve differential equations by using finite difference formulas at evenly spaced grid points to approximate the differential equations. There are three types of difference formulas, which are central, forward and backward differences. Here we use central difference to approximate the equations. The first and second order derivatives of ODEs (1D system) can be written as

$$\frac{d\mathrm{y}}{dx} = \frac{\mathrm{y}_{i+1} - \mathrm{y}_{i-1}}{2h} \tag{6}$$

$$\frac{d^2\mathrm{y}}{dx^2} = \frac{\mathrm{y}_{i+1} - 2\mathrm{y}_i + \mathrm{y}_{i-1}}{h^2} \tag{7}$$

where $i$ is the index of the desired grid point, $i-1$ and $i+1$ are the indices of the neighbouring points, and $h$ is the grid size. A fixed grid size is used for simplicity. For PDEs (2D system), the second order partial derivatives with respect to variable $x$ and the gradient relationship can be written as

$$\frac{\partial^2 \mathrm{u}(x,y)}{\partial x^2} = \frac{\mathrm{u}_{i+1,j} - 2\mathrm{u}_{i,j} + \mathrm{u}_{i-1,j}}{h^2} \tag{8}$$

$$\Delta\mathrm{u}(x,y) = \frac{\partial^2\mathrm{u}(x,y)}{\partial x^2} + \frac{\partial^2\mathrm{u}(x,y)}{\partial y^2} = \frac{\mathrm{u}_{i+1,j} + \mathrm{u}_{i,j+1} - 4\mathrm{u}_{i,j} + \mathrm{u}_{i-1,j} + \mathrm{u}_{i,j-1}}{h^2} \tag{9}$$

The discretized Eqs. (5–9) are then mapped into coefficient matrices which describes the relation between a point and other points in the grid. Solving differential equations are then converted to solving matrix inversions.

## Data availability

All data about the device design can be found in the article and its Supplementary Information. The data that support the plots within this article are available in the University of Cambridge Repository at https://doi.org/10.17863/CAM.109538.

## Code availability

The code used in this study is available from the corresponding author upon request.

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

## Acknowledgements

This work was supported by the European Union's Horizon 2020 research and innovation programme, project INSPIRE, UK EPSRC, project QUDOS (EP/T028475/1), and GlitterinTech Limited, Xuzhou, China. The authors thank CORNERSTONE for providing free access to their second SiN MPW run (funded by the CORNERSTONE 2 project under Grant EP/T019697/1). The authors also thank Prof.. Hui Zhang, Prof. Kan Wu, Dr. Mark Holm, and Mr. Zexing Li for helpful discussions.

## Author contributions

M.C. and Q.C. conceived the idea. M.C. performed numerical simulations, designed the schematics of the photonic chips, built the experimental set-ups, conducted the experiments with helpful advice from A.W., S.Y. and Q.C., and analysed the results. Y.W. assisted with the coherent PIP test and built the electronic models for ridge regression and MNIST training. C.Y. and M.C. designed the layouts of the photonic chips. M.C. and Q.C. wrote the manuscript with inputs from all authors. Q.C. and R.P. supervised the project.

## Competing interests

The authors declare no competing interests.
