## [Peer Review File · Nature Communications]

I/O-efficient iterative matrix inversion with photonic integrated circuitsREVIEWER COMMENTS

Reviewer #1 (Remarks to the Author):

In this manuscript, the authors present a photonic iterative processor (PIP) for matrix inversion. They demonstrate the capabilities of PIP through experiment and simulation of the real-valued matrix inversions, real-valued integral and differential equation solving, and complex-valued matrix inversions. The authors assert that PIP exhibits a reaching over an order of magnitude processing speed enhancement.

However, after carefully evaluating the manuscript, I find that the author's PIP design structure does not demonstrate the advantage and efficiency in the computational complexity of the matrix inversion algorithm. Additionally, the authors have not demonstrated the complete PIP structure in on-chip experiments, instead most of the experiments are done with off-chip devices or computer simulations. Consequently, I remain unconvinced that this manuscript justifies publication in such a high-impact journal. I outline my concerns in detail below:

1. After a thorough reading of the manuscript, I didn't find the PIP structure to be advantageous in terms of computational complexity for the matrix inversion problem. Despite the author's assertion of $P=O(N^0.42)$ computational complexity in their PIP, accompanied by the inclusion of Figure S1 in Supplementary 1.3, I am inclined to withhold my agreement due to the absence of any substantiated theoretical proof. The solitary reliance on result S1, derived from simulation rather than experimentation, falls short of substantiating the author's chip design claims.

From my perspective, whether employing a digital computer or a photonic iterative processor, for the same matrix inversion iterative computation algorithm (e.g., the Richardson method), both necessitate essentially identical numbers of iterations, thereby sharing equivalent computational complexities. Furthermore, when considering the substantial number of I/O devices required for the larger dimensions of a photonic chip, it becomes apparent that PIPs lack I/O efficiency when juxtaposed with traditional digital computers.

2. The author claims that "we report a novel PIP based on reconfigurable photonic integrated circuits that goes beyond linear matrix multiplication and addition.". However, I don't think this description matches the actual experiments done by the author. The focus of the author's investigations revolves around two specific experiments: 4×4 real-valued matrix inversions and 2×2 complex-valued matrix inversions. In the real-valued task, the implementation of iterative computation requires the optical signals to be exported off-chip and pass through active devices like the EDFA and passive devices such as BPF and beamsplitters. The resultant time consumption, notably 130 ns as provided by the author, diminishes the speed advantage purportedly offered by on-chip optical computation. Notably, the discussion and analysis of energy consumption is missed. The inclusion of an additional EDFA to support energy, especially when dealing with larger dimensions (N), results in significantly greater energy consumption than that of a digital computer. Therefore, I don't think this task is an efficient I/O approach.

In the complex-valued part, although the authors have made the loop part on the chip, there are still two problems: The fact that only a 2×2 complex matrix inversion cannot be used to justify the advantages of the algorithm provided by the authors, since a 2×2 matrix is too simple to be implemented with an MZI, and its inverse matrix can be easily solved by other methods without the need for an iterative approach. In addition, authors still used EDFA to provide energy for each iteration, the same scalability problem occurs as with the first task.

3. Figure 1, portraying the conceptual framework of the photonic iterative processor (PIP), tends to embellish the scope of the author's work, creating a potentially misleading impression upon initial reading. Notably, several on-chip structures depicted in Figure 1, such as CW lasers, modulators, switches, filters, detectors, and amplifiers, are, in reality, not fabricated on-chip. Given the substantial divergence in technical intricacy and design complexity between on-chip and off-chip devices, I am inclined to believe that Figure 1 amplifies the significance of the author's contributions beyond what is justified.

4. The author claims that "the PIP is estimated for ridge regression task in both house price prediction and MNIST training". However, the manuscript lacks a detailed description of how MNIST classification is accomplished using the PIP, either in the main text or supplementary materials. Additionally, the authors' claim of a " 10×10 PIP" for MNIST training seems to originate from the simulation of the digital model rather than an on-chip PIP, rendering any asserted

advantages void. Consequently, the authors cannot substantiate the MNIST classification as a pivotal outcome of their work.

5. In the introduction section, the author extensively delves into a review of advancements in the electronic computing realm, a narrative that appears unrelated to the central focus of this paper. Notably, the core problem of iterative matrix inversion lacks sufficient description, diverting attention from the principal theme of the manuscript.

6. Regarding the second experiment involving integral and differential equations, the authors have not elucidated their connection to the overarching theme of "matrix inversion" as indicated in the title. I am of the opinion that this content may be deemed extraneous to the manuscript and may not warrant inclusion in the main body of the text.

7. Figure 5(b) lacks detailed elucidation from the author, and the relationship between the measured and ideal values in the figure is not clearly discernible. Additional clarification and description are necessary to facilitate a comprehensive understanding of the content presented in Figure 5(b).

8. In the description of Fig. 6c and Fig. 6d, I don't understand the significance of the author giving a comparison under the three platforms. The figure shows that SiN requires the longest length, is it representing SiN as the worst choice.

9. In the discussion section, the authors delve into various hybrid and heterogeneous integration methods. However, the relevance of these investigations to the specific research presented in the manuscript is unclear, and it seems that most of these techniques are not actually employed by the authors. The section can at most appear in the introduction parts.

Reviewer #2 (Remarks to the Author):

The authors combine the advantage of executing operations in a fully-optical domain and iterative matrix inversion to empower the computation-to-IO ratio.

While the paper and figures are well structured, I would suggest clarify the following parts:

1. The matrix inversion case could be better explained, at the moment is not clear what are the usage possibilities, despite the large list of applications mentioned. I would better treat the case as a tough-to-be-solved with current SOTA electronics and why. This is not clear, which does not emphasize the importance to move into the optical domain as a first.

2. The computational complexity of $O(N^2)$ – as it is stated, does not stand out, while it would be better to connect to improvements in power consumption too.

3. The final comparison – as indicated in the intro section, but also in Fig. 6 – does not include the SOTA in electronics and comparison with it. There is a comparison for the use-cases treated for the real-valued matrix inversion, but it is not enough to convince this is the way to go.

4. It is not clear to me how to perform the subtraction. Could you clarify, and indicate the operation in figure 1?

5. The PIP generates matrix computation results one column at a time: when coming with the latency calculation, you should differentiate between this scheme and the multiplexing scheme in [39].

6. The authors mention that the filters are used to clean the signals, however the in-band noise is never removed, even with filters and could build up in the iteration process. As this is never mentioned, also in the final calculations, I would suggest to comment on this issue and how this can be solved. And of course when it is an issue, what is the expected degradation in performance.

7. For both the real-valued and the complex-valued matrix inversion, where and how are the engines trained?

8. Complex-valued matrix inversion: could you please give real specific examples where this finds application? The same section is not well explored as the previous on the real-valued operations, so that an extension of the same would be needed to highlight the improvements in real applications.

9. While a comparison among the available photonic platforms is shown in terms of scalability, it is still not clear what actually the target scaling is for each application or cluster of applications in terms of number of I/Os and PIP size and why. I believe the authors should comment on that.

Minor points:

- Page 2: matric -> matrix

- "Encoding one element": is really what the authors mean, or they intend "encoding a column" or a vector?

Dear reviewers,

We appreciate your precious time and thank you for the very detailed and constructive comments. Based on these comments, we have made significant revisions to the main text and the supplementary materials to address your concerns. In particular, we have added sufficient theoretical details, experimental results, simulation results and analyses to substantiate the claims we have made in the paper. In the following, please find our point-by-point replies to the comments. Note that we use **blue texts** for our responses to your comments, **red texts** when quoting texts from the manuscript, and **highlighted red texts** to identify new or modified texts in the manuscript. We also attach the revised manuscript with **highlighted red texts** that identify the changes. We hope we have addressed your questions and concerns.

Reviewer #1 (Remarks to the Author)

In this manuscript, the authors present a photonic iterative processor (PIP) for matrix inversion. They demonstrate the capabilities of PIP through experiment and simulation of the real-valued matrix inversions, real-valued integral and differential equation solving, and complex-valued matrix inversions. The authors assert that PIP exhibits a reaching over an order of magnitude processing speed enhancement.

However, after carefully evaluating the manuscript, I find that the author's PIP design structure does not demonstrate the advantage and efficiency in the computational complexity of the matrix inversion algorithm. Additionally, the authors have not demonstrated the complete PIP structure in on-chip experiments, instead most of the experiments are done with off-chip devices or computer simulations. Consequently, I remain unconvinced that this manuscript justifies publication in such a high-impact journal. I outline my concerns in detail below:

Our response: Thanks for your valuable comments. We apologize for not introducing the motivations of our work properly and not presenting our results with sufficient details, which might cause misunderstandings among readers. We want to point out that the key advantage of our photonic iterative processor (PIP) is its capability of implementing significantly more computations under certain input/output data size than the photonic single-pass processor (PSP) and digital electronic processors, which enhances the computation-to-IO (C-to-IO) ratio substantially, one of the key barriers for optics to gain real benefits over electronics for computing¹. The differences between the concepts of "C-to-IO ratio" and "computational complexity" and the reasons we use "C-to-IO ratio" instead of "computational complexity" in this paper are explained in detail in our response to your comment 1. The photonic iterative processing architecture is first demonstrated with the help of off-chip devices, and the advantage of our PIP is further demonstrated by integrating optical loops on-chip. While our chips have not yet integrated all components due to the use of a photonic integration platform aimed at saving cost (all chips in this work are fabricated in standard MPW runs) and reducing turnaround time, we believe that the scope of our paper is to show the potential of the PIP as a special-purpose accelerator. Additionally, we aim to point out a pathway for future development if more advanced and currently feasible integration technologies are utilized. Simulations are included to substantiate our results and claims and showcase the future scaling of the PIP.

Below, we address your specific concerns in detail and make significant revisions to the manuscript to highlight its contributions and relevance to the high standards of *Nature Communications*. We hope

that these changes have addressed your concerns and substantially improved the quality of our manuscript, thus justifying its consideration for publication in this journal.

1. After a thorough reading of the manuscript, I didn't find the PIP structure to be advantageous in terms of computational complexity for the matrix inversion problem. Despite the author's assertion of $P=O(N^{0.42})$ computational complexity in their PIP, accompanied by the inclusion of Figure S1 in Supplementary 1.3, I am inclined to withhold my agreement due to the absence of any substantiated theoretical proof. The solitary reliance on result S1, derived from simulation rather than experimentation, falls short of substantiating the author's chip design claims.

From my perspective, whether employing a digital computer or a photonic iterative processor, for the same matrix inversion iterative computation algorithm (e.g., the Richardson method), both necessitate essentially identical numbers of iterations, thereby sharing equivalent computational complexities. Furthermore, when considering the substantial number of I/O devices required for the larger dimensions of a photonic chip, it becomes apparent that PIPs lack I/O efficiency when juxtaposed with traditional digital computers.

Our response: Thanks for your valuable comments. We agree with you that we should have presented the concept of computation-to-IO (C-to-IO) ratio with more theoretical details and made more comprehensive comparisons among digital electronic computers, photonic single-pass processors (PSPs) and photonic iterative processors (PIPs) to illustrate the advantages of our PIP structure. In the following, (1) we first distinguish between two concepts: “computational complexity” and “computation-to-I/O (C-to-IO) ratio”. We also explain why we use the “C-to-IO ratio” concept instead of “computational complexity” in this paper. (2) We then compare C-to-IO ratios of digital electronic computers, PSPs, and PIPs. We explain what we mean by “I/O efficient”. (3) Next, we choose several main application scenarios to illustrate the C-to-IO ratio improvement of our PIP in practice. (4) Finally, we state what revisions we have made to improve the manuscript.

(1) The concept of “computational complexity” stems from the field of computer science, which means the amount of resources (computation time and memory storage) required to run an algorithm² for a certain input data size. Specifically, “time complexity” is defined as the maximum number of steps (or elementary operations) required for running an algorithm on an input data size of n , and “space complexity” is defined as the maximum number of tape cells in a Turing machine (or memory space) on an input data size of n . Note the definition of “time complexity” implies there is a reference clock cycle when comparing different algorithms. “Computational complexity” focuses more on how the amount of required resources scales when the input data size increases.

The concept of “ratio of computation-to-input/output data” is first proposed in¹ and denoted as “C-to-IO ratio” in our paper, which means the number of operations implemented by a processor for a certain input and output data size. Though the definition appears to be similar to “time complexity” at a first glance, “C-to-IO ratio” does not take the clock cycle into account while simply calculates the implemented operation numbers under a certain input and output data size. “C-to-IO ratio” focuses more on how a special-purpose processor can be designed to speed up a certain algorithm when the I/O traffic bottleneck exists.

We use “C-to-IO ratio” instead of “computational complexity” in this paper for the following two reasons. (i) The matrix inversion problem we investigated in this paper is a compute-bound computation task that multiple operations are performed on each input data³. Such a task can be

accelerated by reusing the inputted data inside the processing core since the IO traffic bottleneck remains a big issue for speeding up both photonic processors and digital computers^{1,4,5}. The use of “C-to-IO ratio” is a better way to highlight the advantage of our PIP, which is the capability of implementing more operations under a given input data size by reusing the outputs via optical loopback. To put it another way, when implementing matrix inversions and related applications, using the PIP structure enables fewer memory access counts (or less IO traffic) than PSPs or digital computers, thus allowing a higher computation throughput.

(ii) There are some reported photonic processors mentioning they have improved the “computational complexity”, but we feel the concept of “computational complexity” is not well defined and probably inappropriate for benchmarking photonic processors. For “space complexity”, ref 6⁶ reports an MMI-based convolution processing unit with much fewer photonic components, but the input and output data still need to be stored in electronic memory. For “time complexity”, it is not clear if the implied clock cycles should be taken into account. Ref 7⁷ does not consider the clock cycles and reduces the number of operations of convolution by computing in the Fourier domain, Ref 8⁸ takes the light propagation time from the input to the output as one clock period and considers the number of operations implemented in a clock period, which shows an improved time complexity owing to the inherent high parallelism of optical signals. Due to the ambiguity in the definition of “computational complexity” for photonic processors, we do not use this term in this paper and discuss the related scalability issues separately.

(2) We compare the C-to-IO ratios of digital computers, PSPs and PIPs by evaluating the task of inverting an $N \times N$ non-singular matrix using the iterative Richardson method: $\mathbf{X}^{(k+1)} = (\mathbf{I}_N - \omega\mathbf{A})\mathbf{X}^{(k)} + \omega\mathbf{I}_N$ ($k = 0, 1, 2, \dots$), where $\mathbf{M} = \mathbf{I}_N - \omega\mathbf{A}$ is the weight matrix, $\mathbf{X}^{(0)} = \omega\mathbf{I}_N$ is the initial input, and $\mathbf{X}^{(k)}$ is the input data in the k^{th} iteration. In each iteration, the method requires a matrix-matrix multiplication and a matrix-matrix addition. Since here a matrix-matrix addition only involves N operations and addition is less time-consuming than multiplication in digital computers, we only consider multiply-and-accumulate (MAC) as the elementary operation. Assume the iteration is converged after $(P + 1)$ iterations, which is determined by the matrix property and the required accuracy (See Supplementary 1.2 for the derived formula). We choose CPU⁹ (central processing unit, representing the traditional general-purpose serial computing scheme), TPU^{4,10} (tensor processing unit, representing the state of art matrix multiplication unit that computes parallelly and employs a systolic architecture to reuse the input data), PSP¹¹, and our proposed PIP as 4 platforms to compare the C-to-IO ratios and the results are listed in Table R1. The total IO traffic should also include the memory access for loading the weight matrix. The corresponding data flow is illustrated in Fig. R1.

TABLE R1
A COMPARISON OF C-TO-IO RATIOS FOR INVERTING AN $N \times N$ MATRIX USING THE RICHARDSON METHOD

	CPU	TPU	PSP	PIP
Input data size	$2N^2 \cdot P$	$N^2 \cdot P$	$N^2 \cdot P$	N
Output data size	$N^2 \cdot P$	$N^2 \cdot P$	$N^2 \cdot P$	N^2
Memory access counts*	$3N^2 \cdot P$	$N^2 \cdot (2P + 1)$	$N^2 \cdot (2P + 1)$	$2N^2 + N$
Number of operations [†]	$N^3 \cdot P$	$N^3 \cdot P$	$N^3 \cdot P$	$N^3 \cdot P$
C-to-IO ratio	$N/3$	$NP/(2P + 1)$	$NP/(2P + 1)$	$N^3P/(2N^2 + N)$

* This includes the number of times a processor needs to fetch input data, store output data, and load weight matrix. For CPU, the weight matrix is also fetched in a similar way as the input data and is therefore also counted in the input data size.

† Note the computations in the first iteration are neglected in the analyses for two reasons: (1) if the process converges for $P = 1$, then the matrix to be inverted is essentially an identity matrix whose inverse is itself and does not need to be computed. (2) In the first iteration, the algorithm essentially implements N MAC operations, which can be neglected compared to the N^2 MAC operations in the following iterations.

Figure R1 | Illustration of data flows for inverting an $N \times N$ matrix in (a) CPU (b) TPU (c) PSP and (d) PIP. For each set of input and output data, the CPU and PSP compute one column of the matrix-matrix multiplication result in one iteration of the Richardson method. The TPU generates the matrix-matrix multiplication result in a systolic way (see Ref 4). The PIP directly computes one column of the inverse matrix.

As shown in Table R1, our proposed PIP has the highest C-to-IO ratio among the 4 computing platforms and thus we believe our PIP structure is an I/O efficient approach for accelerating matrix inversions. A theoretical proof is provided below:

$$\text{C-to-IO (PIP)} - \text{C-to-IO (CPU)} = \frac{N^3 P}{2N^2 + N} - \frac{N}{3} = \frac{N^2(P-1)}{3(2N^2 + N)} \geq 0, \text{ for } P \geq 1.$$

$$\text{C-to-IO (PIP)} - \text{C-to-IO (TPU/PSP)} = \frac{N^3 P}{2N^2 + N} - \frac{NP}{2P+1} = \frac{N^2 P(2NP - N - 1)}{(2N^2 + N)(2P+1)} > 0, \text{ for } N \geq 2, P > 0.$$

(3) For the original Fig. S1, we randomly generate 10000 matrices for each matrix size ranging from 2×2 to 256×256 and calculate the number of iterations for convergence (P) under 1% inversion error according to Eq. S4: $P = \left\lceil \frac{\ln(1/\varepsilon)}{\ln|(\lambda_N + \lambda_1)/(\lambda_N - \lambda_1)|} \right\rceil$, where ε is a preset acceptable error level, λ_1 and λ_N are smallest and largest eigenvalues of the matrix to be inverted, and $\lceil \cdot \rceil$ is the ceiling operation. We fit a curve for the number of iterations for convergence (P) versus matrix size. Then we use P to calculate the C-to-IO ratio. We realize that it might not be sufficient to represent the matrix inversion problems encountered in practical applications. In the following, we choose two major application cases to illustrate the C-to-IO ratios of our PIP in practice, which are ridge regression and MIMO

(multiple-input and multiple-output) precoding technology. The ridge regression task is to solve $\hat{\mathbf{\beta}} = (\mathbf{X}^T \mathbf{X} + \lambda \mathbf{I})^{-1} \mathbf{X}^T \mathbf{y}$, where $\hat{\mathbf{\beta}}$ is the parameter vector to be fitted (solved), \mathbf{X} , \mathbf{y} are the independent and dependent variables respectively that encode the samples, \mathbf{I} is an identity matrix and λ is the ridge parameter that can be adjusted to improve the fitting results. The MIMO precoding using the minimum mean square error (MMSE) method is: $\hat{\mathbf{x}} = (\mathbf{H}^H \mathbf{H} + \lambda \mathbf{I})^{-1} \mathbf{H}^H \mathbf{y}$, where $\hat{\mathbf{x}}$ is the estimated transmitted signal, \mathbf{H} is the channel matrix, \mathbf{y} is the received signal, and $\lambda = \frac{N_r}{SNR}$ is ratio between the number of transmit antennas, N_r , and the signal-to-noise ratio, SNR.

Figure R2 | Convergence iteration, P , of the PIP for different application scenarios (a) Ridge regression with an accepted error of $\varepsilon = 0.2\%$. (b) Ridge regression with an accepted error of $\varepsilon = 1\%$. (c) Ridge regression with an accepted error of $\varepsilon = 5\%$. (d) MIMO precoding with an accepted error of $\varepsilon = 0.2\%$. (e) MIMO precoding with an accepted error of $\varepsilon = 1\%$. (f) MIMO precoding with an accepted error of $\varepsilon = 5\%$. In each figure, the convergence iterations, P , for reaching the accepted error level ε are simulated and fitted for different matrix sizes and λ . λ is the ridge parameter in a ridge regression task and the ratio between number of transmit antennas and SNRs in a MIMO precoding using the minimum mean square error (MMSE) method.

The convergence iterations, P , for calculating the matrix inversions in these two applications are shown in Fig. S2 for different accepted error levels of $\varepsilon = 0.2\%$, 1% , and 5% and different parameters, $\lambda = 0.111$, 0.222 , and 0.333 which corresponds to different noise levels. $\lambda = 1$ represents an extreme case where the convergence is the fastest but the “noise level” is the same as the useful “signal level” which shall not happen in practice. For the ridge regression task, the convergence iteration, P and matrix size, N follows a linear relationship and P is apparently much larger than N . For the MIMO

precoding task, P is larger than N for almost all error levels and SNRs shown in the figure. These results indicate that our PIP has a much higher (at least N times higher for an $N \times N$ inversion) C-to-IO ratio than PSP and digital computers in these two applications. A mathematical proof is provided below.

$$\text{C-to-IO (PIP)} / \text{C-to-IO (CPU)} = \frac{N^3P}{2N^2+N} \div \frac{N}{3} = \frac{3N}{2N+1} \geq N, \text{ for } P \geq N.$$

$$\text{C-to-IO (PIP)} / \text{C-to-IO (TPU/PSP)} = \frac{N^3P}{2N^2+N} \div \frac{NP}{2P+1} = \frac{N(2P+1)}{2N+1} \geq N, \text{ for } P \geq N.$$

(4) Please find the revisions we have made to the manuscript and the supplementary below.

Introduction:

The input/output (IO) data movement in a processor typically has limited bandwidth and consumes considerable energy, leading to performance bottlenecks in processing systems that receive input data and store output data through an attached host with memory¹⁻⁴. The IO issue becomes particularly severe when the problem size exceeds the capacity of the processor, a situation that is often encountered in practice. Thus, enhancing the computation-to-IO (C-to-IO) ratio becomes a key focus, i.e. increasing the number of implemented operations for a given input/output data size. In computation intensive tasks such as matrix multiplications and inversions, where multiple operations are performed on each piece of input data, the C-to-IO ratio can be improved by reusing the input data. The tensor processing unit (TPU) exemplifies this approach in accelerating neural network inference tasks. The TPU utilizes a systolic architecture to compute matrix multiplications in parallel by reusing the input data, demonstrating significant speed and energy efficiency improvements over central processing units (CPUs) and graphics processing units (GPUs)². Nevertheless, the challenge of keeping pace with the ever-growing demands of compute-intensive applications has urged people to keep seeking faster and more energy-efficient solutions.

Photonic processors have long been considered as a promising alternative in algebra computations, owing to its inherent high parallelism and low energy consumption⁴⁻⁷. Photonic processors have shone through in the field of matrix multiplications, notably been demonstrated in voice recognition⁸, image classification^{9,10}, and optical communication¹¹. However, few research has been done on using photonic processors to accelerate matrix inversion, a more computationally expensive operation that is fundamental to many scientific and engineering problems such as numerical computations¹²⁻¹⁶, statistics^{12,17}, wireless communication systems¹⁸⁻²² and neural network training^{23,24}. In most practical applications, the matrices to be inverted are diagonally dominant or sparse, and approximate inversion results are often sufficient for subsequent processing stages. Iterative inversion methods generally outperform direct methods in these scenarios since one can easily balance speed and accuracy by terminating the iterative process after several iterations, while interrupting the inversion process of direct methods midway yields incorrect results¹². Classical iterative matrix inversion algorithms essentially perform matrix multiplications and additions iteratively. Implementing such algorithms on a widely reported photonic single-pass processor (PSP)^{8,9,25,26} necessitates repetitive inputting and outputting of computation results in each iteration, thus generating heavy IO traffic.

In this paper, we report a novel photonic iterative processor (PIP) based on reconfigurable photonic integrated circuits for speeding up matrix inversion tasks. The inclusion of optical loopback enables iterative computations for direct matrix inverting with an enhanced I/O efficiency. The processor core is a matrix-vector multiplier comprising MZI units, which reduces the computation overhead by directly encoding the matrix elements on MZI arrays. Table 1 compares the C-to-IO ratios for inverting an $N \times N$ matrix using the iterative Richardson method¹² on four different platforms including the CPU, TPU, PSP, and our proposed PIP. The Richardson method is assumed to converge after $P + 1$ iterations. According to the table, the PIP exhibits the highest C-to-IO ratio of

$N^3P/(2N^2 + N)$ among the four computing platforms, followed by the PIP, the TPU, and the CPU. In most application scenarios, P is much larger than N , indicating an improvement of at least N times in the C-to-IO ratio for PIPs compared to other platforms (See Supplementary 1 for more details).

TABLE 1

A COMPARISON OF C-TO-IO RATIOS FOR INVERTING AN $N \times N$ MATRIX USING THE RICHARDSON METHOD

	CPU	TPU	PSP	PIP
Input data size	$2N^2 \cdot P$	$N^2 \cdot P$	$N^2 \cdot P$	N
Output data size	$N^2 \cdot P$	$N^2 \cdot P$	$N^2 \cdot P$	N^2
Memory access counts[‡]	$3N^2 \cdot P$	$N^2 \cdot (2P + 1)$	$N^2 \cdot (2P + 1)$	$2N^2 + N$
Number of operations[‡]	$N^3 \cdot P$	$N^3 \cdot P$	$N^3 \cdot P$	$N^3 \cdot P$
C-to-IO ratio	$N/3$	$NP/(2P + 1)$	$NP/(2P + 1)$	$N^3P/(2N^2 + N)$

* This includes the number of times a processor needs to fetch input data, store output data, and load weight matrix (the matrix to be multiplied in each iteration). For CPU, the weight matrix is also fetched in a similar way as the input data and is therefore included in the input data size.

† Note the computations in the first iteration are neglected in the analyses for two reasons: (1) if the process converges for $P = 1$, then the matrix to be inverted is essentially an identity matrix whose inverse is itself and does not need to be computed. (2) In the first iteration, the algorithm essentially implements N multiply-and-accumulate (MAC) operations, which can be neglected compared to the N^2 MAC operations in the following iterations.

We demonstrate, to the best of our knowledge, the first lossless reconfigurable PIP system that is capable of directly inverting matrices and solving integral and differential equations. Such a PIP system computes 4×4 real-valued matrix inversions with an accuracy of $>97\%$, and a net inversion time of $2.6 \mu\text{s}$, which is solely bounded by the length of fibre-based optical loops. The lossless PIP is then reconfigured to numerically solve real-valued integral and differential equations, reaching a mean absolute error of < 0.02 . The first coherent PIP with on-chip optical loops is also demonstrated to break the loop-length limitation on the net inversion time. The coherent PIP is demonstrated to operate 2×2 complex-valued matrix inversions with an accuracy of $> 98\%$, and a net inversion time of 1.2 ns . Benefiting from the much-reduced IO demand, the proposed PIP is capable of reaching over two order of magnitudes processing time enhancement compared with a single-pass optical processor, by emulating ridge regression tasks and MIMO precoding tasks. Our results indicate a promising way towards ever-powerful optical processors that could surpass IO limits.

Supplementary 1.3:

1.3 A comparison of C-to-IO ratios

The computation-to-IO (C-to-IO) ratios of digital computers, photonic single-pass processors (PSPs) and photonic iterative processors (PIPs) are compared to illustrate the advantage of the PIP architecture in alleviating the IO bottleneck and speeding up matrix inversion tasks. The C-to-IO ratios are characterized by evaluating the task of inverting an $N \times N$ non-singular matrix using the iterative Richardson method: $\mathbf{X}^{(k+1)} = (\mathbf{I}_N - \omega\mathbf{A})\mathbf{X}^{(k)} + \omega\mathbf{I}_N$ ($k = 0, 1, 2, \dots$), where $\mathbf{M} = \mathbf{I}_N - \omega\mathbf{A}$ is the weight matrix, $\mathbf{X}^{(0)} = \omega\mathbf{I}_N$ is the initial input, and $\mathbf{X}^{(k)}$ is the input data in the k^{th} iteration. In each iteration, the method requires a matrix-matrix multiplication and a matrix-matrix addition. Since here a matrix-matrix addition only involves N operations and addition is less time-consuming than multiplication in digital computers, we only consider multiply-and-accumulate (MAC) as the elementary operation. Assume the iteration is converged after $(P + 1)$ iterations, which is determined by the matrix property and the required accuracy (See Supplementary 1.2 for the derived formula). We choose CPU¹ (central processing unit, representing the traditional general-purpose serial computing scheme), TPU^{2,3} (tensor processing unit, representing the state of art matrix multiplication unit that computes parallelly and employs a systolic architecture to reuse the input data), PSP⁴, and our proposed PIP as 4

platforms to compare the C-to-IO ratios and the results are listed in Table 1 in the main text. The total IO traffic should also include the memory access for loading the weight matrix. The corresponding data flow is illustrated in Fig. S1. As shown in Table 1, the PIP has the highest C-to-IO ratio among the 4 computing platforms and is thus an I/O efficient approach for accelerating matrix inversions. A theoretical proof is provided below:

$$\text{C-to-IO (PIP)} - \text{C-to-IO (CPU)} = \frac{N^3 P}{2N^2 + N} - \frac{N}{3} = \frac{N^2(P-1)}{3(2N^2 + N)} \geq 0, \text{ for } P \geq 1.$$

$$\text{C-to-IO (PIP)} - \text{C-to-IO (TPU/PSP)} = \frac{N^3 P}{2N^2 + N} - \frac{NP}{2P+1} = \frac{N^2 P(2NP - N - 1)}{(2N^2 + N)(2P+1)} > 0, \text{ for } N \geq 2, P > 0.$$

Figure S2 | Convergence iteration, P , of the PIP for different application scenarios (a) Ridge regression with an accepted error of $\varepsilon = 0.2\%$. (b) Ridge regression with an accepted error of $\varepsilon = 1\%$. (c) Ridge regression with an accepted error of $\varepsilon = 5\%$. (d) MIMO precoding with an accepted error of $\varepsilon = 0.2\%$. (e) MIMO precoding with an accepted error of $\varepsilon = 1\%$. (f) MIMO precoding with an accepted error of $\varepsilon = 5\%$. In each figure, the convergence iterations, P , for reaching the accepted error level ε are simulated and fitted for different matrix sizes and λ . λ is the ridge parameter in a ridge regression task and the ratio between number of transmit antennas and SNRs in a MIMO precoding using the minimum mean square error (MMSE) method.

In the following, two major application cases are chosen to illustrate the C-to-IO ratio enhancement of the PIP in practice, which are ridge regression and MIMO (multiple-input and multiple-output) precoding technology. The ridge regression task is to solve $\hat{\beta} = (X^T X + \lambda I)^{-1} X^T y$, where $\hat{\beta}$ is the parameter vector to be fitted (solved), X, y are the independent and dependent variables respectively that encode the samples, I is an identity matrix and λ is the ridge parameter that can be adjusted to improve the fitting results. The MIMO precoding using the minimum mean square error (MMSE) method is: $\hat{x} = (H^H H + \lambda I)^{-1} H^H y$, where \hat{x} is the estimated

transmitted signal, \mathbf{H} is the channel matrix, \mathbf{y} is the received signal, and λ represents the additive noise during transmission.

The convergence iterations, P , for calculating the matrix inversions in these two applications are shown in Fig. S2 for different accepted error levels of $\epsilon = 0.2\%$, 1% , and 5% and different parameters, $\lambda = 0.111$, 0.222 , and 0.333 which corresponds to different noise levels. $\lambda = 1$ represents an extreme case where the convergence is the fastest but the “noise level” is the same as the useful “signal level” which shall not happen in practice. For the ridge regression task, the convergence iteration, P and matrix size, N follows a linear relationship and P is apparently much larger than N . For the MIMO precoding task, P is larger than N for almost all error levels and SNRs shown in the figure. These results indicate that our PIP has a much higher (at least N times higher for an $N \times N$ inversion) C-to-IO ratio than PSP and digital computers in these two applications. A mathematical proof is provided below.

$$\text{C-to-IO (PIP)} / \text{C-to-IO (CPU)} = \frac{N^3 P}{2N^2 + N} \div \frac{N}{3} = \frac{3NP}{2N+1} \geq N, \text{ for } P \geq N.$$

$$\text{C-to-IO (PIP)} / \text{C-to-IO (TPU/PSP)} = \frac{N^3 P}{2N^2 + N} \div \frac{NP}{2P+1} = \frac{N(2P+1)}{2N+1} \geq N, \text{ for } P \geq N.$$

2. The author claims that “we report a novel PIP based on reconfigurable photonic integrated circuits that goes beyond linear matrix multiplication and addition.”. However, I don't think this description matches the actual experiments done by the author. The focus of the author's investigations revolves around two specific experiments: 4×4 real-valued matrix inversions and 2×2 complex-valued matrix inversions.

Our response: Thank you for pointing out this presentation problem. We agree with you that the focus of our research is photonic iterative processor for accelerating matrix inversion problems. Matrix multiplications and additions are intermediate operations that are implemented during the inversion process. This sentence in the introduction does not directly states the main focus of our work. We have revised the first sentence of the last paragraph in the **Introduction** section on page 2.

In this paper, we report a novel PIP based on reconfigurable photonic integrated circuits for speeding up matrix inversion tasks.

In the real-valued task, the implementation of iterative computation requires the optical signals to be exported off-chip and pass through active devices like the EDFA and passive devices such as BPF and beamsplitters. The resultant time consumption, notably 130 ns as provided by the author, diminishes the speed advantage purportedly offered by on-chip optical computation. Notably, the discussion and analysis of energy consumption is missed. The inclusion of an additional EDFA to support energy, especially when dealing with larger dimensions (N), results in significantly greater energy consumption than that of a digital computer. Therefore, I don't think this task is an efficient I/O approach.

Our response: You are correct that the speed advantage is limited if the optical loop is partially integrated which is mainly due to the long fibre length ($\sim 35\text{m}$) inside the EDFA. In the real-valued experiments, the main focus is to validate the idea of using optical loopback to implement iterative computing using coherent light sources. The speed advantage of our PIP is further demonstrated by integrating optical loops on-chip which is shown in the complex-valued experiments.

We apologize for not including the energy consumption analyses. Please find the analyses and discussions below. Admittedly, in the current real-valued demonstration, the heaters and off-chip EDFAs consume considerable energy which is due to the integration platform we used. However, we

believe the energy efficiency of our PIP architecture in a fully integrated platform will not be a limitation to its performance in matrix inversion tasks and exhibits even better energy efficiency than the state-of-the-art electronic and photonic processors when wavelength multiplexing techniques are employed. We substantiate our claims with the following reasons.

(1) We first analyse the energy efficiency of the PIP qualitatively and compare with that of single-pass photonic processors and electronic processors. The energy efficiency analysis can be decomposed into two parts: the energy efficiency of the processing core and the IO access. We first consider the energy efficiency of the processing core as what has been done in most literatures. For photonic matrix processors, the power losses during signal splitting and combining are intrinsic, whether they are single-pass or iterative. The power loss in a PSP needs to be compensated by either boosting up the signal power at input¹², or by amplifying in the electrical domain after photodetection¹³. The same amount of energy is supplied by semiconductor optical amplifiers (SOAs) in a PIP in each iteration. Therefore, the additional energy required to compensate for signal losses in a photonic processor core ultimately remains the same for both PSPs and PIPs in matrix inversion tasks. PSPs are reported to consume less power than digital computers, which supports our assertion that the energy efficiency of PIPs will not be a limitation. As for comparing the energy efficiency of the PIP core and the electronic processors, we consider the scaling of the dominating power consumption on these two hardware platforms. The required power consumption of SOAs in a fully integrated PIP for a matrix-matrix multiplication operation scales quadratically with the processor size. In contrast, the energy required to move electrical data within the processing unit and to implement the multiply-accumulate operations for a matrix-matrix multiplication operation scales cubically with the processor size. Therefore, a properly designed photonic processor should in principle be more energy efficient than an electronic processor.

We then consider the energy consumption for IO access. As shown in Table 1, the PIP has a much higher C-to-IO ratio and a much-reduced IO access counts than electronic processors and PSPs for inverting an $N \times N$ matrix. Hence, the PIP consumes less energy consumption for data IO access.

(3) We now provide a detailed analyses of the energy efficiency of fully integrated PIPs with different sizes, which is presented in Supplementary 6.6.

Supplementary 6.6:

6.6 Energy efficiency estimation of an $N \times N$ PIP

TABLE S7

ESTIMATED ENERGY CONSUMPTION OF THE PIP CORE FOR $N \times N$ MATRIX INVERSIONS	
Component	Equation
Laser	$E_{laser} = P_{laser}/\eta \cdot N \cdot P \cdot t_{loop}$
Modulator	$E_{mod} = V_{mod}^2/R \cdot N \cdot P \cdot t_{loop}$
SOA	$E_{SOA} = (P_{out} - P_{in})/\eta \cdot N^2 \cdot P \cdot t_{loop}$
PD	$E_{PD} = \mathcal{R}P_{rec} \cdot V_{bias} \cdot N^2 \cdot t_{loop}$
Weight bank	$E_{weight} = P_{heater} \cdot N^3 \cdot P \cdot t_{loop}$
Optical switch	$E_{switch} = V_{mod}^2/R \cdot N^2 \cdot P \cdot t_{loop}$

Energy efficiency of the PIP core is defined as number of operations per second per energy consumption of the PIP core. Table S7 lists the equations to estimate the energy consumption of the PIP core for inverting an $N \times N$ matrix. P_{laser} is the output power of the laser, which should be at least -5 dBm for PIP size not exceeding 64×64 and 10 dBm for 128×128 and 256×256 PIPs according to Supplementary 6.5. η is the wall-plug efficiency of the laser and the semiconductor optical amplifier (SOA), which has a best reported value of around 30%²⁹. V_{mod} is the driving voltage of the modulator and the optical switch, which is around 250 mV³⁰ for the best reported electro-optic modulators. $R = 50\Omega$ is the load impedance for modulators and optical switches. P_{out} and P_{in} are input and output powers of the SOA and are size dependent since the loop loss increases as the matrix size increases. The estimated required gain for an $N \times N$ processor is presented in Supplementary 6.1. $\mathcal{R} \approx 0.6A \cdot W^{-1}$ is the responsivity of the photodetector. P_{rec} is the received optical power at the photodetector, which should be same as P_{laser} in the lossless PIP system. The minimal required signal power P_{laser} for reaching a certain inversion accuracy is also size dependent. The inversion accuracy vs. input signal power for different-sized processor is shown in Fig. 6e. $V_{bias} \approx 3V$ is the bias voltage of photodetector. P_{heat} is the power consumption of thermo-optic phase shifters in the weight bank, whose best reported value is $0.05mW/\pi$ phase shift. t_{loop} is the core processing time in a single iteration, which depends on the loop length and the effective index of the integration platform as shown in Fig. 6c and Supplementary 6.3. Fig. S14 illustrates the estimated energy consumption and energy efficiency of the PIP core. Fig. S14a shows the energy consumption of the PIP core for computing matrix inversions in the ridge regression task and the MIMO task respectively. The minimal and maximal energy consumptions are illustrated respectively based on the minimal and maximal inversion time shown in Table S4.

Figure. S14b shows the estimated energy efficiency of the PIP core (with and without multiplexing techniques) in terms of tera-operations per second per watt, together with the energy efficiency of the demonstrated PIP systems in this work and several state-of-the-art electronic and photonic processors to showcase the advancement in energy efficiency of our proposed PIP. Though the demonstrated coherent PIP and lossless PIP is less energy efficient than electronic processors when only the processing core is considered mainly due to the limited integration level, the predicted energy efficiency of a fully integrated lossless PIP outperforms state-of-the-art electronic and photonic processors. Considering the much-reduced IO access counts of our PIP. It is safe to say the proposed PIP is more energy efficient in matrix inversion tasks compared to traditional electronic processors and photonic processors.

Figure S14 | Estimated energy efficiency of the lossless PIP core for inverting an $N \times N$ matrix. (a) Energy consumption of the PIP core. (b) Energy efficiency of the PIP core.

In the complex-valued part, although the authors have made the loop part on the chip, there are still two problems: The fact that only a 2×2 complex matrix inversion cannot be used to justify the advantages of the algorithm provided by the authors, since a 2×2 matrix is too simple to be implemented with an MZI, and its inverse matrix can be easily solved by other methods without the need for an iterative approach. In addition, authors still used EDFA to provide energy for each iteration, the same scalability problem occurs as with the first task.

Our response: You are correct that 2×2 matrix inversion can be solved according to the formula:

$$\begin{bmatrix} a & b \\ c & d \end{bmatrix}^{-1} = \frac{1}{ad-b} \begin{bmatrix} d & -b \\ -c & a \end{bmatrix}.$$

However, the main purpose of the 2×2 complex-valued experiments is to demonstrate the concept of integrating optical loops on-chip for further speed improvement. Given that this is the first attempt to realize an on-chip PIP for matrix inversion, the size of our coherent PIP with loops integrated on-chip is limited to 2×2 to minimize risks. The advantages of our PIP structure for matrix inversion tasks include: (1) A much higher C-to-IO ratio compared to the state-of-the-art electronic processors and photonic single-pass processors and hence tens (without multiplexing techniques) to thousands (with multiplexing techniques) of latency improvement in ridge regression and MIMO precoding tasks. (2) A slightly better (without multiplexing) or much higher (with multiplexing) energy efficiency compared to the state-of-the-art electronic processors and PSPs. The analyses and discussions are presented in Introduction and Discussion in the main text and Supplementary 1 and 6.

As we have stated in the main text, in complex-valued experiments, the EDFAs are used to compensate for the coupling losses since we have not integrated balanced photodetectors on-chip. This part of energy is included in the calculations in Fig. 6d but is not intrinsic to our PIP. The scalability of energy efficiency is analysed in detail in Supplementary 6.

3. Figure 1, portraying the conceptual framework of the photonic iterative processor (PIP), tends to embellish the scope of the author's work, creating a potentially misleading impression upon initial reading. Notably, several on-chip structures depicted in Figure 1, such as CW lasers, modulators, switches, filters, detectors, and amplifiers, are, in reality, not fabricated on-chip. Given the substantial divergence in technical intricacy and design complexity between on-chip and off-chip devices, I am inclined to believe that Figure 1 amplifies the significance of the author's contributions beyond what is justified.

Our response: We apologize for any misunderstanding our work may have caused you. Figure 1 is intended to illustrate the concept of the PIP architecture that we ultimately want to demonstrate. Constrained by currently accessible MPW foundry services, we were not able to integrate all components on-chip. In the real-valued demonstrations, the weight bank and signal splitting and coupling parts are integrated on-chip. In the complex-valued demonstrations, the weight bank, signal splitting and coupling, optical loopback paths, optical switches are integrated on-chip. In addition, we have included sufficient theoretical details and analyses based on the performances of on-chip components to substantiate the claim that our PIP is an IO-efficient and energy-efficient approach for matrix inversion tasks. We hope you will agree that Fig. 1 is instructive to readers and point out a clear pathway for the PIP processor. As we have described in “**Discussion-Photonic integration platforms for realising a fully integrated PIP**”, technologies for heterogeneously integrated lasers and amplifiers, etc are already available.

Discussion

Photonic integration platforms for realising a fully integrated PIP

In this paper, both the lossless and coherent PIP systems are formed with a SiN chip core and off-the-shelf components. To improve the latency, scalability, energy efficiency and cost-effectiveness of the PIP system, it is essential to move towards a fully integrated processor on a single chip. Such a candidate chip design is illustrated in Fig. 1. A number of hybrid and heterogeneous integration methods have been investigated to combine the power of III-V with the full capability of SOI, spanning from flip-chip bonding²³, die/wafer bonding^{24,25}, micro-transfer printing²⁶, to direct epitaxial growth^{27,28}. Epitaxial growth may represent an ultimate path but die/wafer bonding features a higher technology readiness level in a short term. Membrane technology^{29,30}, such as the InP membrane on silicon (IMOS), also represents a significant platform that co-integrates passive and active components. Also, high bandwidth modulators and photodetectors are available on the silicon-on-insulator platform³¹, III-V platform^{32,33} and the hybrid integration platform^{34,35}. The capability of reconfiguring phase shifters can at nanosecond- or even picosecond-scales can significantly reduce the weights reloading time, achieving even faster computations.

4. The author claims that “the PIP is estimated for ridge regression task in both house price prediction and MNIST training”. However, the manuscript lacks a detailed description of how MNIST classification is accomplished using the PIP, either in the main text or supplementary materials. Additionally, the authors' claim of a "10×10 PIP" for MNIST training seems to originate from the simulation of the digital model rather than an on-chip PIP, rendering any asserted advantages void. Consequently, the authors cannot substantiate the MNIST classification as a pivotal outcome of their work.

Our response: In this paper, apart from real-valued and complex-valued matrix inversion demonstrations, we also include analyses and simulations on two applications where matrix inversion contributes notable compute cost, which are MIMO precoding and ridge regression. We investigate the latency improvement and energy consumption of the matrix inversion step in these two applications assuming the inverse is calculated using a PIP. Finally, we include the MNIST training task to showcase the potential applications of our PIP in the recently rapidly developing artificial intelligence field. The MNIST classification task includes two stages: training and inference. The compute cost of a single training is much more than that of a single inference¹⁴. We focus on the latency improvement of our PIP in the training stage. Different from the traditional gradient descent-based training methods, we use reservoir computing to train the weights in the neural network, which can be reduced to a ridge regression problem¹⁵ as $\mathbf{W}_{out} = \mathbf{Y}_d \mathbb{O}_{total}^T (\mathbb{O}_{total} \mathbb{O}_{total}^T + \lambda \mathbf{I})^{-1}$, where \mathbf{W}_{out} is the weight matrix, \mathbf{Y}_d is the desired output, \mathbb{O}_{total} is a linear feature vector containing the training data points, λ is the ridge parameter and \mathbf{I} is the identity matrix. A large number of ridge parameters λ need to be chosen to find the optimal fit of the training results, which requires repetitive computations of matrix inversion. By offloading the matrix inverse step to our PIP, the latency and energy efficiency of the training process can be largely improved. In our simulations of the MNIST training, we assume the inverse is computed on a PIP and the remaining computations are implemented in digital electronic computers. The processing time of a 10×10 PIP and a 10×10 PSP is estimated according to the estimations and predictions detailed in Supplementary 6.1-6.3.

We revise the descriptions about the MNIST training task in **Discussion** and **Supplementary 7**. The **Abstract** is revised as well.

Discussion:

We further evaluate the processing time improvement of the proposed PIP over PSPs in a more complex machine learning task, specifically, training the MNIST dataset of handwritten digits using reservoir computing. Reservoir computing simplifies the training process to a ridge regression task, which necessitates a large number of ridge parameters to achieve optimal training results. This requires repetitive computations of matrix inversions, assumed in this paper to be performed using a 10×10 PIP. The estimations for length and processing time of this operation are detailed in Supplementary 6.2-6.3. The remaining computations during the training process are conducted on digital electronic computers. For the task of training 60000 MNIST samples using a 10×10 PIP, it is necessary to perform 6000 regression estimations in each training epoch, equivalent to calculating 6000 matrix inversion results. We observe a latency improvement when training 60000 MNIST samples for at least 10 epochs with the PIP, compared to a single-pass processor: 53 seconds for TO modulation and 16 seconds for EO modulation, respectively (Supplementary 7).

Supplementary 7:

The MNIST classification task includes two stages: training and inference. The compute cost of a single training is much more than that of a single inference. We focus on the latency improvement of our PIP in the training stage. Different from the traditional gradient descent-based training methods, we use reservoir computing to train the weights in the neural network, which can be reduced to a ridge regression problem³⁴ as $\mathbf{W}_{out} = \mathbf{Y}_d \mathbb{O}_{total}^T (\mathbb{O}_{total} \mathbb{O}_{total}^T + \lambda \mathbf{I})^{-1}$, where \mathbf{W}_{out} is the weight matrix, \mathbf{Y}_d is the desired output, \mathbb{O}_{total} is a linear feature vector containing the training data points, λ is the ridge parameter and \mathbf{I} is the identity matrix. A large number of ridge parameters λ need to be chosen to find the optimal fit of the training results, which requires repetitive computations of matrix inversion. By offloading the matrix inverse step to our PIP, the latency and energy efficiency of the training process can be largely improved.

For the task of training 60000 MNIST samples, the latency improvement of a PIP over a PSP is estimated by the multiplication of the time difference for each regression estimator and either a variation of parameter or a certain number of epochs. For a conservative estimation, we choose the minimum number of λ and epoch number. Conventionally, at least 10 epochs are required for the MNIST task for convergence of the model which gives us a lower bound estimation of the time difference³⁵. For the task of training 60000 MNIST samples using a 10×10 PIP, it is necessary to perform 6000 regression estimations in each training epoch, equivalent to calculating 6000 matrix inversion results. We observe a latency improvement when training 60000 MNIST samples for at least 10 epochs with the PIP, compared to a single-pass processor: 53 seconds for TO modulation and 16 seconds for EO modulation, respectively.

Abstract:

With much less demand for data movement among compute and memory units, up to two orders of magnitude speed enhancement by using a PIP instead of a single-pass optical processor is estimated for ridge regression tasks and MIMO precoding tasks.

5. In the introduction section, the author extensively delves into a review of advancements in the electronic computing realm, a narrative that appears unrelated to the central focus of this paper. Notably, the core problem of iterative matrix inversion lacks sufficient description, diverting attention from the principal theme of the manuscript.

Our response: Many thanks for your valuable suggestions on how to properly introduce our work. Echoing with our response to your comment 1, we have made the following revisions to the

Introduction: (1) Trim the descriptions about electronic computing. (2) Explain the IO bottleneck and introduce the concept of computation-to-IO (C-to-IO) ratio. (3) Add a paragraph introducing the matrix inversion problems and the iterative methods. We illustrate the advantage of the PIP structure by comparing the C-to-IO ratios of digital electronic computers, PSPs, and PIPs for matrix inversion tasks. Please find the revised introduction below.

Introduction:

The input/output (IO) data movement in a processor typically has limited bandwidth and consumes considerable energy, leading to performance bottlenecks in processing systems that receive input data and store output data through an attached host with memory¹⁻⁴. The IO issue becomes particularly severe when the problem size exceeds the capacity of the processor, a situation that is often encountered in practice. Thus, enhancing the computation-to-IO (C-to-IO) ratio becomes a key focus, i.e. increasing the number of implemented operations for a given input/output data size. In computation intensive tasks such as matrix multiplications and inversions, where multiple operations are performed on each piece of input data, the C-to-IO ratio can be improved by reusing the input data. The tensor processing unit (TPU) exemplifies this approach in accelerating neural network inference tasks. The TPU utilizes a systolic architecture to compute matrix multiplications in parallel by reusing the input data, demonstrating significant speed and energy efficiency improvements over central processing units (CPUs) and graphics processing units (GPUs)². Nevertheless, the challenge of keeping pace with the ever-growing demands of compute-intensive applications has urged people to keep seeking faster and more energy-efficient solutions.

Photonic processors have long been considered as a promising alternative in algebra computations, owing to its inherent high parallelism and low energy consumption⁴⁻⁷. Photonic processors have shone through in the field of matrix multiplications, notably been demonstrated in voice recognition⁸, image classification^{9,10}, and optical communication¹¹. However, few research has been done on using photonic processors to accelerate matrix inversion, a more computationally expensive operation that is fundamental to many scientific and engineering problems such as numerical computations¹²⁻¹⁶, statistics^{12,17}, wireless communication systems¹⁸⁻²² and neural network training^{23,24}. In most practical applications, the matrices to be inverted are diagonally dominant or sparse, and approximate inversion results are often sufficient for subsequent processing stages. Iterative inversion methods generally outperform direct methods in these scenarios since one can easily balance speed and accuracy by terminating the iterative process after several iterations, while interrupting the inversion process of direct methods midway yields incorrect results¹². Classical iterative matrix inversion algorithms essentially perform matrix multiplications and additions iteratively. Implementing such algorithms on a widely reported photonic single-pass processor (PSP)^{8,9,25,26} necessitates repetitive inputting and outputting of computation results in each iteration, thus generating heavy IO traffic.

In this paper, we report a novel photonic iterative processor (PIP) based on reconfigurable photonic integrated circuits for speeding up matrix inversion tasks. The inclusion of optical loopback enables iterative computations for direct matrix inverting with an enhanced I/O efficiency. The processor core is a matrix-vector multiplier comprising MZI units, which reduces the computation overhead by directly encoding the matrix elements on MZI arrays. Table 1 compares the C-to-IO ratios for inverting an $N \times N$ matrix using the iterative Richardson method¹² on four different platforms including the CPU, TPU, PSP, and our proposed PIP. The Richardson method is assumed to converge after $P + 1$ iterations. According to the table, the PIP exhibits the highest C-to-IO ratio of $N^3P/(2N^2 + N)$ among the four computing platforms, followed by the PIP, the TPU, and the CPU. In most application scenarios, P is much larger than N , indicating an improvement of at least N times in the C-to-IO ratio for PIPs compared to other platforms (See Supplementary 1 for more details).

TABLE 1**A COMPARISON OF C-TO-IO RATIOS FOR INVERTING AN $N \times N$ MATRIX USING THE RICHARDSON METHOD**

	CPU	TPU	PSP	PIP
Input data size	$2N^2 \cdot P$	$N^2 \cdot P$	$N^2 \cdot P$	N
Output data size	$N^2 \cdot P$	$N^2 \cdot P$	$N^2 \cdot P$	N^2
Memory access counts*	$3N^2 \cdot P$	$N^2 \cdot (2P + 1)$	$N^2 \cdot (2P + 1)$	$2N^2 + N$
Number of operations†	$N^3 \cdot P$	$N^3 \cdot P$	$N^3 \cdot P$	$N^3 \cdot P$
C-to-IO ratio	$N/3$	$NP/(2P + 1)$	$NP/(2P + 1)$	$N^3P/(2N^2 + N)$

* This includes the number of times a processor needs to fetch input data, store output data, and load weight matrix (the matrix to be multiplied in each iteration). For CPU, the weight matrix is also fetched in a similar way as the input data and is therefore included in the input data size.

† Note the computations in the first iteration are neglected in the analyses for two reasons: (1) if the process converges for $P = 1$, then the matrix to be inverted is essentially an identity matrix whose inverse is itself and does not need to be computed. (2) In the first iteration, the algorithm essentially implements N multiply-and-accumulate (MAC) operations, which can be neglected compared to the N^2 MAC operations in the following iterations.

6. Regarding the second experiment involving integral and differential equations, the authors have not elucidated their connection to the overarching theme of "matrix inversion" as indicated in the title. I am of the opinion that this content may be deemed extraneous to the manuscript and may not warrant inclusion in the main body of the text.

Our response: Thanks for pointing out our presentation problem. The core operation of numerically solving integral and differential equations is matrix inversion. We demonstrate the equation solving examples and show the results in a separate section to the real-valued matrix inversion since in the equation solving examples, matrices with sizes larger than the 4×4 processor size need to be inverted. We employ block matrix computation techniques to decompose the 8×8 (integral equation and ODE) and 16×16 (PDE) matrix inversion tasks into 4×4 matrix computations including inversions, multiplications, additions and subtractions and implement all 4×4 matrix computations using the 4×4 lossless PIP system. Since matrix inversion is the main focus of our paper, we move the descriptions of matrix multiplication, addition and subtraction methods into **Supplementary 5.2**.

We included explanations of how to convert the integral and differential equation solving problem into matrix inversion problem in the **Method** section in the original manuscript: "Eq. (2) can be expressed in a matrix form as $\mathbf{f} = \mathbf{c} + \frac{b-a}{N} \mathbf{Kf}$. The solution of the linear equation is $\mathbf{f} = (\mathbf{I} - \frac{b-a}{N} \mathbf{K})^{-1} \mathbf{c}$ " and "The discretized equations (6)-(10) are then mapped into coefficient matrices which describes the relation between a point and other points in the grid. Solving differential equations are then converted to solving matrix inversions". The block matrix computation methods are introduced in **Supplementary 5.1**. We apologize for not pointing out the connections between equation solving and matrix inversion directly in the main text as this is not straightforward for readers from different fields. We have revised this section and highlight its connections to matrix inversions as follows:

Solving real-valued integral and differential equation with matrix inversions

Integral and differential equations offer a powerful tool for quantifying the dynamics of systems that change over time or space, making them widely used in scientific research and engineering. Numerical solutions instead of analytical solutions are usually taken for real-world problems, which is essentially the problem of matrix inversions (See Method and Supplementary 3 and 5). Using the lossless PIP system that operates iteratively to solve integral and differential equations provides a novel computing paradigm that significantly reduces the

demand of data movement. We adopt the system to solve an integral equation (IE, Fredholm integral equation of the second kind, Eq. (3a)), a second order ordinary differential equation (2nd order ODE, Eq. (3b)) with both using an 8-point discretization, and a partial differential equation (PDE, Poisson equation, Eq. (3c)) using a 4-point discretization.

$$f(t) = 1 + \int_{-1}^1 0.2\sqrt{t^2 + s^2}f(s)ds, t \in [-1,1] \quad (3a)$$

$$d^2/dx^2 [f(x)] - 2x \cdot d/dx [f(x)] - 50f(x) = -1, x \in [-1,1], f(-1) = f(1) = 1 \quad (3b)$$

$$\partial^2 u(x, y)/\partial x^2 + \partial^2 u(x, y)/\partial y^2 = -2\pi^2 \sin(\pi x) \sin(\pi y), x, y \in [0,1], \partial_u = 0 \quad (3c)$$

An N -point discretization corresponds to $N \times N$ matrix **inversions** for IE and ODE, while it corresponds to $N^2 \times N^2$ matrix **inversions** for PDE.

7. Figure 5(b) lacks detailed elucidation from the author, and the relationship between the measured and ideal values in the figure is not clearly discernible. Additional clarification and description are necessary to facilitate a comprehensive understanding of the content presented in Figure 5(b).

Our response: Thanks for pointing out our presentation problems. Figure 5b is a polar plot used to show the complex-valued results during the inversion process. We intended to show the measured and ideal inverse results of a 2×2 diagonal matrix, A_3 , in three iterations. We agree that Fig. 5b does not show the results in each iteration in a clear way. We now use normal plots to visualize the changes in the real and imaginary parts of the two diagonal elements. In addition, we have added the inversion results of two arbitrary non-diagonal matrices, A_4 , and A_5 in Fig. 5d-g. The revisions we have made to the “Complex-valued matrix inversions” section are presented below.

Complex-valued matrix inversions:

One of the major applications of complex-valued matrix inversion is in MIMO decoding and precoding¹⁹⁻²¹, which are fundamental to 5G/6G wireless communication systems. These algorithms require the inversion of complex-valued channel matrices, whose dimensions significantly increase due to the growing data demand from an increased number of end-users. Large scale complex-valued matrix inversion is computationally expensive and faces speed and power efficiency limitations in digital electronic computers. Unlike traditional methods that store a complex number as two separate real parts and process each part individually, photonic processors manipulate the amplitude and phase of optical signals simultaneously, enabling truly complex-valued computations with enhanced efficiency. Additionally, the PIP system can further improve the C-to-IO ratio and increase the inversion speed through its iterative optical loopback, significantly reducing IO access.

Here we show a coherent PIP system as shown in Fig. 5a for complex-valued matrix inversions. With optical loopback paths integrated on-chip, stable phase control can be achieved together with ultrafast processing time. Optical switches are integrated on-chip to facilitate device **characterisation**. Coherent outcomes are read out by off-chip balanced photodetectors (BPD) and captured in the OSC. EDFAs are used to compensate for coupling loss only and BPFs are used to remove excess ASE noise. **The coherent PIP system is used to find the inverse of a 2×2 diagonal matrix $A_3 = \begin{bmatrix} 0.92 + 0.07i & 0 \\ 0 & 1.07 - 0.07i \end{bmatrix}$ and two 2×2 non-diagonal matrices, $A_4 = \begin{bmatrix} 0.98 - 0.01i & 0.07 - 0.01i \\ 0.08 - 0.05i & 1.01 - 0.12i \end{bmatrix}$, and $A_5 = \begin{bmatrix} 0.94 + 0.03i & -0.05 - 0.06i \\ 0.02 + 0.01i & 0.98 + 0.04i \end{bmatrix}$.** The measured and ideal inversion results for the three matrices are shown in Fig. 5b, 5d, and 5f respectively, indicating a very good agreement with the ideal process. Figure 5c, 5e, and 5g shows the evolution of inversion accuracy of three

matrices during multiple iterations. The inversion accuracy reaches 98.8%, 98.3% and 98.9% for A_3 , A_4 , and A_5 , respectively.

Figure 5 | Demonstrated coherent PIP system and complex-valued matrix inversions. (a) Experimental set-up of the coherent PIP system. (b) Ideal and measured inversion process of two diagonal elements of A_3 . (c) Evolution of inversion accuracy of A_3 during convergence. (d) Ideal and measured inversion process of A_4 . (e) Evolution of inversion accuracy of A_4 during convergence. (f) Ideal and measured inversion process of A_5 . (g) Evolution of inversion accuracy of A_5 during convergence.

8. In the description of Fig. 6c and Fig. 6d, I don't understand the significance of the author giving a comparison under the three platforms. The figure shows that SiN requires the longest length, is it representing SiN as the worst choice.

Our response: In the original manuscript, Fig. 6c and 6d are related to processing time analyses. The reason we compare the loop length and single-iteration's processing time on three platforms is to predict the performance limitations and scalability of the PIP on three mainstream photonic integration platforms. We feel this part is not very relevant to the main claims of our paper and have moved these two estimations into Supplementary 6.2. We use the state-of-the-art reported values to estimate the loop length and due to the relatively small refractive index difference, the PIP on a SiN platform has the longest loop length. However, the length difference is not very significant that the single-iteration's processing time is similar on three platforms in an optimised situation. Considering the loop gain estimation in Supplementary 6.1, the SiN platform has the lowest loss among three platforms and a SiN PIP requires the least gain among the three. Hence, the introduced ASE noise is the least on the SiN platform.

6.2 Loop length estimation of an $N \times N$ PIP

After determining the optimal number of SOA stages, the loop length of the PIP needs to be estimated to evaluate the processing time in a single iteration. Table S3 lists the lengths of each building block in the PIP based on either reported designs in the literature or practical layout designs. The estimated total loop length, which is twice the single-pass length, is shown in Fig. S11. This is a relatively conservative estimate since the loop length can be shortened with careful design. As shown in Fig. S11, the single-pass length for a 256×256 processor remains below 100 mm. Considering the typical wafer diameters of SOI-based or SiN-based platforms to be 12-inch (300 mm) and of InP-based platforms to be 4-inch (100 mm), a 256×256 PIP can easily fit within the commercial wafer sizes, and the device footprint is not a limiting factor to its scalability.

TABLE S3
LOOP LENGTH ESTIMATION OF AN $N \times N$ PIP ON THREE PHOTONIC INTEGRATION PLATFORMS

	SOI (μm)	SiN (μm)	IMOS (μm)
Adders & Splitters	$11.3^{19} \cdot (2\log_2 N + 2)$	$13^{11} \cdot (2\log_2 N + 2)$	$11.8 \cdot (2\log_2 N + 2)$
Bend	$4^{13} \cdot 2(2\log_2 N + 2)$	$20^{11} \cdot 2(2\log_2 N + 2)$	$5 \cdot 2(2\log_2 N + 2)$
Crossing	$10 \cdot \log_2 N$	$33 \cdot \log_2 N$	$16 \cdot \log_2 N$
Phase shifter	200	200	200
MZI weight bank	$(2L_{MMI} + 2L_{bend} + L_{phase\ shifter}) \cdot N$		
Switch	$3L_{MMI} + 8L_{bend} + 2L_{phase\ shifter}$		
SOA & BPF	$(500^* + 40) \cdot k_{opt}$	$(500 + 80) \cdot k_{opt}$	$(500 + 48) \cdot k_{opt}$

* A $500 \mu\text{m}$ SOA provides gain up to 15 dB²⁰.

Figure S11 | Loop length and single iteration's processing time estimation of an $N \times N$ lossless PIP. (a) Estimated loop length of the PIP. The total loop length is estimated by doubling the single-pass length which is a direct summation of the length of each building block. (b) Estimated single-iteration's time of the PIP.

9. In the discussion section, the authors delve into various hybrid and heterogeneous integration methods. However, the relevance of these investigations to the specific research presented in the manuscript is unclear, and it seems that most of these techniques are not actually employed by the authors. The section can at most appear in the introduction parts.

Our response: We hope you will agree that the **Discussion** section is a place to explore the performance limitations of the PIP and discuss possible future work. We include a paragraph detailing the hybrid and heterogeneous integration methods in order to show the readers the technologies for building a fully integrated PIP are already available and our proposed PIP is not unrealistic. In order to ensure the flow of the **Discussion** section, we have added sub-titles to aid the readers:

Photonic integration platforms for realising a fully integrated PIP

In this paper, both the lossless and coherent PIP systems are formed with a SiN chip core and off-the-shelf components. To improve the latency, scalability, energy efficiency and cost-effectiveness of the PIP system, it is essential to move towards a fully integrated processor on a single chip. Such a candidate chip design is illustrated in Fig. 1. A number of hybrid and heterogeneous integration methods have been investigated to combine the power of III-V with the full capability of SOI, spanning from flip-chip bonding²³, die/wafer bonding^{24,25}, micro-transfer printing²⁶, to direct epitaxial growth^{27,28}. Epitaxial growth may represent an ultimate path but die/wafer bonding features a higher technology readiness level in a short term. Membrane technology^{29,30}, such as the InP membrane on silicon (IMOS), also represents a significant platform that co-integrates passive and active components. Also, high bandwidth modulators and photodetectors are available on the silicon-on-insulator platform³¹, III-V platform^{32,33} and the hybrid integration platform^{34,35}. The capability of reconfiguring phase shifters can at nanosecond- or even picosecond-scales can significantly reduce the weights reloading time, achieving even faster computations.

Reviewer #2 (Remarks to the Author)

The authors combine the advantage of executing operations in a fully-optical domain and iterative matrix inversion to empower the computation-to-IO ratio.

While the paper and figures are well structured, I would suggest clarify the following parts:

1. The matrix inversion case could be better explained, at the moment is not clear what are the usage possibilities, despite the large list of applications mentioned. I would better treat the case as a tough-to-be-solved with current SOTA electronics and why. This is not clear, which does not emphasize the importance to move into the optical domain as a first.

Our response: Many thanks for your valuable suggestions on how to properly introduce the motivations and focus of our work. The main applications for matrix inversion include numerical computations, statistics, wireless communication systems and neural network training. We have made the following revisions to the Introduction to emphasize the motivations of using optics to compute matrix inversions: (1) Explain the IO bottleneck and introduce the concept of computation-to-IO (C-to-IO) ratio. (2) Add a paragraph introducing the matrix inversion problems and the iterative methods. We illustrate the advantage of the PIP structure by comparing the C-to-IO ratios of digital electronic computers, PSPs, and PIPs for matrix inversion tasks. Please find the revised introduction below.

Introduction:

The input/output (IO) data movement in a processor typically has limited bandwidth and consumes considerable energy, leading to performance bottlenecks in processing systems that receive input data and store output data through an attached host with memory¹⁻⁴. The IO issue becomes particularly severe when the problem size exceeds the capacity of the processor, a situation that is often encountered in practice. Thus, enhancing the computation-to-IO (C-to-IO) ratio becomes a key focus, i.e. increasing the number of implemented operations for a given input/output data size. In computation intensive tasks such as matrix multiplications and inversions, where multiple operations are performed on each piece of input data, the C-to-IO ratio can be improved by reusing the input data. The tensor processing unit (TPU) exemplifies this approach in accelerating neural network inference tasks. The TPU utilizes a systolic architecture to compute matrix multiplications in parallel by reusing the input data, demonstrating significant speed and energy efficiency improvements over central processing units (CPUs) and graphics processing units (GPUs)². Nevertheless, the challenge of keeping pace with the ever-growing demands of compute-intensive applications has urged people to keep seeking faster and more energy-efficient solutions.

Photonic processors have long been considered as a promising alternative in algebra computations, owing to its inherent high parallelism and low energy consumption⁴⁻⁷. Photonic processors have shone through in the field of matrix multiplications, notably been demonstrated in voice recognition⁸, image classification^{9,10}, and optical communication¹¹. However, few research has been done on using photonic processors to accelerate matrix inversion, a more computationally expensive operation that is fundamental to many scientific and engineering problems such as numerical computations¹²⁻¹⁶, statistics^{12,17}, wireless communication systems¹⁸⁻²² and neural network training^{23,24}. In most practical applications, the matrices to be inverted are diagonally dominant or sparse, and approximate inversion results are often sufficient for subsequent processing stages. Iterative inversion methods generally outperform direct methods in these scenarios since one can easily balance speed and accuracy by terminating the iterative process after several iterations, while interrupting the inversion process of direct methods midway yields incorrect results¹². Classical iterative matrix inversion algorithms essentially perform matrix multiplications and additions iteratively. Implementing such algorithms on a widely reported

photonic single-pass processor (PSP)^{8,9,25,26} necessitates repetitive inputting and outputting of computation results in each iteration, thus generating heavy IO traffic.

In this paper, we report a novel photonic iterative processor (PIP) based on reconfigurable photonic integrated circuits for speeding up matrix inversion tasks. The inclusion of optical loopback enables iterative computations for direct matrix inverting with an enhanced I/O efficiency. The processor core is a matrix-vector multiplier comprising MZI units, which reduces the computation overhead by directly encoding the matrix elements on MZI arrays. Table 1 compares the C-to-IO ratios for inverting an $N \times N$ matrix using the iterative Richardson method¹² on four different platforms including the CPU, TPU, PSP, and our proposed PIP. The Richardson method is assumed to converge after $P + 1$ iterations. According to the table, the PIP exhibits the highest C-to-IO ratio of $N^3P/(2N^2 + N)$ among the four computing platforms, followed by the PIP, the TPU, and the CPU. In most application scenarios, P is much larger than N , indicating an improvement of at least N times in the C-to-IO ratio for PIPs compared to other platforms (See Supplementary 1 for more details).

TABLE 1
A COMPARISON OF C-TO-IO RATIOS FOR INVERTING AN $N \times N$ MATRIX USING THE RICHARDSON METHOD

	CPU	TPU	PSP	PIP
Input data size	$2N^2 \cdot P$	$N^2 \cdot P$	$N^2 \cdot P$	N
Output data size	$N^2 \cdot P$	$N^2 \cdot P$	$N^2 \cdot P$	N^2
Memory access counts[*]	$3N^2 \cdot P$	$N^2 \cdot (2P + 1)$	$N^2 \cdot (2P + 1)$	$2N^2 + N$
Number of operations[†]	$N^3 \cdot P$	$N^3 \cdot P$	$N^3 \cdot P$	$N^3 \cdot P$
C-to-IO ratio	$N/3$	$NP/(2P + 1)$	$NP/(2P + 1)$	$N^3P/(2N^2 + N)$

* This includes the number of times a processor needs to fetch input data, store output data, and load weight matrix (the matrix to be multiplied in each iteration). For CPU, the weight matrix is also fetched in a similar way as the input data and is therefore included in the input data size.

† Note the computations in the first iteration are neglected in the analyses for two reasons: (1) if the process converges for $P = 1$, then the matrix to be inverted is essentially an identity matrix whose inverse is itself and does not need to be computed. (2) In the first iteration, the algorithm essentially implements N multiply-and-accumulate (MAC) operations, which can be neglected compared to the N^2 MAC operations in the following iterations.

2. The computational complexity of $O(N^{1.42})$ – as it is stated, does not stand out, while it would be better to connect to improvements in power consumption too.

Our response: Thanks for your constructive comments. We want to differentiate the two concepts between “computational complexity” and the “computation-to-IO (C-to-IO) ratio”. In short, the “computational complexity” focuses more on the scalability of a processor, which we have explored thoroughly in the revised **Discussion** and **Supplementary 6**. The main claim of our work is the much higher “C-to-IO ratio” of the PIP compared to PSPs and digital electronic processors. Given the much-reduced IO access counts, our PIP exhibits up to two orders of magnitude speed enhancement in ridge regression and MIMO precoding tasks. The original $O(N^{1.42})$ is from the simulation results of random matrices, which we think do not have enough meanings in practice. In the revised manuscript, we choose two inversion-intensive tasks to evaluate the processing time enhancement of our PIP, leading to the conclusion that the PIP stands out among the state-of-the-art electronic processors and PSPs as accelerators for applications involving intensive matrix inversion computations. These are detailed in our response to your comment 1 and our revisions in the **Introduction**, **Discussion-Processing time (latency) of an $N \times N$ PIP** in the main text, and **Supplementary 1** and **6**.

We apologize for not including the energy consumption analyses. Please find the analyses and discussions in the **Discussion-Energy efficiency of an $N \times N$ PIP** in the main text, and **Supplementary 6.6**. Though the demonstrated energy efficiency of the lossless PIP and coherent PIP does not stand out due to the energy loss during coupling and the lower power efficiency of discrete components. The estimated energy efficiency of our PIP, especially when wavelength multiplexing techniques are used, stands out among other PSPs and digital electronic processors, indicating that the PIP structure is an energy-efficient architecture for matrix inversion tasks.

Discussion

Energy efficiency of an $N \times N$ PIP

Energy efficiency of the PIP core is defined as number of operations per second per energy consumption of the PIP core. The components consuming energy during the matrix inversion process include lasers, modulators, semiconductor optical amplifiers (SOAs), photodetectors (PDs), the MZI weight bank, and optical switches. The energy efficiency of the lossless PIP and the coherent PIP, together with the estimated energy efficiency of the PIP (with and without wavelength multiplexing techniques) and that of the state-of-the-art electronic and photonic processors are shown in Fig. 6f, indicating that the PIP structure is an energy-efficient architecture for matrix inversion tasks. More details can be found in Supplementary 6.6.

Supplementary

6.6 Energy efficiency estimation of an $N \times N$ PIP

TABLE S7

ESTIMATED ENERGY CONSUMPTION OF THE PIP CORE FOR $N \times N$ MATRIX INVERSIONS

Component	Equation
Laser	$E_{laser} = P_{laser} / \eta \cdot N \cdot P \cdot t_{loop}$
Modulator	$E_{mod} = V_{mod}^2 / R \cdot N \cdot P \cdot t_{loop}$
SOA	$E_{SOA} = (P_{out} - P_{in}) / \eta \cdot N^2 \cdot P \cdot t_{loop}$
PD	$E_{PD} = \mathcal{R} P_{rec} \cdot V_{bias} \cdot N^2 \cdot t_{loop}$
Weight bank	$E_{weight} = P_{heater} \cdot N^3 \cdot P \cdot t_{loop}$
Optical switch	$E_{switch} = V_{mod}^2 / R \cdot N^2 \cdot P \cdot t_{loop}$

Energy efficiency of the PIP core is defined as number of operations per second per energy consumption of the PIP core. Table S7 lists the equations to estimate the energy consumption of the PIP core for inverting an $N \times N$ matrix. P_{laser} is the output power of the laser, which should be at least -5 dBm for PIP size not exceeding 64×64 and 10 dBm for 128×128 and 256×256 PIPs according to Supplementary 6.5. η is the wall-plug efficiency of the laser and the semiconductor optical amplifier (SOA), which has a best reported value of around 30%²⁹. V_{mod} is the driving voltage of the modulator and the optical switch, which is around 250 mV³⁰ for the best reported electro-optic modulators. $R = 50\Omega$ is the load impedance for modulators and optical switches. P_{out} and P_{in} are input and output powers of the SOA and are size dependent since the loop loss increases as the matrix size increases. The estimated required gain for an $N \times N$ processor is presented in Supplementary 6.1. $\mathcal{R} \approx 0.6A \cdot W^{-1}$ is the responsivity of the photodetector. P_{rec} is the received optical power at the photodetector, which should be same as P_{laser} in the lossless PIP system. The minimal required signal power P_{laser} for reaching a certain inversion accuracy is also size dependent. The inversion accuracy vs. input signal power for different-sized processor is shown in Fig. 6e. $V_{bias} \approx 3V$ is the bias voltage of photodetector. P_{hea} is the power consumption of thermo-optic phase shifters in the weight bank, whose best reported value

is $0.05\text{mW}/\pi$ phase shift. t_{loop} is the core processing time in a single iteration, which depends on the loop length and the effective index of the integration platform as shown in Fig. 6c and Supplementary 6.3. Fig. S14 illustrates the estimated energy consumption and energy efficiency of the PIP core. Fig. S14a shows the energy consumption of the PIP core for computing matrix inversions in the ridge regression task and the MIMO task respectively. The minimal and maximal energy consumptions are illustrated respectively based on the minimal and maximal inversion time shown in Table S4.

Figure. S14b shows the estimated energy efficiency of the PIP core (with and without multiplexing techniques) in terms of tera-operations per second per watt, together with the energy efficiency of the demonstrated PIP systems in this work and several state-of-the-art electronic and photonic processors to showcase the advancement in energy efficiency of our proposed PIP. Though the demonstrated coherent PIP and lossless PIP is less energy efficient than electronic processors when only the processing core is considered mainly due to the limited integration level, the predicted energy efficiency of a fully integrated lossless PIP outperforms state-of-the-art electronic and photonic processors. Considering the much-reduced IO access counts of our PIP. It is safe to say the proposed PIP is more energy efficient in matrix inversion tasks compared to traditional electronic processors and photonic processors.

Figure S14 | Estimated energy efficiency of the lossless PIP core for inverting an $N \times N$ matrix. (a) Energy consumption of the PIP core. (b) Energy efficiency of the PIP core.

3. The final comparison – as indicated in the intro section, but also in Fig. 6 – does not include the SOTA in electronics and comparison with it. There is a comparison for the use-cases treated for the real-valued matrix inversion, but it is not enough to convince this is the way to go.

Our response: Thanks for your valuable comments. According to our responses to your first two comments, we believe that it is enough to convince the readers that photonic iterative processors stand out in processing speed and energy efficiency in applications which require a lot of matrix inversion computations due to the much-reduced IO access. Specifically, the comparisons with the SOTA in electronics are included in Table S6 and Fig. 6f.

4. It is not clear to me how to perform the subtraction. Could you clarify, and indicate the operation in figure 1?

Our response: Matrix subtraction is essentially the same as matrix addition with only a π phase shift in the signal. The phase shift $e^{j\pi} = -1$ changes the sign, converting addition into subtraction.

Considering implementing vector subtraction, $\mathbf{a} - \mathbf{b}$, where \mathbf{a} and \mathbf{b} are $N \times 1$ vectors. Before computation, $-\mathbf{b}$ is preloaded into the first column of the $N \times N$ weight bank. During initial preparation, a unit vector $\mathbf{e}_1 = [1, 0, 0, \dots, 0]^T$ is generated by modulating the laser in modulators. After passing the weight bank and the loss compensation blocks, $-\mathbf{b}$ is retrieved and appears at the input summation block. By synchronizing the signal timings, the vector \mathbf{a} is concurrently applied to the modulators and arrives at the input summation block with $-\mathbf{b}$. After the first iteration, the summation between \mathbf{a} and $-\mathbf{b}$, or equivalently, the subtraction between \mathbf{a} and \mathbf{b} is performed.

Since matrix subtraction is not the main focus of our work, we mentioned it in the last sentence of “**Results-Photonic iterative processor architecture**” in the main text as:

The PIP is also capable of computing matrix addition and multiplication in a single iteration by proper configuration. Detailed descriptions can be found in the supplementary 5.2.

We included the detailed operations of multiplication, addition/subtraction and inversion in Supplementary 5.2 as:

5.2 Configurations of the PIP for different matrix computations

Fig. S9 illustrates different configurations of the PIP for different matrix computations in detail. To compute matrix addition or subtraction, i.e. $\mathbf{A} \pm \mathbf{B}$ (the subtraction process is similar to that of addition, with only a π phase shift difference), one of the matrix operands, \mathbf{A} , is encoded in the weight bank before computation. In the initial preparation, a unit vector is injected to the input to retrieve one column of matrix \mathbf{A} , with part of the unit vector appearing at the output. The input is next updated to one column of matrix \mathbf{B} , which is added to the retrieved column of \mathbf{A} in the 1st iteration. The loop is then terminated with the output (after splitting loss compensation) showing the addition/subtraction result of one column of \mathbf{A} and \mathbf{B} .

To compute matrix multiplication, i.e. $\mathbf{A} \times \mathbf{B}$, \mathbf{A} is again first encoded in the weight bank. Differently, in the initial preparation, one column of \mathbf{B} is injected to the input, with part of it arriving directly at the output. Next, the remaining part passes the weight bank to complete a matrix-vector multiplication, and the output in the 1st iteration presents the multiplication result. The loop is afterwards terminated.

For the computation of the inverse matrix, multiple iterations are needed to let the output converge

Figure S9 | Configurations of the PIP for different matrix computations. Configurations of the PIP for different matrix computations. Matrix additions, subtractions, and multiplications are solved in 1 iteration while inversions are solved in multiple iterations.

autonomously. The initial step is to load $\mathbf{M} = \mathbf{I}_N - \omega\mathbf{A}$ in the weight bank. A unit vector weighted by ω is injected to the input in the input preparation, with part of it reaching the output. In the 1st iteration, the weighted unit vector is again added to the multiplication result of \mathbf{M} and the input vector. This procedure is repeated until the output converges at the n^{th} iteration to one column of \mathbf{A}^{-1} , and the loop is subsequently terminated.

5. The PIP generates matrix computation results one column at a time: when coming with the latency calculation, you should differentiate between this scheme and the multiplexing scheme in [39].

Our response: You are correct that we have made a mistake in calculating the net inversion time (core processing time) of the PIP. The PIP shown in Fig. 1 generates the matrix computation results one column at a time, which means the net inversion time (core processing time) is $t_{core} = t_{loop} \cdot P \cdot N$. When wavelength multiplexing is used, the net inversion time is $t_{core} = t_{loop} \cdot P$. We have included two more complex-valued inversion experiments and update our shortest net inversion time of real-valued experiments and complex-valued experiments to 2.6 μs and 1.2 ns respectively. We also include these two schemes in the latency improvement analyses shown in Fig. 6c, 6d, and the energy efficiency analysis shown in Fig. 6f.

6. The authors mention that the filters are used to clean the signals, however the in-band noise is never removed, even with filters and could build up in the iteration process. As this is never mentioned, also in the final calculations, I would suggest to comment on this issue and how this can be solved. And of course when it is an issue, what is the expected degradation in performance.

Our response: You are correct that the in-band ASE noise builds up in the iterative process. We have performed simulations to estimate the inversion accuracy degradation of the PIP as detailed in **Discussion-Inversion accuracy of an $N \times N$ PIP** and **Supplementary 6.5**. In short, the ASE noise can be reduced as much as possible by using a very narrow bandwidth filter but can never be removed. The degradation in accuracy becomes more significant as the processor size increases. For ultra-large-sized problems, the compromise of signal integrity due to PIC insertion loss is still a current bottleneck, and it could be solved by employing block matrix inversion techniques as demonstrated in **Solving real-valued integral and differential equation with matrix inversions** in the main text and **Supplementary 6.4**.

Discussion

Inversion accuracy of an $N \times N$ PIP

Inversion accuracy is defined in terms of matrix norm as: $accuracy = (1 - \varepsilon) \times 100\% = (1 - \|\mathbf{A}_{meas}^{-1} - \mathbf{A}_{ideal}^{-1}\| / \|\mathbf{A}_{ideal}^{-1}\|) \times 100\%$, where ε is the inversion error, \mathbf{A}_{meas}^{-1} is the measured or simulated inversion results, and \mathbf{A}_{ideal}^{-1} is the ideal or theoretical inversion results calculated in a traditional digital electronic computers. Three main error sources when performing matrix inversions on a fully integrated PIP include: 1) quantization error, 2) ASE noise introduced during amplification, and 3) thermal and shot noise introduced during detection. The estimated inversion accuracy of a PIP with sizes ranging from 2×2 to 256×256 is shown in Fig. 6e, indicating a relatively good inversion accuracy for a processor size of up to 256×256 . More details can be found in **Supplementary 6.5**.

Supplementary

6.5 Inversion accuracy of an $N \times N$ PIP

As briefly mentioned in Supplementary 1.2, inversion accuracy is defined in terms of matrix norm as: $accuracy = (1 - \epsilon) \times 100\% = (1 - \|A_{meas}^{-1} - A_{ideal}^{-1}\| / \|A_{ideal}^{-1}\|) \times 100\%$, where ϵ is the inversion error, A_{meas}^{-1} is the measured or simulated inversion results, and A_{ideal}^{-1} is the ideal or theoretical inversion results calculated in a traditional digital electronic computers. Four main error sources when performing matrix inversions on the PIP include: 1) quantization error, 2) ASE noise introduced during amplification, 3) thermal and shot noise introduced during detection, and 4) phase drift in the fibres. The four error sources are analysed below, together with an interpretation of the simulated inversion accuracy results shown in Fig. 6e in the main text. Methods for mathematical modelling and choices of simulation parameters can be found in our previous paper³.

Figure S13 | Simulated inversion accuracy of an $N \times N$ lossless PIP. (a) Inversion accuracy vs. Input signal power of an $N \times N$ lossless PIP. (b) Inversion accuracy vs. Filter bandwidth of an $N \times N$ lossless PIP. The matrices to be inverted are 1000 randomly generated matrices for each size.

1) Quantization error: During matrix weights loading and outputs acquisition, the digital-to-analogue and analogue-to-digital conversion (DAC and ADC) inevitably introduces quantization errors. In the lossless PIP demonstration, a 10-bit DAC and ADC resolution is used limited by the resolutions of the oscilloscope. Higher-bit resolution can be employed if the applications require higher accuracies. In the simulation results which predict the scalability of the PIP shown in Fig. 6e and Fig. S13a, a 16-bit resolution is used.

2) ASE noise introduced during amplification: ASE noise from amplification contributes most to the computation error, and it degrades the signal quality a bit in each iteration. In the lossless PIP demonstration, a bandpass filter with a 3 dB bandwidth of 0.1 nm is used to suppress the ASE noise. Narrower-bandwidth filters such as ring filters²³ can be employed if the application requires higher accuracy. Additionally, in the current partially integrated system, coupling between fibre arrays and edge couplers introduces extra loss (an average of ~12dB loss is measured per 2 facets for 4 pairs of input and output ports). Higher gain is required to compensate for the coupling loss, which in turn introduces additional ASE noise. This can be eliminated in a fully integrated chip. In the simulation shown in Fig. 6e and Fig. S13a, a filter with bandwidth of 906 kHz²⁴ is used. We presented the simulated inversion accuracy vs. filter bandwidth (currently reported laser power is at 19 dBm²⁵ but more efforts are on the way) in Fig. S13b to show the limitations in a more practical integration scheme. Figure 6e is presented again in Fig. S13a to ease the comparisons and analyses. As shown in Fig. S13, we investigate the performance of the PIP with a size up to 256x256. For PIP size not exceeding 64x64, an input signal power of > -5 dBm is enough to guarantee an inversion accuracy of $>90\%$. The input signal power,

however, needs to be enhanced to maintain a certain level of SNR to handle the increased on-chip loss for the PIP of larger scales. A singular value decomposition (SVD) – based weight bank²⁶ can be used to improve the inversion accuracy owing to its reduced signal splitting and combining loss. Nevertheless, the computation overhead for implementing SVD is non-negligible. On the other side, a filter with 3 GHz passband is capable to guarantee a 90% inversion accuracy of a 32×32 PIP, and in order to fight against the increased ASE power, the filter passband consistently goes down for larger-scale PIPs. For problem size exceeding 256×256, block matrix inversion techniques should be employed, considering the current PIC insertion loss and feasible laser power.

3) Thermal and shot noise introduced during detection: thermal noise originates from the TIA circuits, while shot noise arises from photo-conversion. Thermal noise dominates when signal powers are low while shot noise dominates when signal powers are high. In the experiment, the detection noises can be subtracted from the signals in the post-processing step.

4) Phase drift in the fibres: This error source only exists in a partially integrated PIP system. The thermal-optic effect and thermally induced fibre elongation leads to intrinsic periodic fibre phase drift in common experimental conditions²⁷. A typical method of dealing with the fibre phase drift is to use piezoelectrically stretched coiled fibre to compensate for the phase drift²⁸. However, this method requires extra components for each phase-sensitive path and complicated control circuits and algorithms. In this paper, we use averaging to counteract the phase drift in the lossless PIP. The phase stability problem does not exist for the coherent PIP system and is not considered in the simulations.

7. For both the real-valued and the complex-valued matrix inversion, where and how are the engines trained?

Our response: We hope we have correctly understood what you mean by “engines”, which are the weight matrices $(\mathbf{I}_N - \omega\mathbf{A})$ in matrix inversion tasks. We have explained in the “**Results-Photonic iterative processor architecture**” section the meanings of the notations:

Matrix inversion is computed through the Richardson method:

$$\mathbf{X}^{(k+1)} = (\mathbf{I}_N - \omega\mathbf{A})\mathbf{X}^{(k)} + \omega\mathbf{I}_N \quad (k = 0,1,2, \dots) \quad (1)$$

where \mathbf{A} is an $N \times N$ matrix operand to be inverted, \mathbf{I}_N is the $N \times N$ identity matrix, ω is a parameter used to adjust the convergence of the inversion algorithm and the matrix operand is encoded in the weight bank via $\mathbf{I}_N - \omega\mathbf{A}$. $\mathbf{X}^{(k+1)}$ and $\mathbf{X}^{(k)}$ are output matrices after $k+1$ and k iterations, and $\mathbf{X}^{(0)} = \omega\mathbf{I}_N$ is the initial input matrix that initiates the computation.

In case you want to know more about the characterisation of MZIs and how to encode the weight matrix into MZI units, please see detailed explanations about the characterisation of both real-valued and complex-valued weight matrices in the following. We encode the $N \times N$ weight matrix in a weight bank consisting of $N \times N$ MZI units. As an optical signal propagates through one MZI unit, its amplitude and phase will change, which is equivalent to one complex-valued multiplication. In order to encode the elements of the $N \times N$ weight matrix and set each MZI to the required transmission point, the 16 MZI weights in the real-valued experiment and the 4 MZI weights in the complex-valued experiment are characterized before computations. The corresponding output powers and the applied voltages are recorded and stored in a lookup table in a digital electronic computer. To implement a matrix inversion task, the weight matrix $(\mathbf{I}_N - \omega\mathbf{A})$ is first calculated in the digital electronic computer. Then the corresponding voltages for each thermo-optic phase shifter are found in the lookup table. The weight matrix is loaded into our PIP by applying voltages through a homemade electrical control system.

For the real-valued matrix inversion, the transmission power reflects the encoded real-valued matrix elements. we provided a detailed description of the MZI weight characterization methods in **Methods** section in the original manuscript (See the quoted red texts below). Additional details and characterization results are presented in **Supplementary 2**. We acknowledge that the term “calibration” does not precisely describe what we have done. Therefore, we have replaced “calibration” and “calibrate” with “characterisation” and “characterise” in both the **Methods** section and **Supplementary 2**. The specific changes are not listed here to save place but can be found in highlighted red texts in the manuscript).

Methods:

Chip characterisation (lossless PIP). 16 MZI weights are characterised independently by launching a laser into the chip and measuring the output light intensity while sweeping the applied voltage to the heaters on the MZI arms. 16 transmission-voltage (T-V) or transmission-power (T-P) curves are recorded and fitted to form lookup tables for loading matrix weights. The effective weight of each matrix element is a combination of the attenuation in the weighting bank, the loss in the loop including the MZI unit, and the gain in that loop. This is achieved by creating an optical loop between input and output waveguides for a single MZI. The effective weight is then determined by launching an optical pulse into the loop and measuring the ratio of subsequent output pulses. Note that there is an EDFA within the loop whose purpose is to compensate for optical losses in the loop. Finally, the attenuation in the MZI unit is adjusted to provide the required attenuation for each matrix element.

For the complex-valued matrix inversion, both the transmission power and the phase shift of each MZI unit needs to be characterised. We have added a paragraph in the **Methods** section to describe the characterisation method and more details are presented in **Supplementary 2**.

Methods:

Chip characterisation (coherent PIP). In order to characterise the 4 MZI weights, two loop switches, one input switch, and two coherent detectors need to be characterised first. The “On/Off (Bar/Cross)” states of two loop switches are characterised by measuring the transmission of a light signal through two edge-coupled ports of the switch while sweeping the applied voltage to the heaters on the MZI arms. Configuring the loop switches to “On/Off (Bar/Cross)” states corresponds to maintaining/terminating the iterative process. The “Bar/Cross” state of the input switch, which corresponds to injecting unit vectors to two input ports, is characterised with both loop switches configured to “Off” states. Then the two coherent detectors, each comprising an MZI, are characterized respectively by setting the input switch to either “Bar” or “Cross” state and measuring the photo-detected electrical signals with both loop switches in “Off” states. The effective attenuation and phase shift of each MZI weight unit are the total attenuation and phase shift within the loop, which are determined by launching optical pulses into two input ports respectively and measuring the ratio of the first two pulses.

Supplementary 2:

2.4 Characterisation of the loop switch and the input switch

The structure of the two loop switches and the input switch is shown in Fig. S6 – Structure A, which is essentially a 2×2 MZI switch with an MZI-type variable splitter¹⁶ to ensure a high extinction ratio (ER) of the switch element. $\tau_L^2: \kappa_L^2$, $\tau_R^2: \kappa_R^2$, and $\tau_2^2: \kappa_2^2$ are coupling ratios of three multimode interference (MMI) couplers, whose values are fixed after fabrication. Structure B is equivalent to structure A, with the MZI-type variable splitter replaced with a tunable coupler. For an MZI with ideal coupling ratios, the condition for reaching the “Bar” states is $\tau_1 = \tau_2$, $\phi_1 = \phi_2 + \pi$, and the condition for reaching the “Cross” state is $\tau_1 = \kappa_2$, $\phi_1 = \phi_2$. In the 2×2 high-ER MZI, θ_1 and θ_2 are swept to generate the required coupling ratio $\tau_1^2: \kappa_1^2$.

To characterise the high-ER switch, an optical pulse is first launched into one of the input ports of the switch. The output power of one of the output ports is monitored while sweeping ϕ_1 and ϕ_2 to find the minimal transmission point. At the minimal transmission point, θ_1 and θ_2 are then swept to find the minimal and maximal transmission points which correspond to the “Cross” and “Bar” states.

Figure S6 | Schematic of a 2×2 MZI switch with an MZI-type variable splitter. Structure A illustrates the schematic of the high-ER 2×2 MZI switch with an MZI-type variable splitter. $\tau_L^2:\kappa_L^2$, $\tau_R^2:\kappa_R^2$, and $\tau_2^2:\kappa_2^2$ are coupling ratios of three multimode interference (MMI) couplers, whose values are fixed after fabrication. Structure B is equivalent to structure A, with the MZI-type variable splitter replaced with a tunable coupler.

2.5 Characterisation of the coherent detector

Due to the limited space, here we integrate an MZI structure as the coherent detector and monitor the photo-

$$E_{i1} = A_1(t)e^{j(\omega_1 t + \alpha_1)}$$

$$E_{i2} = A_2(t)e^{j(\omega_2 t + \alpha_2)}$$

$$\theta_1 - \theta_2 = \theta$$

$$E_{o1}(t) = je^{j(\theta/2 + \theta_2 + \phi)} [\sin(\theta/2)A_1(t)e^{j(\omega_1 t + \alpha_1)} + \cos(\theta/2)A_2(t)e^{j(\omega_2 t + \alpha_2 + \gamma)}]$$

$$E_{o2}(t) = je^{j(\theta/2 + \theta_2)} [\cos(\theta/2)A_1(t)e^{j(\omega_1 t + \alpha_1)} - \sin(\theta/2)A_2(t)e^{j(\omega_2 t + \alpha_2 + \gamma)}]$$

$$I_1(t) = \mathcal{R}[\sin^2(\theta/2)|A_1(t)|^2 + \cos^2(\theta/2)|A_2(t)|^2 + 2\sin(\theta/2)\cos(\theta/2)A_1(t)A_2(t)\cos(\alpha_1 - \alpha_2 - \gamma)]$$

$$I_2(t) = \mathcal{R}[\cos^2(\theta/2)|A_1(t)|^2 + \sin^2(\theta/2)|A_2(t)|^2 - 2\sin(\theta/2)\cos(\theta/2)A_1(t)A_2(t)\cos(\alpha_1 - \alpha_2 - \gamma)]$$

$$I(t) = \mathcal{R}[-\cos\theta|A_1(t)|^2 + \cos\theta|A_2(t)|^2 + 2\sin\theta A_1(t)A_2(t)\cos(\alpha_1 - \alpha_2 - \gamma)]$$

$$\theta = \frac{\pi}{2}, \gamma = 0 \quad I(t) = 2\mathcal{R}A_1(t)A_2(t)\cos(\alpha_1 - \alpha_2) \quad \text{Real part}$$

$$\theta = \frac{\pi}{2}, \gamma = \frac{\pi}{2} \quad I(t) = 2\mathcal{R}A_1(t)A_2(t)\sin(\alpha_1 - \alpha_2) \quad \text{Imaginary part}$$

Figure S7 | Schematic and principles of the coherent detector. The MZI unit is integrated on-chip while the balanced photodetector is off-chip. By setting the phase difference between two MZI arms to $\pi/2$ and configuring the phase shifter γ to 0 and $\pi/2$, the real and imaginary parts of a complex-valued number are read out respectively.

detected current using an off-chip balanced detector as shown in Fig. S7. The working principles are also illustrated in Fig. S7. By setting the phase difference between two MZI arms to $\pi/2$ and configuring the phase shifter γ to 0 and $\pi/2$, the real and imaginary parts of a complex-valued number are read out respectively.

To characterise θ , an optical pulse is launched into one input port of the coherent detector, and the photo-detected current is monitored while sweeping the voltages applied to θ_1 and leaving θ_2 untouched. The minimal and maximal current points correspond to $\theta = 0$ and $\theta = \pi/2$ respectively. Before characterising γ , θ should be set to $\pi/2$. Then two coherent pulses should be launched into two ports of the coherent detector respectively and the photo-detected current is monitored while sweeping the voltage applied to γ . The zero point corresponds to $\gamma = 0$. The maximal/minimal point corresponds to $\gamma = \pi/2$ depending on the initial phase.

2.6 Characterisation of the complex-valued weight bank in a closed-loop form

The complex-valued MZI units are characterised by launching optical pulses into two input ports respectively while calculating the ratio of output pulses after the first iteration and the initial preparation (See Supplementary 5.2). Assume the initial weight matrix is $\begin{bmatrix} m_{11} & m_{12} \\ m_{21} & m_{22} \end{bmatrix}$ and the input pulse is $\begin{bmatrix} E_i \\ 0 \end{bmatrix}$. The first and second photo-detected signals are $\begin{bmatrix} E_i \\ 0 \end{bmatrix}$ and $\begin{bmatrix} (1 + m_{11})E_i \\ m_{21}E_i \end{bmatrix}$ respectively. The ratio between the output pulses after the first iteration and the initial preparation corresponds to the first column of the weight matrix. The amplitude and phase of a complex-valued number correspond to the amplitude and phase of an MZI unit, which are determined by the “differential” and “common-mode” phase shifts of two MZI arms and can be adjusted accordingly. Similarly, the second column of the weight matrix can be characterised by launching $\begin{bmatrix} 0 \\ E_i \end{bmatrix}$ via configuring the state of the input switch.

8. Complex-valued matrix inversion: could you please give real specific examples where this finds application? The same section is not well explored as the previous on the real-valued operations, so that an extension of the same would be needed to highlight the improvements in real applications.

Our response: Thanks for your constructive suggestions. The main application of complex-valued matrix inversion is MIMO precoding¹⁷, which is one of the most promising technologies for 5G/6G wireless communication systems featuring increased spectral capacity while attaining high energy efficiencies. The system model can be expressed as: $\mathbf{y} = \mathbf{H}\mathbf{x} + \mathbf{n}$. $\mathbf{y} \in \mathbb{C}^{N \times 1}$ is the complex-valued received vector, $\mathbf{x} \in \mathbb{C}^{M \times 1}$ is the complex-valued transmit vector, $\mathbf{H} \in \mathbb{C}^{N \times M}$ is the complex-valued channel matrix, whose elements obey complex normal distributions, $\mathcal{CN}(0,1)$, and $\mathbf{n} \in \mathbb{C}^{M \times 1}$ is the additive white Gaussian noise. Linear MIMO detection and precoding requires the computation of matrix inversions: $(\mathbf{H}^H \mathbf{H} + \lambda \mathbf{I})^{-1}$. The ever-increasing data demand has made the massive MIMO system more attractive. The large-scale matrix inversion is computationally expensive for traditional digital electronic computers. Off-loading the most computationally intensive matrix inversion to a PIP can significantly enhance the C-to-IO ratio, which is a promising way to improve the throughput of MIMO systems.

We have included more detailed introduction and motivation of the “Complex-valued matrix inversions” subsection. In addition, we have performed simulations of the processing speed (latency) improvement of using a PIP instead of a PSP in MIMO precoding tasks which is shown in Fig. 6 and Supplementary 6.

Complex-valued matrix inversions:

One of the major applications of complex-valued matrix inversion is in MIMO decoding and precoding¹⁹⁻²¹, which are fundamental to 5G/6G wireless communication systems. These algorithms require the inversion of complex-valued channel matrices, whose dimensions significantly increase due to the growing data demand from an increased number of end-users. Large scale complex-valued matrix inversion is computationally expensive and faces speed and power efficiency limitations in digital electronic computers. Unlike traditional methods that store a complex number as two separate real parts and process each part individually, photonic processors manipulate the amplitude and phase of optical signals simultaneously, enabling truly complex-valued computations with enhanced efficiency. Additionally, the PIP system can further improve the C-to-IO ratio and increase the inversion speed through its iterative optical loopback, significantly reducing IO access.

9. While a comparison among the available photonic platforms is shown in terms of scalability, it is still not clear what actually the target scaling is for each application or cluster of applications in terms of number of I/Os and PIP size and why. I believe the authors should comment on that.

Our response: Thanks for your comment. We have included two main inversion-intensive applications in the revised manuscript which are ridge regression tasks and MIMO precoding tasks. We analyse the scalability in terms of processing time in detail and present the results in Fig. 6c and 6d in the main text and Supplementary 1.3 and 6.3. The energy efficiency of the PIP is application independent.

Supplementary 1.3:

In the following, two major application cases are chosen to illustrate the C-to-IO ratio enhancement of the PIP in practice, which are ridge regression and MIMO (multiple-input and multiple-output) precoding technology. The ridge regression task is to solve $\hat{\beta} = (X^T X + \lambda I)^{-1} X^T y$, where $\hat{\beta}$ is the parameter vector to be fitted (solved), X, y are the independent and dependent variables respectively that encode the samples, I is an identity matrix and λ is the ridge parameter that can be adjusted to improve the fitting results. The MIMO precoding using the minimum mean square error (MMSE) method is: $\hat{x} = (H^H H + \lambda I)^{-1} H^H y$, where \hat{x} is the estimated

Figure S2 | Convergence iteration, P , of the PIP for different application scenarios (a) Ridge regression with an accepted error of $\epsilon = 0.2\%$. (b) Ridge regression with an accepted error of $\epsilon = 1\%$. (c) Ridge regression with an accepted error of $\epsilon = 5\%$. (d) MIMO precoding with an accepted error of $\epsilon = 0.2\%$. (e) MIMO precoding with an accepted error of $\epsilon = 1\%$. (f) MIMO precoding with an accepted error of $\epsilon = 5\%$. In each figure, the convergence iterations, P , for reaching the accepted error level ϵ are simulated and fitted for different matrix sizes and λ . λ is the ridge parameter in a ridge regression task and the ratio between number of transmit antennas and SNRs in a MIMO precoding using the minimum mean square

transmitted signal, \mathbf{H} is the channel matrix, \mathbf{y} is the received signal, and $\lambda = \frac{N_r}{SNR}$ is ratio between the number of transmit antennas, $N_r = N$, and the signal-to-noise ratio, SNR.

The convergence iterations, P , for calculating the matrix inversions in these two applications are shown in Fig. S2 for different accepted error levels of $\varepsilon = 0.2\%$, 1% , and 5% and different parameters, $\lambda = 0.111$, 0.222 , and 0.333 which corresponds to different noise levels. $\lambda = 1$ represents an extreme case where the convergence is the fastest but the “noise level” is the same as the useful “signal level” which shall not happen in practice. For the ridge regression task, the convergence iteration, P and matrix size, N follows a linear relationship and P is apparently much larger than N . For the MIMO precoding task, P is larger than N for almost all error levels and SNRs shown in the figure. These results indicate that our PIP has a much higher (at least N times higher for an $N \times N$ inversion) C-to-IO ratio than PSP and digital computers in these two applications. A mathematical proof is provided below.

$$\text{C-to-IO (PIP)} / \text{C-to-IO (CPU)} = \frac{N^3 P}{2N^2 + N} \div \frac{N}{3} = \frac{3NP}{2N+1} \geq N, \text{ for } P \geq N.$$

$$\text{C-to-IO (PIP)} / \text{C-to-IO (TPU/PSP)} = \frac{N^3 P}{2N^2 + N} \div \frac{NP}{2P+1} = \frac{N(2P+1)}{2N+1} \geq N, \text{ for } P \geq N.$$

Supplementary 6.3:

6.3 Processing time (Latency) estimation of an $N \times N$ PIP

The processing time of the proposed PIP consists of two parts: (1) Core processing time and (2) IO access time. The two parts are discussed below.

1) Core processing time

The core processing time refers to the duration starting from when a signal is launched into the processor until the computation results are ready for acquisition. It is determined by the single iteration’s processing time of the optical signals in the loop and the number of iterations required for the Richardson method to converge to the desired accuracy. The single iteration’s processing time is calculated by $t_{loop} = L_{loop} \cdot n_{eff} / c$. L_{loop} is the estimated loop length of an $N \times N$ PIP as shown in Fig. S11. n_{eff} is the effective index of the waveguide on three platforms, which are 2.43, 1.98, and 2.16 for SOI, SiN, and IMOS platforms respectively. $c = 3 \times 10^8 \text{m/s}$ is the speed of light in vacuum. The single iteration’s processing time of an $N \times N$ PIP, t_{loop} , is shown in Fig. 6c in the main text, ranging from several tens of picoseconds for a 2×2 PIP to ~ 1 ns for a 256×256 PIP. The number of iterations for convergence under an error level ε is: $P = \left\lceil \frac{\ln(1/\varepsilon)}{\ln|(\lambda_N + \lambda_1)/(\lambda_N - \lambda_1)|} \right\rceil$ as derived in Supplementary 1.2. P is application dependent and is usually larger than N . The PIP architecture shown in Fig. 1 in this paper generates the matrix inversion results one column a time, corresponding to a core processing time of $t_{core} = t_{loop} \cdot P \cdot N$. Using wavelength multiplexing techniques³ can reduce the core processing time to $t_{core_wdm} = t_{loop} \cdot P$.

2) IO access time

The IO traffic of the PIP system for inverting an $N \times N$ matrix includes loading weight matrix into the MZI weight bank, launching input data to the modulators, and reading out the inversion results. As shown in Table 1 in the main text, the total memory access counts for the PIP to invert an $N \times N$ matrix is $2N^2 + N$, which includes N^2 access counts for loading the weight matrix, N access counts for launching the input data, and N^2 access counts for reading out the inversion results.

Loading the weight matrix is a one-time process that consumes time scaling with N^2 . After characterisation (time for characterisation is neglected since it is a one-time process), N^2 matrix elements are loaded into the on-

chip weight bank for an $N \times N$ processor by applying voltages to the phase shifters. A typical way to apply one voltage is to send a data from a desktop computer to a digital-to-analogue converter (DAC) via SPI communication protocols, with a highest data transmission rate of 60 Mbps. For each voltage applied on the heater, the voltage is encoded with 16-bits, and the SPI communication used to set the voltages has a data transmission rate of 60 Mbps, which is far more time consuming than the core processing time. In an optimised electronic-photonic co-integrated design, the transmission rate between the local buffer and the DAC can reach up to 180 Gb/s⁵. Even at this high data movement speed, the weight matrix loading time remains a heavy burden compared to the core processing time. For a 256×256 processor, the time for loading the weight matrix is $t_{load} = \frac{N^2 \cdot B(\text{bit})}{\text{Data Rate (Gbit/s)}} = \frac{N^2 \cdot 16\text{bit}}{180\text{Gbit/s}} = 5825.4 \text{ ns}$, which is 5000 times more than the single iteration's processing time. For the ridge regression task and the MIMO precoding task shown in Fig. S2, the weight matrix loading time is at least 2.7 times and 1.3 times more than the matrix inversion time. Though one could argue that utilizing parallel data transfer paths might alleviate the IO bottleneck, this approach is achieved at the expense of higher energy consumption and increased complexity in hardware routing and control. After the voltage is applied, it takes certain time for the phase shifters to reach its stable state. For TO phase shifters, the stabilization (heating and cooling) time is at least $5 \mu\text{s}$ ²⁰, while for electro-optic (EO) phase shifters, the stabilization time is only around 20 ps²¹, which is negligible compared to the core processing time and the data movement time.

The complete $N \times N$ inverse matrix is obtained by launching N input unit vectors to the modulators, which requires to access the memory N times. Similarly, the time for inputting data is calculated as $t_{input} = \frac{N \cdot B(\text{bit})}{\text{Data Rate (Gbit/s)}}$. To read out the inversion results, which is an $N \times N$ matrix, a total of N^2 access counts is needed, corresponding to an outputting time of $t_{output} = \frac{N^2 \cdot B(\text{bit})}{\text{Data Rate (Gbit/s)}}$. The total IO access time of our PIP is thus estimated to be $t_{IO_PIP} = \frac{(2N^2 + N) \cdot B(\text{bit})}{\text{Data Rate (Gbit/s)}}$. Table S4 lists the worst-case core processing time (t_{loop}) among three integration platforms, minimal/maximal iteration numbers shown in Fig. S2 ($P_{Ridge}(\text{min}/\text{max})$, $P_{MIMO}(\text{min}/\text{max})$), minimal/maximal inversion time ($t_{core_Ridge}(\text{min}/\text{max})$, $t_{core_MIMO}(\text{min}/\text{max})$) and IO access time (t_{IO_PIP}) of an $N \times N$ PIP.

TABLE S4
PROCESSING TIME DECOMPOSITION OF AN $N \times N$ PIP FOR MATRIX INVERSION

	t_{loop} (ns)	$P_{Ridge}(\text{min}/\text{max})$	$P_{MIMO}(\text{min}/\text{max})$	$t_{core_Ridge}(\text{min}/\text{max})$ (ns)	$t_{core_MIMO}(\text{min}/\text{max})$ (ns)	t_{IO_PIP} (ns)
2×2	0.07	5/17	18/72	0.7/2.4	2.5/10.1	0.89
4×4	0.08	7/34	54/310	2.3/11.0	17.4/100	3.2
8×8	0.12	12/64	104/899	11.5/61.2	99.4/859.2	12.09
16×16	0.15	21/120	144/1800	50.2/286.8	344.2/4302.3	46.93
32×32	0.21	39/233	167/2809	258.8/1546.4	1108.3/1864.3	184.89
64×64	0.32	75/457	178/3419	1543.6/9405.9	3663.6/70369	733.87
128×128	0.55	147/906	183/3690	10314/63567	12840/258899	2924.1
256×256	1.01	292/1804	187/3808	75186/464510	48150/980510	11673.6

In addition, Table S5 lists the IO access time of an $N \times N$ PSP ($t_{IO_PSP} = \frac{N^2(2P+1) \cdot B(\text{bit})}{\text{Data Rate (Gbit/s)}}$) for both tasks and the total processing time of inverting an $N \times N$ matrix on an $N \times N$ PIP and an $N \times N$ PSP. $B(\text{bit}) = 16(\text{bit})$, $\text{Data Rate (Gbit/s)} = 180\text{Gb/s}$. As shown in Table S5, the latency improvement of the PIP in implementing matrix inversions is increasingly more significant as the processor size increases. Note that the IO access time remains the same whether wavelength multiplexing techniques are used.

TABLE S5

PROCESSING TIME COMPARISON BETWEEN AN $N \times N$ PIP AND AN $N \times N$ PSP FOR MATRIX INVERSION

	$t_{IO_PSP_Ridge}$ (min/max) (μs)	$t_{IO_PSP_MIMO}$ (min/max) (μs)	$t_{tot_PIP_Ridge}$ (min/max) (μs)	$t_{tot_PIP_MIMO}$ (min/max) (μs)	$t_{tot_PSP_Ridge}$ (min/max) (μs)	$t_{tot_PSP_MIMO}$ (min/max) (μs)
2x2	0.004/0.01	0.01/0.05	0.002/0.003	0.003/0.011	0.005/0.015	0.016/0.062
4x4	0.02/0.10	0.16/0.88	0.006/0.014	0.021/0.10	0.024/0.11	0.17/0.98
8x8	0.14/0.73	1.2/10.2	0.024/0.073	0.11/0.87	0.15/0.80	1.3/11.1
16x16	0.98/5.5	6.6/81.9	0.097/0.33	0.39/4.3	1.0/5.8	6.9/86.2
32x32	7.2/42.5	30.5/511.5	0.44/1.7	1.3/18.8	7.4/44.1	31.6/530.1
64x64	55.0/333.1	130.0/2490	2.3/10.1	4.4/71.1	56.5/342.5	133.6/2560.4
128x128	429.6/2640.4	534.5/10749	13.2/66.5	15.8/261.8	439.9/2703.9	547.3/11008
256x256	3407.9/21024	2184.5/44372	86.9/476.2	59.8/992.2	3483.1/21488	2232.7/45353

Figure S12 | Latency improvement of an $N \times N$ lossless PIP over an $N \times N$ PSP in ridge regression and MIMO precoding tasks. (a) Latency improvement in inversion operations in the ridge regression task. (b) Latency improvement in inversion operations in the MIMO precoding task. (c) Total latency improvement of the PIP in the ridge regression task. (d) Total latency improvement in the MIMO precoding task. “Min” and “Max” corresponds to the minimal and maximal processing time as shown in Table S4 and S5. “WDM” represents the latency improvement when wavelength multiplexing techniques are used. Points above the “Ref” line indicate where the PIP has shorter latency than the PSP while points below the “Ref” line indicate where PIP has longer latency than the PSP. The vertical axis is the logarithm of the ratio between the total processing time of the PSP and the PIP to the base of 2.

Fig. S12 a-b showcase the latency improvement of an $N \times N$ PIP over an $N \times N$ PSP (t_{tot_PSP}/t_{tot_PIP}) for implementing matrix inversions in ridge regression and MIMO precoding tasks in a more intuitive way. Fig. S12 c-d show the total latency improvement considering the time for implementing other matrix operations including one matrix-matrix addition, two matrix-matrix multiplications and one matrix-vector multiplication. Each operation can be implemented in a single pass on both the PIP and the PSP. The latency improvement for the complete task is worse than that of implementing matrix inversion solely since matrix multiplication and matrix addition have much lower C-to-IO ratio than matrix inversion. Still, up to >40 times latency improvement of the PIP over PSP is estimated for 256×256 ridge regression task and MIMO precoding task respectively thanks to the much higher C-to-IO ratio and much less reduced IO access time of the PIP. The latency improvement of the PIP when wavelength multiplexing techniques are used is also exhibited in Fig. S12, showing up to >350 and >710 times improvement for 256×256 ridge regression task and MIMO precoding task respectively. Note that we have shown in Table 1 in the main text that the state-of-the-art electronic processor has the same C-to-IO ratio as the PSP, which indicates huge potential of our proposed PIP in improving the latency of matrix-inversion-intensive tasks, especially when the problem size is large.

Minor points:

- Page 2: matric -> matrix

Our response: Thanks for pointing out our typo. In order to better introduce the iterative matrix inversion and the advantage of our PIP, we have rewritten the sentences in the introduction as:

In most practical applications, the matrices to be inverted are diagonally dominant or sparse, and approximate inversion results are often sufficient for subsequent processing stages.

- “Encoding one element”: is really what the authors mean, or they intend “encoding a column” or a vector?

Our response: Yes. The $N \times N$ weight matrix is encoded in a set of $N \times N$ MZI units. As an optical signal propagates through the MZI unit, its amplitude and phase are changed. Thus, each MZI unit encodes one element of the $N \times N$ matrix.

References

1. McMahon, P. L. The physics of optical computing. *Nat Rev Phys* 1–18 (2023).
2. Sipser, M. Introduction to the Theory of Computation. *SIGACT News* **27**, 27–29 (1996).
3. Kung, H. T. Why systolic architectures? *MC* **15**, 37–46 (1982).
4. Jouppi, N. P. *et al.* In-Datacenter Performance Analysis of a Tensor Processing Unit. in *Proceedings of the 44th Annual International Symposium on Computer Architecture* 1–12 (Association for Computing Machinery, New York, NY, USA, 2017).
5. Alexoudi, T. *et al.* Optics in Computing: From Photonic Network-on-Chip to Chip-to-Chip Interconnects and Disintegrated Architectures. *J. Lightwave Technol.* **37**, 363–379 (2019).
6. Meng, X. *et al.* Compact optical convolution processing unit based on multimode interference. *Nat Commun* **14**, 3000 (2023).
7. Zhu, H. H. *et al.* Space-efficient optical computing with an integrated chip diffractive neural network. *Nat Commun* **13**, 1044 (2022).
8. Wu, K., Soci, C., Shum, P. P. & Zheludev, N. I. Computing matrix inversion with optical networks. *Opt. Express, OE* **22**, 295–304 (2014).
9. Nahmias, M. A. *et al.* Photonic Multiply-Accumulate Operations for Neural Networks. *IEEE JOURNAL OF SELECTED TOPICS IN QUANTUM ELECTRONICS* **26**, (2020).

10. An in-depth look at Google's first Tensor Processing Unit (TPU). *Google Cloud Blog* <https://cloud.google.com/blog/products/ai-machine-learning/an-in-depth-look-at-googles-first-tensor-processing-unit-tpu>.
11. Zhou, H. *et al.* Photonic matrix multiplication lights up photonic accelerator and beyond. *Light Sci Appl* **11**, 30 (2022).
12. Feng, H. *et al.* Integrated lithium niobate microwave photonic processing engine. *Nature* 1–8 (2024).
13. Ashtiani, F., Geers, A. J. & Aflatouni, F. An on-chip photonic deep neural network for image classification. *Nature* **606**, 501–506 (2022).
14. Villalobos, P. Trading Off Compute in Training and Inference. *Epoch* <https://epochai.org/blog/trading-off-compute-in-training-and-inference> (2023).
15. Gauthier, D. J., Bollt, E., Griffith, A. & Barbosa, W. A. S. Next generation reservoir computing. *Nat Commun* **12**, 5564 (2021).
16. Suzuki, K. *et al.* Ultra-high-extinction-ratio 2×2 silicon optical switch with variable splitter. *Opt. Express*, *OE* **23**, 9086–9092 (2015).
17. Rosário, F., Monteiro, F. A. & Rodrigues, A. Fast Matrix Inversion Updates for Massive MIMO Detection and Precoding. *IEEE Signal Processing Letters* **23**, 75–79 (2016).

REVIEWER COMMENTS

Reviewer #1 (Remarks to the Author):

The author has made significant changes and revisions in response to my earlier questions and suggestions, demonstrating considerable effort and improvements. However, my main concern is about the scale of their photonic iterative processor (PIP). A 4-mode real-valued chip (or 2-mode complex-valued chip) is not sufficient to effectively showcase the strengths and significance of their method and to solve complex problems, especially considering the state-of-the-art SiN chip has 20 modes. Although I understand that different from these single-pass chips, they conceived the idea of photonic loops, which is the highlight of the paper, the on-chip demonstration that includes only a 2-mode complex-valued matrix is not impressive. If some improvements can be made to strengthen the chip itself, either from the chip scale or integrated device design perspective, it would elevate the quality of this work significantly. I would be glad to see the manuscript after those changes.

There are some other comments:

1. My previous concerns regarding Figure 1 remain unresolved. The chips have not yet integrated all components as shown in Figure 1 and cannot be considered a fully integrated system, as the authors mentioned, "we were not able to integrate all components on-chip". I understand that it is just a conceptual figure, however, to avoid being misleading and giving the impression of exaggeration, I would suggest, at least, marking every part to indicate whether they are integrated or not in this paper. This would provide clear guidance for future developments in this area, which would be meaningful for benchmarking the progress of achievements in photonic iterative techniques.
2. I appreciate the author's efforts to clarify and expand upon the concept of the computation-to-I/O (C-to-IO) ratio, and I agree that it can serve as a criterion for performance evaluation. In the newly added Table R1, the PIP structure does have an advantage over other models, but the advantage appears marginal when the values of N or P are small. That raises my concern about the demonstration not being impressive enough. I understand that accessing a large chip size may not be easy or immediate. As I mentioned before, I suggest providing some evidence to strengthen the chip itself, either from the chip scale or integrated device design perspective. Besides, can the author provide some discussion on the scalability of their processor, and when N and P are at what values, what types and scales of tasks can they be applicable to, and demonstrate a distinct advantage?
3. Since the demonstration is not a fully integrated processor, without increasing chip size and the integration of key components, the proposed advantages and the capacity to handle complex tasks like MNIST classification and normal dimension MIMO precoding are likely to remain just theoretical. What are the main hurdles or challenges in achieving full integration, and what possible techniques can be employed to address them? Can the author provide further elaboration on this topic? (more detailed than the current discussion part)
4. When operating without amplification, how many reuses of the chip can be achieved by directly reconnecting output ports to input ports?
5. The authors have added analyses regarding the speed advantage and energy consumption of the photonic integrated processor (PIP). I see the power consumption of the current system, and what will be the power consumption after full integration?
5. A minor point: Underneath Figure 5 (or anywhere else similar), I believe it is unnecessary to include detailed values of the matrices in the main text.

Reviewer #2 (Remarks to the Author):

The authors combine the advantage of executing operations in a fully-optical domain and iterative matrix inversion to empower the computation-to-IO ratio. This is a newly implemented concept and seems to find application in many domains.

The authors have extensively answered to my comments and doubts in a proper way, so I am satisfied with the answers.

Dear reviewers,

We appreciate your precious time and thank you again for the very detailed and constructive comments. Based on these comments, we have made significant revisions to the main text and the supplementary materials to address your concerns. In the following, please find our point-by-point replies to the comments. Note that we use **blue texts** for our responses to your comments, **red texts** when quoting texts from the manuscript, and **highlighted red texts** to identify new or modified texts in the manuscript. We also attach the revised manuscript with **highlighted red texts** that identify the changes. We hope we have addressed your questions and concerns.

Reviewer #1 (Remarks to the Author)

The author has made significant changes and revisions in response to my earlier questions and suggestions, demonstrating considerable effort and improvements. However, my main concern is about the scale of their photonic iterative processor (PIP). A 4-mode real-valued chip (or 2-mode complex-valued chip) is not sufficient to effectively showcase the strengths and significance of their method and to solve complex problems, especially considering the state-of-the-art SiN chip has 20 modes. Although I understand that different from these single-pass chips, they conceived the idea of photonic loops, which is the highlight of the paper, the on-chip demonstration that includes only a 2-mode complex-valued matrix is not impressive. If some improvements can be made to strengthen the chip itself, either from the chip scale or integrated device design perspective, it would elevate the quality of this work significantly. I would be glad to see the manuscript after those changes.

Our response: Thank you for your valuable comments on improving the quality of our manuscript. We apologise for not being able to verify the PIP technology on a larger-sized chip or on a chip with more integrated components, however, we believe that our current experimental demonstrations, simulation results and analyses sufficiently outline the advantages of the PIP structure in enhancing the IO efficiency for matrix-inversion-intensive tasks. Also, it is a reasonable approach to demonstrate a novel concept on a relatively small-scale device as a proof-of-concept. For example, the recent two papers on optical computing published in Nature Photonics: Vahid et al. experimentally demonstrate a 2×2 and a 3×3 vector-matrix multiplier on a silicon photonics platform using a novel 2D inverse-design method¹, and SeyedMohammad et al. experimentally demonstrate the process of finding the optimal communication channels of arbitrary optical systems using 2-mode (2×2) photonic integrated MZI meshes².

Although we have included detailed scalability analyses of the PIP's performance (including loop gain, loop length, processing time, inversion accuracy, and energy efficiency) for a PIP size ranging from 2×2 to 256×256 in previous manuscripts and the response letter, we apologise for not making the logic behind all the experiments, simulations, analyses and advantages of the PIP structure clear to readers. We have now reorganised the contents of our manuscripts, and included more analyses, simulations and discussions to strengthen our argument and further highlight the benefits of the PIP structure.

Please find our point-by-point responses to your other comments below.

There are some other comments:

1. My previous concerns regarding Figure 1 remain unresolved. The chips have not yet integrated all components as shown in Figure 1 and cannot be considered a fully integrated system, as the authors mentioned, "we were not able to integrate all components on-chip". I understand that it is just a conceptual figure, however, to avoid being misleading and giving the impression of exaggeration, I would suggest, at least, marking every part to indicate whether they are integrated or not in this paper. This would provide clear guidance for future developments in this area, which would be meaningful for benchmarking the progress of achievements in photonic iterative techniques.

Our response:

Thanks for your helpful suggestions. We apologise if we gave readers the impression of exaggerating our work. As you kindly pointed out, Figure 1 was intended to show the ultimate goal of the PIP structure, serving as guidance for future development. We have now added a few sentences to the caption of Figure 1 to clarify which components we have integrated on the 4×4 lossless PIP and the 2×2 coherent PIP. Please find the revised Figure 1 below:

Figure 1 | Conceptual figure of the photonic iterative processor (PIP). Architecture of the proposed iterative photonic processor. The PIP serves as a photonic accelerator for inverting matrices which is widely used in equation solving, communication systems, robotics trajectory control, etc. Matrix inversion is solved by the Richardson method, whose results can be obtained by multiple iterations of the light signal in the PIP. Matrix addition, subtraction, and multiplication results can also be computed by a single iteration of the light signal in the PIP. For the lossless PIP system, the 4×4 reconfigurable weighting bank, adders and splitters are integrated on-chip. For the coherent PIP system, the 2×2 reconfigurable weighting bank, adders, splitters, optical switches, coherent detectors, and optical loopback are integrated on-chip. Monolithic integrations are possible and separately discussed in the section titled "Photonic integration techniques for a fully integrated PIP".

2. I appreciate the author’s efforts to clarify and expand upon the concept of the computation-to-I/O (C-to-IO) ratio, and I agree that it can serve as a criterion for performance evaluation. In the newly added Table R1, the PIP structure does have an advantage over other models, but the advantage appears marginal when the values of N or P are small. That raises my concern about the demonstration not being impressive enough. I understand that accessing a large chip size may not be easy or immediate. As I mentioned before, I suggest providing some evidence to strengthen the chip itself, either from the chip scale or integrated device design perspective. Besides, can the author provide some discussion on the scalability of their processor, and when N and P are at what values, what types and scales of tasks can they be applicable to, and demonstrate a distinct advantage?

Our response: Thanks for acknowledging the validity of the C-to-IO ratio concept in characterising the IO efficiency of a processor. In order to enhance the clarity on the benefits of the PIP, we summarise the experimentally demonstrated and simulated IO advantages in Table R1 and R2 respectively (the formulas for computing the saved processing time, T_{save} , saved energy consumption, E_{save} , total processing time improvement ratio, $\frac{t_{total_PSP}}{t_{total_PIP}}$, and improvement in C-to-IO ratio, $\frac{C_to_IO_{PIP}}{C_to_IO_{PSP}}$ of the PIP are listed in Eq. (R1) – Eq. (R4). More details are now included in Supplementary 6).

TABLE R1
ACHIEVED IO ADVANTAGES IN THE EXPERIMENTS

	Parameters	T_{save} (μs)	E_{save} (nJ)	$\frac{t_{total_PSP}}{t_{total_PIP}}$	$\frac{C_to_IO_{PIP}}{C_to_IO_{PSP}}$
A_1^{-1}	$q = 0, N = 4, t_{loop} = 130 \text{ ns}, P = 5$	93.3	2.2	4.5	4.9
A_2^{-1}	$q = 0, N = 4, t_{loop} = 130 \text{ ns}, P = 8$	157.3	3.8	6.6	7.6
IE	$q = 1, N = 4, t_{loop} = 130 \text{ ns}, P_1 = 6, P_2 = 6$	250.7	6.0	1.7	1.75
ODE	$q = 1, N = 4, t_{loop} = 130 \text{ ns}, P_1 = 5, P_2 = 8$	272.0	6.5	1.8	1.82
PDE	$q = 2, N = 4, t_{loop} = 130 \text{ ns}, P_1 = P_3 = 9, P_2 = P_4 = 8$	736.0	17.7	1.2	1.2
A_3^{-1}	$q = 0, N = 2, t_{loop} = 300 \text{ ps}, P = 3$	12.0	0.29	2.8	2.8
A_4^{-1}	$q = 0, N = 2, t_{loop} = 300 \text{ ps}, P = 2$	6.7	0.16	2.0	2.0
A_5^{-1}	$q = 0, N = 2, t_{loop} = 300 \text{ ps}, P = 2$	6.7	0.16	2.0	2.0

TABLE R2
PREDICTED IO ADVANTAGES OF TWO PRACTICAL TASKS

	Parameters	T_{save}	E_{save}	$\frac{t_{total_PSP}}{t_{total_PIP}}$
MIMO	$q = 0, N = 4, t_{loop} = 83.8 \text{ ps}, P = 54, N_{samp} = 10^3 - 10^6 \text{ per day}$	0.15 ms-25 min per day	27.3 μJ-273 J per day	8.1 (43.9)
MIMO	$q = 0, N = 8, t_{loop} = 124.7 \text{ ps}, P = 104, N_{samp} = 10^3 - 10^6 \text{ per day}$	1.18 ms-3.3h per day	211.8 μJ-2.1 kJ per day	11.2 (90.9)
massive MIMO	$q = 0, N = 128, t_{loop} = 616.8 \text{ ps}, P = 183, N_{samp} = 10^3 - 10^6 \text{ per day}$	0.53 s - 1472 h per day	95.7 mJ-957 kJ per day	31.6 (178)
massive MIMO	$q = 0, N = 256, t_{loop} = 1.1 \text{ ns}, P = 186, N_{samp} = 10^3 - 10^6 \text{ per day}$	2.16 s - 6000 h per day	0.39 J-3.9 MJ per day	33.9 (181.6)
MNIST training	$q = 0, N = 10, t_{loop} = 94.3 \text{ ps}, P = 14, N_{samp} = 6000, N_{epoch} = 10$	14.4 ms	2.6 mJ	8.5 (13.2)

$$T_{save} = [2N^2(\sum_{i=1}^q P_i - 1) + 2^q N(N - 1)] \cdot T_0 \quad (R1)$$

$$E_{save} = [2N^2(\sum_{i=1}^q P_i - 1) + 2^q N(N - 1)] \cdot E_0 \quad (R2)$$

$$\frac{t_{total_PSP}}{t_{total_PIP}} = \frac{(1.5 \cdot 8^q - 2 \cdot 4^q + 0.5 \cdot 2^q) \cdot t_{loop} \cdot N + t_{loop} \cdot (\sum_{i=1}^{2^q} P_q) \cdot N + (2N^2 \sum_{i=1}^{2^q} P_q) + 2^q N^2 + 6(8^q - 4^q)N^2 \cdot T_0}{(1.5 \cdot 8^q - 2 \cdot 4^q + 0.5 \cdot 2^q) \cdot t_{loop} \cdot N + t_{loop} \cdot (\sum_{i=1}^{2^q} P_q) \cdot N + (2^q \cdot (2N^2 + N) + 6(8^q - 4^q)N^2) \cdot T_0} \quad (R3)$$

$$\frac{C_{to_IO_PIP}}{C_{to_IO_PIP}} = \frac{2N^2 \sum_{i=1}^{2^q} P_q + 2^q N^2 + 6(8^q - 4^q)N^2}{2^q \cdot (2N^2 + N) + 6(8^q - 4^q)N^2} \quad (R4)$$

q is the number of times a matrix is decomposed before computing on a PIP, N is the PIP size, t_{loop} is the single-iteration's processing time, and P is the number of iterations for convergence. T_0 and E_0 are the processing time and energy consumption of a single IO access.

As illustrated in Table R1, we have experimentally demonstrated >7 times improvement in the C-to-IO ratio using the 4×4 real-valued chip and >2 times improvement using the 2×2 complex-valued chip (equivalent to a 4×4 real-valued chip in terms of the amount of implemented MAC operations and IO access). For the equation solving experiments, even when block matrix inversion methods are used to solve 8×8 and 16×16 matrix inversions on a 4×4 real-valued chip, we still demonstrate >1.8 times IO efficiency improvement. **These numbers reveal that the IO advantage demonstrated in our experiments is not marginal even when N or P are small.**

To predict the future development of this technology and the scalability of the PIP, we estimate the loop length and t_{loop} based on the best reported values (shown in the previous Supplementary 6.2-6.3). Using these values to estimate the IO advantages of matrix-inversion-intensive tasks such as MIMO precoding and reservoir training of deep learning datasets clearly indicates the future development of the PIP and its potential in breaking the IO bottlenecks in practical applications. For the above mentioned two examples, the main model is implemented on electronic processors, while the matrix inversion part is offloaded to a PIP to save processing time and power consumption. For a normal MIMO system or the current massive MIMO system, the size of the channel matrix is typically 4×4 and up to 8×8 . For the future massive MIMO system, the channel matrix will be scaled to $128 \times 128^{3,4}$ or even larger. According to Table R2, the processing time improvement ratio $\frac{t_{total_PSP}}{t_{total_PIP}}$ is 8.1, 11.2, 31.6, and 33.9 for 4×4 , 8×8 , 128×128 , and 256×256 PIPs, indicating an order-of-magnitude improvement of the PIP. In practice, the MIMO channel matrices are time-varying, which means continuous matrix inversions are required. The updating period of a MIMO channel ranges from several minutes in a quasi-static environment to several microseconds in high-speed scenarios. This means $\sim 10^3$ to $\sim 10^{10}$ matrices need to be inverted per day, resulting in substantial amount of saved processing time and power consumption by using the PIP. For the training of the deep learning datasets using reservoir computing, the processing time improvement ratio $\frac{t_{total_PSP}}{t_{total_PIP}}$ is 8.5. Similarly, the total saved processing time and energy consumption becomes significant since a total of 60000 matrices needs to be inverted for the training task. The IO advantages of the PIP are more significant for datasets with more features. For both examples analysed above, wavelength multiplexing techniques can be used to further enhance the IO advantages, leading to up to two orders of magnitude improvement in IO efficiency for MIMO precoding tasks and at least an order of magnitude improvement for the MNIST training task. Scalability of the PIP is also strengthened as a separate part in the "Discussion" section.

Please find our revisions to the manuscript below.

(1) We emphasise in the abstract that we have demonstrated notable IO advantages of the PIP in the experiments. And we predict the PIP's huge potential in matrix-inversion-intensive applications through simulations.

Abstract:

Photonic integrated circuits have been extensively explored for optical processing with the aim of breaking the speed bottleneck of digital electronics. However, the input/output (IO) bottleneck remains one of the key barriers. Here we report a novel photonic iterative processor (PIP) for matrix-inversion-intensive applications. The direct reuse of inputted data in the optical domain unlocks the potential to break the IO bottleneck. We demonstrate notable IO advantages with a lossless PIP for real-valued matrix inversion and integral-differential equation solving, as well as a coherent PIP with optical loops integrated on-chip, enabling complex-valued computation and a net inversion time of 1.2 ns. Furthermore, we estimate at least an order of magnitude enhancement in IO efficiency of a PIP over photonic single-pass processors and the state-of-the-art electronic processors for reservoir training tasks and MIMO precoding tasks, indicating the huge potential of PIP technology in practical applications.

(2) We add additional information about the IO advantages of the PIP in the last paragraph of the introduction.

Introduction:

We demonstrate, to the best of our knowledge, the first lossless reconfigurable PIP system that is capable of directly inverting matrices and solving integral and differential equations. Such a PIP system computes 4×4 real-valued matrix inversions with an accuracy of >97%, a C-to-IO ratio improvement of up to >7 times, a core energy efficiency of 4.6 MOPS/W (mega operations per second per watt), and a net inversion time of 2.6 μs, which is solely bounded by the length of fibre-based optical loops. The lossless PIP is then reconfigured to numerically solve real-valued integral and differential equations, reaching a mean absolute error of < 0.02, and up to >1.8 times C-to-IO ratio improvement. The first coherent PIP with on-chip optical loops is also demonstrated to break the loop-length limitation on the net inversion time. The coherent PIP is demonstrated to operate 2×2 complex-valued matrix inversions with an accuracy of > 98%, a core energy efficiency of 3.6 GOPS/W, and a net inversion time of 1.2 ns. Its enhancement in C-to-IO ratio reaches up to 2.8 times. Benefiting from the much-reduced IO demand, the proposed PIP is capable of reaching at least an order of magnitude IO efficiency enhancement compared with a single-pass optical processor and the state-of-the-art electronic processors, by emulating MIMO precoding tasks and reservoir training tasks. Our results indicate a promising way towards ever-powerful optical processors that could surpass IO limits.

(3) We add a subsection: “Demonstrated IO advantages of the PIPs” in the “Result” section of the main text to show the demonstrated improvement in IO efficiency of the lossless PIP and the coherent PIP.

IO advantages of the PIPs

We use saved processing time, T_{save} , saved energy consumption, E_{save} , improvement in total processing time, $\frac{t_{total_PSP}}{t_{total_PIP}}$, and improvement in C-to-IO ratio, $\frac{C_to_IO_PIP}{C_to_IO_PSP}$ to quantify the demonstrated IO advantages of the lossless PIP and the coherent PIP. The results are shown in Table 2. q is the number of times a matrix is decomposed before computing on a PIP, N is the PIP size, t_{loop} is the single-iteration's processing time, and P

is the number of iterations for convergence. T_0 and E_0 are the processing time and energy consumption of a single IO access. Details about calculating each specs can be found in Supplementary 6.

TABLE 2
ACHIEVED IO ADVANTAGES IN THE EXPERIMENTS

Parameters	T_{save} (μ s)	E_{save} (nJ)	$\frac{t_{total_PSP}}{t_{total_PIP}}$	$\frac{C_to_IO_{PIP}}{C_to_IO_{PSP}}$
A_1^{-1} $q = 0, N = 4, t_{loop} = 130 \text{ ns}, P = 5$	93.3	2.2	4.5	4.9
A_2^{-1} $q = 0, N = 4, t_{loop} = 130 \text{ ns}, P = 8$	157.3	3.8	6.6	7.6
IE $q = 1, N = 4, t_{loop} = 130 \text{ ns}, P_1 = 6, P_2 = 6$	250.7	6.0	1.7	1.75
ODE $q = 1, N = 4, t_{loop} = 130 \text{ ns}, P_1 = 5, P_2 = 8$	272.0	6.5	1.8	1.82
PDE $q = 2, N = 4, t_{loop} = 130 \text{ ns}, P_1 = P_3 = 9, P_2 = P_4 = 8$	736.0	17.7	1.2	1.2
A_3^{-1} $q = 0, N = 2, t_{loop} = 300 \text{ ps}, P = 3$	12.0	0.29	2.8	2.8
A_4^{-1} $q = 0, N = 2, t_{loop} = 300 \text{ ps}, P = 2$	6.7	0.16	2.0	2.0
A_5^{-1} $q = 0, N = 2, t_{loop} = 300 \text{ ps}, P = 2$	6.7	0.16	2.0	2.0

According to the table, >7 times and >2 times improvement in the C-to-IO ratio are achieved for 4×4 real-valued matrix inversions and 2×2 complex-valued matrix inversions respectively. For solving the integral equation and the ordinary differential equation, 8×8 matrix inversions are performed on the 4×4 lossless PIP through one decomposition, still resulting in ~ 1.8 times improvement in IO efficiency. For solving the partial differential equation, 16×16 matrix inversions are performed on the 4×4 lossless PIP through two decompositions yet resulting in ~ 1.2 times improvement in IO efficiency. These results well verify the IO advantages of the PIP for matrix inversions.

(4) We revise the processing time, accuracy and energy efficiency of an $N \times N$ PIP in the discussion and add a separate subsection: “Scalability of the PIP” in the “Discussion” section.

Processing time (latency) of an $N \times N$ PIP

The processing time of a PIP includes both the core processing time and the IO access time. The core processing time refers to the duration starting from when a signal is launched into the processor until the computation results are ready for acquisition, which is determined by the single-iteration’s processing time, t_{loop} , and the number of iterations to reach convergence, P . The PIP system shown in Fig.1 generates the matrix inversion results one column a time, corresponding to a core processing time of $t_{core} = t_{loop} \cdot P \cdot N$. Using wavelength multiplexing techniques can reduce the core processing time to $t_{core_wdm} = t_{loop} \cdot P$. According to Table 2, the best demonstrated net inversion time, t_{core} , are 2.6 μ s and 1.2 ns for the 4×4 and 2×2 cases, respectively. The loop length of the lossless PIP is mainly limited by the long fibre length (~ 35 m) inside the EDFA, which can be removed by using semiconductor optical amplifiers (SOA) or on-chip gain components. It is worth noting that for the coherent PIP device, additional delay lines (~ 3.9 cm) are integrated as part of the loopback paths to ease the electrical readout, which can be shortened for a faster computation speed. To explore the future scaling of the PIP, we estimate the loop length and the single iteration’s processing time of a fully integrated PIP with sizes ranging from 2×2 to 256×256 as shown in Fig. 6a-b. Both the loop length and the single iteration’s processing time scale approximately linearly with PIP size.

The IO access time is defined as $t_{IO} = \frac{M \cdot B}{DR}$, where M is memory access counts shown in Table 1, B is the bit resolution of the data, and DR is the data transfer rate. t_{IO} includes the weight matrix loading time, input data fetching time, and output data storing time, and contributes to a heavy burden for digital electronic processors

and single-pass photonic processors, while it contributes much less to the total inversion time for the PIP. The total processing (inversion) time of a PIP and a PSP can be estimated according to $t_{total_PIP} = t_{core_PIP} + t_{IO_PIP}$, and $t_{total_PSP} = t_{core_PSP} + t_{IO_PSP}$, respectively. More details can be found in Supplementary 7.2-7.3.

Figure 6 | Performance analyses of the PIP. (a) Estimated loop length of the PIP with sizes ranging from 2×2 to 256×256. Length of each component is based on the reported best values. (b) Estimated single-iteration's processing time with sizes ranging from 2×2 to 256×256. (c) Estimated inversion accuracies of the PIP with sizes ranging from 2×2 to 256×256. (d) Estimated energy efficiency of the PIP core with sizes ranging from 2×2 to 256×256.

Energy efficiency of an $N \times N$ PIP

Energy efficiency of the PIP core is defined as: $Energy\ efficiency = \frac{Number\ of\ operations/Second}{Energy\ consumption}$, which is the number of operations per second per energy consumption of the PIP core. The components consuming energy during the matrix inversion process include lasers, modulators, semiconductor optical amplifiers (SOAs), photodetectors (PDs), the MZI weight bank, and optical switches. The energy efficiency of the lossless PIP and the coherent PIP, together with the estimated energy efficiency of the PIP (with and without wavelength multiplexing techniques) and that of the state-of-the-art electronic and photonic processors are shown in Fig. 6d, indicating that the PIP structure is an energy-efficient architecture for matrix inversion tasks. More details can be found in Supplementary 7.5.

Scalability of the PIP

In the above analyse, the PIP size is limited to 256×256 . While this is sufficient for most real applications, with no more than 5 decompositions (see Supplementary 6 for more details) ensuring the IO advantage for problem sizes up to 8192×8192 , larger-scale PIPs are still preferable for tackling even more complex problems with more significant IO advantages.

The main constraints for further scaling up the PIP are similar to those for any large-scale photonic integrated circuit (PIC), i.e., the chip insertion loss and the available reticle size of the lithography. As shown in Fig. 6c, for PIPs larger than 128×128 , good accuracy can still be achieved but requires high input signal powers, indicating that the round-loop loss is not well sustainable for higher-radix crossbar array-based MVM cores. A singular value decomposition (SVD)-based MVM core can be used to reduce signal splitting and addition losses, however, the accumulated insertion loss of the increased building blocks remains as a challenge. Figure 6c also reveals the estimated inversion accuracy of a 256×256 PIP with an SVD based MVM core, indicating good scalability. Nevertheless, the computation overhead for implementing SVD is non-negligible. Several approaches have been reported to effectively reduce the PIC insertion loss, including the use of a microelectromechanical-system (MEMS) MZI structure³⁰ and high-index doped silica glass waveguide³¹. Wafer-scale photonic integrated circuits are also demonstrated via inter-reticle waveguides³⁰, paving the way for very large-scale PIP systems.

(5) We add a subsection: “Predicted IO advantages of the PIP for matrix-inversion-intensive applications” in the “Discussion” section to showcase the huge potential of the PIP technology in real applications based on simulation results.

Envisioned IO advantage for matrix-inversion-intensive applications

We further evaluate the IO advantages of our PIP in matrix-inversion-intensive problems, i.e., the MIMO precoding task and the reservoir training task which is essentially a ridge regression task. The MIMO precoding task solves $\hat{\mathbf{x}} = (\mathbf{H}^H \mathbf{H} + \lambda \mathbf{I})^{-1} \mathbf{H}^H \mathbf{y}$, while the reservoir training task solves $\mathbf{W}_{out} = \mathbf{Y}_d \mathbb{O}_{total}^T (\mathbb{O}_{total} \mathbb{O}_{total}^T + \lambda \mathbf{I})^{-1}$. The matrices to be inverted in these applications are typically diagonally dominant, and their feasibility of being inverted on a PIP is verified in the experiments. To predict the advantages of using a PIP for matrix inversions in these two applications, we offload the matrix inversion task to a hypothetical PIP based on the previous analyses of the PIP’s processing time, accuracy, and energy consumption, and implement the main model on electronic processors. The estimated IO advantages of these two tasks are summarised in Table 3. More details about these two tasks can be found in Supplementary 7.

TABLE 3
PREDICTED IO ADVANTAGES OF TWO PRACTICAL TASKS

	Parameters	T_{save}	E_{save}	$\frac{t_{total_PSP}}{t_{total_PIP}}$
MIMO	$q = 0, N = 4, t_{loop} = 83.8 \text{ ps}, P = 54, N_{samp} = 10^3 - 10^6 \text{ per day}$	0.15 ms- 25 min per day	27.3 μJ -273 J per day	8.1 (43.9)*
MIMO	$q = 0, N = 8, t_{loop} = 124.7 \text{ ps}, P = 104, N_{samp} = 10^3 - 10^6 \text{ per day}$	1.18 ms-3.3h per day	211.8 μJ - 2.1 kJ per day	11.2 (90.9)
massive MIMO	$q = 0, N = 128, t_{loop} = 616.8 \text{ ps}, P = 183, N_{samp} = 10^3 - 10^6 \text{ per day}$	0.53 s - 1472 h per day	95.7 mJ- 957 kJ per day	31.6 (178)
massive MIMO	$q = 0, N = 256, t_{loop} = 1.1 \text{ ns}, P = 186, N_{samp} = 10^3 - 10^6 \text{ per day}$	2.16 s - 6000 h per day	0.39 J-3.9 MJ per day	33.9 (181.6)
MNIST training	$q = 0, N = 10, t_{loop} = 94.3 \text{ ps}, P = 14, N_{samp} = 6000, N_{epoch} = 10$	14.4 ms	2.6 mJ	8.5 (13.2)

* Numbers in the brackets represent the cases when wavelength multiplexing techniques are used.

For a normal MIMO system or the current massive MIMO system, the size of the channel matrix is typically 4×4 and up to 8×8 . For the future massive MIMO system, the channel matrix will be scaled to 128×128 ^{32,33} or

even larger. At least an order of magnitude IO improvement of the PIP is predicted for PIP sizes larger than 8×8 . In practice, the MIMO channel matrices are time-varying and continuous matrix inversions are required. The updating period of a MIMO channel ranges from several minutes in a quasi-static environment to several microseconds in high-speed scenarios. This means $\sim 10^3$ to $\sim 10^{10}$ matrices need to be inverted per day, resulting in substantial amount of saved processing time and power consumption by using the PIP. For the reservoir training of the deep learning datasets, 8.5 times improvement in IO efficiency is predicted for a 10×10 PIP. Similarly, the total saved processing time and energy consumption becomes significant since a total of 60000 matrices needs to be inverted for the training task. The IO advantages of the PIP are more significant for datasets with more features. For both examples, wavelength multiplexing techniques can be used to enhance the IO advantages, as indicated by the numbers in the brackets in Table 3, leading to up to two orders of magnitude improvement in IO efficiency for MIMO precoding tasks and at least an order of magnitude improvement for the MNIST training task.

3. Since the demonstration is not a fully integrated processor, without increasing chip size and the integration of key components, the proposed advantages and the capacity to handle complex tasks like MNIST classification and normal dimension MIMO precoding are likely to remain just theoretical. What are the main hurdles or challenges in achieving full integration, and what possible techniques can be employed to address them? Can the author provide further elaboration on this topic? (more detailed than the current discussion part)

Our response: Thanks for the questions and helpful suggestions. As shown in Table R1, even if the PIP is not fully integrated on a single chip and with a limited scale, we can still gain notable benefits in speed and power consumption thanks to the reduced IO access counts. Moreover, as shown in Table R2, a fully integrated PIP or a large-scale PIP makes the IO advantage more significant. Having optical gain in the loop supports a significantly increased number of iterations, making the PIP a more appealing technology in breaking the IO bottlenecks in practical matrix-inversion-intensive tasks.

Since the start of research in silicon photonics in 1987⁵, a wide range of devices including bends, splitters, crossings, couplers, ring resonators, Mach-Zehnder modulators, and photodetectors have been well developed. The integration of a light source or an amplifier on silicon still remains an active research area today, as silicon is an indirect bandgap material that precludes the possibility of a silicon laser. However, as we previously mentioned in the Discussion section, several hybrid or heterogeneous integration methods have been developed to integrate active gain components and laser sources on silicon. These include flip-chip bonding⁶, die/wafer bonding^{7,8}, micro-transfer printing⁹, direct epitaxial growths^{10,11}, optical wire bonding^{12,13}, and the membrane technology^{14,15}. Apart from that Intel has successfully commercialized III/V-on-Si transceivers via wafer-bonding¹⁶, making such cross-platform devices is still challenging given the cost, yield, and the establishment of a dedicated process flow. There are an increasing number of in-house demonstrations in the full integration of active and passive components, yet there is no fabrication runs accessible to the public. This is the key reason that whilst the hybrid/heterogeneous integration technology is becoming a more and more commonly used platform, innovations such as our PIP can not be realized immediately.

Please find our revisions to the manuscript below.

(1) We move the original subsection: “Photonic integration techniques for realising a fully integrated PIP” in the “Discussion” section to the end of the “Discussion” to make the flow better and provide more details to elaborate this topic.

Photonic integration techniques for a fully integrated PIP

According to Table 2, even if a PIP is not fully integrated on a single chip and with a limited scale, we can still gain notable benefits in speed and power consumption thanks to the reduced IO access counts. Moreover, as shown in Table 3, a fully integrated PIP as shown in Fig. 1 or a large-scale PIP makes the IO advantage more significant. Having optical gain in the loop supports a significantly increased number of iterations, making the PIP a more appealing technology in breaking the IO bottlenecks in practical matrix-inversion-intensive tasks. However, the integration of a light source or an amplifier on silicon chip still remains an active research area today, as silicon is an indirect bandgap material that precludes the possibility of a silicon laser. A number of hybrid and heterogeneous integration methods have been investigated to combine the power of III-V with the full capability of SOI. These include flip-chip bonding³⁴, die/wafer bonding^{35,36}, micro-transfer printing³⁷, direct epitaxial growth^{38,39}, optical wire bonding^{40,41}, and the membrane technology^{42,43}. There are an increasing number of in-house demonstrations in the full integration of active and passive components, and when they are released to the public in the future, a fully integrated PIP can be realised immediately.

4. When operating without amplification, how many reuses of the chip can be achieved by directly reconnecting output ports to input ports?

Our response: When operating without any optical or electrical amplification, the maximal number of iterations the chip can be reused is determined by the loop loss and the available bit resolution for data readout. Assume the responsivity of a typical photodetector is $0.6 \text{ A} \cdot \text{W}^{-1}$ (photodetectors with higher responsivity such as the avalanche photodiodes can be used to enhance the detection sensitivity and thus the available iteration numbers), the analogue-to-digital (ADC) converter has a high resolution of 16 bits, and the amplitude of the input signal is between -5V and 5V. The finest discernible voltage is $10/2^{16} = 152.6\mu\text{V}$. This corresponds to a minimal detectable optical power of $\frac{152.6\mu\text{V}}{50\Omega} \cdot \text{W}^{-1} = -23 \text{ dBm}$. If we inject the input pulse at a signal power of 19 dBm¹⁷, the total available loop loss is $19 - (-23) = 42\text{dB}$. If the loop loss in a single iteration is $L_0 \text{ dB}$, the number of times the chip can be reused is $n = \lfloor 42/L \rfloor$, where $\lfloor \cdot \rfloor$ is the floor operator.

For the PIP structure shown in Fig.1, the loop loss mainly comes from cascaded couplers for signal splitting and adding, which is $3(\log_2 N + 2) \text{ dB}$. As mentioned in Supplementary 7.4, current coupling loss between the chip and fibre array in the lossless PIP system is around 12 dB/loop, corresponding to 2 iterations for a 2×2 lossless PIP and 1 iteration for a lossless PIP with a size ranging from 4×4 to 256×256 . Without the possibility of integrating SOAs onto the PIP chip, pigtailed SOAs can still be used to compensate for optical power losses and boosts available number of iterations.

5. The authors have added analyses regarding the speed advantage and energy consumption of the photonic integrated processor (PIP). I see the power consumption of the current system, and what will be the power consumption after full integration?

Our response: We apologise for not labelling Figure 6f properly (now Figure 6d in the revised manuscript). The blue and red lines in Fig. 6d represent the predicted power consumption of a fully integrated PIP without and with wavelength multiplexing techniques, respectively. The formulas for estimating the power consumption of key components are listed in Supplementary Table S7, followed by a detailed explanation about the choices of parameters. We show the estimated energy consumption of an $N \times N$ PIP core for ridge regression and MIMO precoding tasks in Fig. S15a. Fig. S15b and Fig. 6d both show the estimated energy efficiency of the demonstrated lossless PIP and coherent PIP systems, a fully integrated PIP with and without wavelength multiplexing technique, and the state-of-the-art electronic and photonic processors. Note the energy efficiency is defined as the number of operations per second per power consumption, which is independent of the task implemented. According to Fig. 6d, a fully integrated PIP system employing the wavelength multiplexing technique is estimated to be more energy efficient than the state-of-the-art electronic and photonic processors for a PIP size ranging from 2×2 to 256×256 . Please find the revised Figure 6d below.

d Estimated energy efficiency of the PIP core

Figure 6 | Performance analyses of the PIP. (d) Estimated energy efficiency of the PIP core with sizes ranging from 2×2 to 256×256 .

6. A minor point: Underneath Figure 5 (or anywhere else similar), I believe it is unnecessary to include detailed values of the matrices in the main text.

Our response: Thanks for this suggestion. We have now put the detailed values of matrices involved in real-valued matrix inversions and complex-valued matrix inversions into Supplementary. Please find our revisions below.

Real-valued matrix inversions:

We use the set-up shown in Fig. 2a to demonstrate two real-valued matrix inversion examples (detailed matrix values, A_1 and A_2 , can be found in Supplementary 4.1 and 4.2). As shown in Fig. 3a and Fig. 3d, A_1 and A_2 are loaded into the weighting bank as $M_1 = I_4 - A_1$ and $M_2 = I_4 - A_2$ respectively (See Methods and Supplementary for matrix weights characterisation). By injecting different unit vectors $X_j^{(0)} = e_j$ ($j = 1,2,3,4$), different columns of the inversed matrices are obtained. Fig. 3b and Fig. 3e indicate a very good agreement between the ideal inverse results and the measured inverse results for the two inversion examples, respectively. The evolutions of inversion accuracy of A_1 and A_2 during convergence are traced and exhibited in Fig. 3c and Fig. 3f, reaching an inversion accuracy of 97.5% and 97.2% respectively.

Comple-valued matrix inversions:

Here we show a coherent PIP system as shown in Fig. 5a for complex-valued matrix inversions. With optical loopback paths integrated on-chip, stable phase control can be achieved together with ultrafast processing time. Optical switches are integrated on-chip to facilitate device characterisation. Coherent outcomes are read out by off-chip balanced photodetectors (BPD) and captured in the OSC. EDFAs are used to compensate for coupling loss only and BPFs are used to remove excess ASE noise. We use the coherent PIP system to demonstrate three complex-valued matrix inversion examples (detailed matrix values, A_3 , A_4 and A_5 , can be found in Supplementary 4.3 - 4.5). The measured and ideal inversion results for the three matrices including both the amplitude and the phase changes of each element of the inverse matrices are shown in Fig. 5b, 5d, and 5f respectively, indicating a good agreement with the ideal process. Figure 5c, 5e, and 5g shows the evolution of inversion accuracy of three matrices during multiple iterations. The inversion accuracy reaches 98.8%, 98.3% and 98.9% for A_3 , A_4 , and A_5 , respectively.

Supplementary:

4.3 Recorded waveforms for inverting A_3 .

The matrix to be inverted is $A_3 = \begin{bmatrix} 0.92 + 0.07i & 0 \\ 0 & 1.07 - 0.07i \end{bmatrix}$. By choosing $\omega = 1$ to simplify preprocessing, the weight matrix to be encoded is: $I_N - A_3 = \begin{bmatrix} 0.08 - 0.07i & 0 \\ 0 & -0.07 + 0.07i \end{bmatrix}$. The recorded waveforms during the inversion process are shown in Fig. S9a. z_{ij} is the element in the i^{th} row and j^{th} column of the inverse matrix. The duration of one iteration is approximately 300 ps, determined by the integrated loop length. After 3 iterations, all 2 diagonal outputs converge, indicating a net inversion time of 1.8 ns. The final inversion results are obtained by subtracting the noise from the converged pulse value. The measured inversion result, A_{meas}^{-1} , and ideal inversion result, A_{ideal}^{-1} are listed below:

$$A_{meas}^{-1} = \begin{bmatrix} 1.08 - 0.08i & 0 \\ 0 & 0.93 + 0.06i \end{bmatrix} \quad (S31)$$

$$A_{ideal}^{-1} = \begin{bmatrix} 1.09 - 0.07i & 0 \\ 0 & 0.93 + 0.07i \end{bmatrix} \quad (S32)$$

4.4 Recorded waveforms for inverting A_4 .

The matrix to be inverted is $A_4 = \begin{bmatrix} 0.98 - 0.01i & 0.07 - 0.01i \\ 0.07 - 0.05i & 1.01 - 0.12i \end{bmatrix}$. By choosing $\omega = 1$ to simplify preprocessing, the weight matrix to be encoded is: $I_N - A_4 = \begin{bmatrix} 0.02 + 0.01i & -0.07 + 0.01i \\ -0.07 + 0.05i & -0.01 + 0.12i \end{bmatrix}$. The recorded waveforms during the inversion process are shown in Fig. S9b. The duration of one iteration is approximately 300 ps, determined by the integrated loop length. After 2 iterations, all 4 outputs converge, indicating a net inversion time of 1.2 ns. The final inversion results are obtained in the same manner as described in Supplementary 4.3. The measured inversion result, A_{meas}^{-1} , and ideal inversion result, A_{ideal}^{-1} are listed below:

$$A_{meas}^{-1} = \begin{bmatrix} 1.02 + 0.00i & -0.07 + 0.01i \\ -0.07 + 0.05i & 0.99 + 0.12i \end{bmatrix} \quad (S33)$$

$$A_{ideal}^{-1} = \begin{bmatrix} 1.02 + 0.01i & -0.07 + 0.01i \\ -0.08 + 0.04i & 0.99 + 0.12i \end{bmatrix} \quad (S34)$$

4.5 Recorded waveforms for inverting A_5 .

The matrix to be inverted is $A_5 = \begin{bmatrix} 0.94 + 0.03i & -0.05 - 0.06i \\ 0.02 + 0.01i & 0.98 + 0.04i \end{bmatrix}$. By choosing $\omega = 1$ to simplify preprocessing, the weight matrix to be encoded is: $I_N - A_5 = \begin{bmatrix} 0.06 - 0.03i & 0.05 + 0.06i \\ -0.02 - 0.01i & 0.02 - 0.04i \end{bmatrix}$. The recorded waveforms during the inversion process are shown in Fig. S9c. The duration of one iteration is approximately 300 ps, determined by the integrated loop length. After 2 iterations, all 4 outputs converge, indicating a net inversion time of 1.2 ns. The final inversion results are obtained in the same manner as described in Supplementary 4.3. The measured inversion result, A_{meas}^{-1} , and ideal inversion result, A_{ideal}^{-1} are listed below:

$$A_{meas}^{-1} = \begin{bmatrix} 1.06 - 0.03i & 0.05 + 0.06i \\ -0.02 - 0.01i & 1.02 - 0.04i \end{bmatrix} \quad (S35)$$

$$A_{ideal}^{-1} = \begin{bmatrix} 1.06 - 0.04i & 0.05 + 0.06i \\ -0.02 - 0.01i & 1.02 - 0.04i \end{bmatrix} \quad (S36)$$

Figure S9 | Recorded waveforms for matrix inversions using the coherent PIP system. A 5 ns pulse is launched into the loop to obtain the computation result in each iteration. The duration of one iteration is approximately 300 ps, which is determined by the length of the loop in the experiment. (a) Inversion process of calculating A_3^{-1} . After 3 iterations, all 2 diagonal outputs converge, indicating an inversion time of 1.8 ns. (b) Inversion process of calculating A_4^{-1} . After 2 iterations, all 4 outputs converge, indicating an inversion time of 1.2 ns. (c) Inversion process of calculating A_5^{-1} . After 2 iterations, all 4 outputs converge, indicating an inversion time of 1.2 ns.

1. Nikkhah, V. *et al.* Inverse-designed low-index-contrast structures on a silicon photonics platform for vector–matrix multiplication. *Nat. Photon.* **18**, 501–508 (2024).
2. SeyedinNavadeh, S. *et al.* Determining the optimal communication channels of arbitrary optical systems using integrated photonic processors. *Nat. Photon.* **18**, 149–155 (2024).
3. Larsson, E. G., Edfors, O., Tufvesson, F. & Marzetta, T. L. Massive MIMO for next generation wireless systems. *IEEE Communications Magazine* **52**, 186–195 (2014).
4. Lu, L., Li, G. Y., Swindlehurst, A. L., Ashikhmin, A. & Zhang, R. An Overview of Massive MIMO: Benefits and Challenges. *IEEE Journal of Selected Topics in Signal Processing* **8**, 742–758 (2014).
5. Soref, R. & Bennett, B. Electrooptical effects in silicon. *IEEE Journal of Quantum Electronics* **23**, 123–129 (1987).
6. Lin, S. *et al.* Efficient, tunable flip-chip-integrated III-V/Si hybrid external-cavity laser array. *Opt. Express, OE* **24**, 21454–21462 (2016).
7. Roelkens, G. *et al.* III-V/Si photonics by die-to-wafer bonding. *Materials Today* **10**, 36–43 (2007).
8. Liang, D. *et al.* Low-Temperature, Strong SiO₂-SiO₂ Covalent Wafer Bonding for III–V Compound Semiconductors-to-Silicon Photonic Integrated Circuits. *J. Electron. Mater.* **37**, 1552–1559 (2008).
9. Zhang, J. *et al.* III-V-on-Si photonic integrated circuits realized using micro-transfer-printing. *APL Photonics* **4**, 110803 (2019).
10. Shi, Y. *et al.* Optical pumped InGaAs/GaAs nano-ridge laser epitaxially grown on a standard 300-mm Si wafer. *Optica, OPTICA* **4**, 1468–1473 (2017).
11. Scherrer, M. *et al.* In-Plane Monolithic Integration of Scaled III-V Photonic Devices. *Applied Sciences* **11**, 1887 (2021).
12. Gu, Z. *et al.* Optical transmission between III-V chips on Si using photonic wire bonding. *Opt. Express, OE* **23**, 22394–22403 (2015).
13. Billah, M. R. *et al.* Hybrid integration of silicon photonics circuits and InP lasers by photonic wire bonding. *Optica, OPTICA* **5**, 876–883 (2018).
14. Matsuo, S. *et al.* Directly modulated buried heterostructure DFB laser on SiO₂/Si substrate fabricated by regrowth of InP using bonded active layer. *Opt. Express, OE* **22**, 12139–12147 (2014).
15. Jiao, Y. *et al.* Indium Phosphide Membrane Nanophotonic Integrated Circuits on Silicon. *Phys. Status Solidi A* **217**, 1900606 (2020).
16. Jones, R. *et al.* Heterogeneously Integrated InP/Silicon Photonics: Fabricating Fully Functional Transceivers. *IEEE Nanotechnology Magazine* **13**, 17–26 (2019).
17. Matsui, Y. *et al.* Narrow linewidth tunable semiconductor laser. in *2016 Compound Semiconductor Week (CSW) [Includes 28th International Conference on Indium Phosphide & Related Materials (IPRM) & 43rd International Symposium on Compound Semiconductors (ISCS)* 1–2 (2016). doi:10.1109/ICIPRM.2016.7528538.

Reviewer #2 (Remarks to the Author)

The authors combine the advantage of executing operations in a fully-optical domain and iterative matrix inversion to empower the computation-to-IO ratio. This is a newly implemented concept and seems to find application in many domains.

The authors have extensively answered to my comments and doubts in a proper way, so I am satisfied with the answers.

Our response: Thanks for acknowledging the novelty of the PIP architecture in enhancing the IO efficiency for matrix-inversion-intensive tasks. Based on our experimental and simulation results and analyses on the scalability of the PIP, we believe the PIP technology will be a very appealing approach for matrix-inversion-intensive tasks.

REVIEWERS' COMMENTS

Reviewer #1 (Remarks to the Author):

The author has addressed most of my concerns, thus I recommend its publication.